


# Urban Aerosol Size Distributions: A Global Perspective

Tianren Wu[1,2] and Brandon E. Boor[1,2*]

[1]Lyles School of Civil Engineering, Purdue University, 550 Stadium Mall Drive, West Lafayette, Indiana 47907, United States

[2]Ray W. Herrick Laboratories, Center for High Performance Buildings, Purdue University, 177 South Russell Street, West Lafayette, Indiana 47907, United States

**Correspondence:** Brandon E. Boor (bboor@purdue.edu)

## Abstract

Urban aerosol measurements are necessary to establish associations between air pollution and human health outcomes and to evaluate the efficacy of air quality legislation and emissions standards. The measurement of urban aerosol particle size distributions (PSDs) is of particular importance as they enable for characterization of size-dependent processes that govern a particle's transport, transformation, and fate in the urban atmosphere. PSDs also improve our ability to link air pollution to health effects through evaluation of particle deposition in the respiratory system and inhalation toxicity. To provide guidance for the evolution of urban aerosol observations, this paper reviews and critically analyzes the current state-of-knowledge on urban aerosol PSD measurements by synthesizing $n$=793 PSD observations made between 1998 to 2017 in $n$=125 cities in $n$=51 countries around the globe. Significant variations in the shape and magnitude of urban aerosol number and mass PSDs were identified among different geographical regions. In general, number PSDs in Europe (EU), North America, Australia, and New Zealand (NAAN) are dominated by nucleation and Aitken mode particles. PSDs in Central, South, and Southeast Asia (CSSA) and East Asia (EA) are shifted to larger sizes, with a meaningful contribution from the accumulation mode. Urban mass PSDs are typically bi-modal, presenting a dominant peak in the accumulation mode and a secondary peak in the coarse mode. Most PSD observations published in the literature are short-term, with only 14% providing data for longer than six months. There is a paucity of PSDs measured in Africa (AF), CSSA, Latin America (LA), and West Asia (WA), demonstrating the need for long-term aerosol measurements across wide size ranges in many cities around the globe.

Inter-region variations in PSDs have important implications for population exposure, driving large differences in the urban aerosol inhaled deposited dose rate received in each region of the human respiratory system. Similarly, inter-region variations in the shape of PSDs impact the penetration of urban aerosols through filters in building ventilation systems, which serve as an important interface between the outdoor and indoor atmospheres. Geographical variations in urban aerosol effective densities were also reviewed. Size-resolved urban aerosol effective density functions from 3 to 10,000 nm were established for different geographical regions and intra-city sampling locations in order to accurately translate number PSDs to mass PSDs, with significant variations observed between near-road and urban background sites. The results of this critical review demonstrate that global initiatives are urgently needed to develop infrastructure for routine and long-term monitoring of urban





aerosol PSDs spanning the nucleation to coarse modes. Doing so will advance our understanding of spatiotemporal trends in urban PSDs throughout the world and provide a foundation to more reliably elucidate the impact of urban aerosols on atmospheric processes, human health, and climate.

## 1 Introduction

Urban air pollution is a major global environmental health challenge. Aerosols are a key constituent of urban air pollution and include a diverse mixture of liquid and solid particles spanning in size from a single nanometer to tens of micrometers. Urban aerosol measurements are critical for monitoring the extent of urban air pollution, identifying pollutant sources, understanding aerosol transport and transformation mechanisms, and evaluating human exposure and health outcomes (Asmi et al., 2011; Azimi et al., 2014; Harrison et al., 2011a; Hussein et al., 2003; Morawska et al., 1998; Peng et al., 2014; Shi et al., 1999; Shiraiwa et al., 2017; Vu et al., 2015; Wu et al., 2008). Human exposure to aerosols in urban environments is responsible for adverse health effects, including mortality and morbidity due to cardiovascular and respiratory diseases, asthma, and neural diseases (Allen et al., 2017; Burnett et al., 2018; Delfino et al., 2005; Meldrum et al., 2017; Oberdörster et al., 2005; Rychlik et al., 2019; Shiraiwa et al., 2017; Sioutas et al., 2005). Improved characterization of urban aerosols is needed to better understand the impact of aerosol exposure on human health and to evaluate the efficacy of current and future air quality legislation.

Of particular importance are measurements of urban aerosol particle size distributions (PSDs). Measurement of urban aerosol PSDs provides a basis for in-depth evaluation of size-resolved aerosol transport and transformation processes in the urban atmosphere, air pollution source apportionment, aerosol deposition in the human respiratory system, and associated toxicological effects on the human body. Despite the atmospheric and health relevance of urban PSDs, long-term aerosol measurements are often focused on size-integrated concentration metrics, such as $PM_{2.5}$, that lack essential size-resolved information. While urban aerosol PSD measurements have been conducted in cities around the globe, they are often short in duration and not performed as part of routine air quality monitoring. Urban PSDs provide a more complete assessment of an aerosol population, beyond what can be achieved with size-integrated metrics. Of particular importance are urban PSDs that capture the ultrafine particle regime (UFP, 1 to 100 nm). UFPs tend to dominate number PSDs, penetrate deep into the lung, and are associated with various deleterious human health outcomes.

Presently, there are no comprehensive literature reviews synthesizing urban aerosol PSD observations from around the globe in order to identify geospatial trends in the structure of number and mass PSDs. Previous literature reviews of urban aerosol PSDs have focused on major emission sources and source apportionment techniques (Vu et al., 2015) and the implications of urban aerosol PSDs on indoor air quality (Azimi et al., 2014). There has been a large number of urban aerosol PSD



observations conducted over the past few decades. The objective of this critical review is to provide a comprehensive overview of urban aerosol PSD observations from around the globe.

This study reviews $n=793$ urban aerosol PSD observations from $n=125$ cities in $n=51$ countries measured between 1998 to 2017. Urban aerosol PSD data spanning the nucleation to coarse modes (3 to 10,000 nm) was extracted from the literature, fit
to multi-modal lognormal distribution functions, and agglomerated by geographical region in order to identify trends in the physical characteristics of aerosol populations from different regions. This represents the first attempt, to the authors' knowledge, to understand geographical variations in urban aerosol PSDs at a global scale. The geographical distribution of measurement locations and the categorization of the collected PSDs enables for identification of gaps in urban aerosol PSD measurements. This will help motivate future research efforts and frame forthcoming urban air pollution measurement needs.
Along with urban aerosol PSDs, size-resolved urban aerosol effective densities were also reviewed. The effective density is an important aerosol morphological parameter that provides a basis to reliably translate measured number PSDs to mass PSDs. The implications of geographical variations in urban aerosol PSDs on human inhalation exposure and aerosol filtration in buildings were evaluated to demonstrate the broad utility of routine PSD monitoring.

## 2 Methodology for establishing the current state-of-knowledge on urban aerosol PSD observations

An expansive literature search was conducted on short- and long-term stationary and mobile measurements of urban aerosol number and mass PSDs between 1998 and 2017. The aim was to capture any potentially relevant peer-reviewed resources in which urban aerosol PSDs have been reported. Two academic search indices, Web of Science and ScienceDirect, along with Google Scholar, were used to conduct the literature search. Search terms included: urban aerosol, particle size distribution, urban aerosol size distribution, aerosol size distribution, urban particulate matter, scanning mobility particle sizer urban,
differential mobility particle sizer urban, and urban aerosol MOUDI, among others. Approximately 3,400 peer-reviewed journal articles and reports were initially screened to determine if they contained suitable information. Approximately 200 of them, which reported urban or semi-urban aerosol PSDs in the sub-micron regime (< 1000 nm), with some also covering the coarse regime (> 1000 nm), were selected for detailed analysis (Fig. 1). These articles presented $n=793$ individual PSDs (182 of which covered both the sub-micron and coarse regime) from $n=125$ cities in $n=51$ countries around the globe (Table 1).
Most PSDs reported number-based concentrations (e.g. measured with a Scanning Mobility Particle Sizer (SMPS) or Aerodynamic Particle Sizer (APS)), while some report mass-based concentrations (e.g. measured by inertial impactors) or columnar volume concentrations (e.g. measured by sun/sky radiometers). For PSDs reported only in the form of figures without lognormal fitting parameters, as was most common among the references, the GRABIT tool in MATLAB (The MathWorks, Inc., Natick, MA, USA) and WebPlotDigitizer (https://automeris.io/WebPlotDigitizer, Version 4.0) were utilized
to extract the data points of the PSDs. The PSDs were subsequently reproduced in MATLAB. A consistent particle size definition, the electrical mobility diameter ($D_{em}$), was used for all PSDs, as described in Sect. 3 and 4.



The PSDs were classified by geographical region: Africa (AF), Central, South, and Southeast Asia (CSSA), East Asia (EA), Europe (EU), Latin America (LA), North America, Australia, and New Zealand (NAAN), and West Asia (WA) (Table 1). The PSDs in each geographical region were separated into two site types depending on the measurement location within the city. 'Urban' indicates that the measurement was conducted in urban areas that are not strongly affected by localized traffic emissions. 'Traffic (near-road)' indicates that the environment was strongly influenced by traffic emissions, e.g. street canyon or roadside.

## 3 Methodology for evaluating geographical trends in size-resolved urban aerosol effective densities

### 3.1 Introduction to size-resolved urban aerosol effective density functions

Evaluation of geographical variations in size-resolved aerosol morphological features is needed to better characterize urban aerosol populations around the world. This section outlines the development of size-resolved urban aerosol effective density ($\rho_{eff}$) functions from $D_{em}$ = 3 to 10000 nm. Size-dependent differences in the aerosol particle density ($\rho_p$) and dynamic shape factor ($\chi$) are best captured together through $\rho_{eff}$. The $\rho_{eff}$ functions serve three purposes in this study: (1.) to translate urban aerosol number PSDs to mass PSDs, (2.) to convert aerodynamic diameter ($D_a$)-based PSDs to $D_{em}$-based PSDs, and (3.) to provide a summary of $\rho_{eff}$ measurements in the urban atmosphere. (2.) is necessary to enable a consistent particle size definition to be used in compiling PSD observations from around the globe, as described in Sect. 4. The size-resolved $\rho_{eff}$ functions include a combination of direct measurements of $\rho_{eff}$ in the urban atmosphere (Sect. 3.2) and approximations for size fractions where direct measurements have not yet been reported in the literature, such as sub-10 nm and coarse mode particles (Sect. 3.3-3.5). The integration of the size-resolved $\rho_{eff}$ functions with the urban aerosol PSD observations is presented in Sect. 3.6.

Different definitions of $\rho_{eff}$ have been used in previous studies (DeCarlo et al., 2004). In the current study, $\rho_{eff}$ is defined as the ratio of the measured particle mass ($m_p$) to the volume calculated from $D_{em}$ assuming spheres (DeCarlo et al., 2004; Hu et al., 2012; McMurry et al., 2002; Qiao et al., 2018) (Eq. 1):

$$\rho_{eff} = \frac{6m_p}{\pi D_{em}^3}, \tag{1}$$

Only empirical $\rho_{eff}$ values defined in this manner were collected from the literature. The particle volume ($V_p$) is defined as the volume taken up by all of the solid and liquid material in the particle and void space enclosed within the particle envelope (DeCarlo et al., 2004). For an irregular particle, the volume equivalent diameter ($D_{ve}$) represents the diameter of a sphere that has the same volume as $V_p$ (DeCarlo et al., 2004; Hinds, 2012; Seinfeld and Pandis, 2012) (Eq. 2):

$$V_p = \frac{\pi}{6} D_{ve}^3, \tag{2}$$

The ratio of $m_p$ to $V_p$ is referred to as the particle density ($\rho_p$), as shown in Eq. (3):





$$\rho_p = \frac{6m_p}{\pi D_{ve}^3}, \tag{3}$$

The relationship between $D_{em}$ and $D_{ve}$ is given by DeCarlo et al. (2004) and Seinfeld and Pandis (2012):

$$\frac{D_{em}}{C_c(D_{em})} = \frac{D_{ve}\chi}{C_c(D_{ve})}, \tag{4}$$

where $C_c$ is the Cunningham Slip Correction Factor (Allen and Raabe, 1982, 1985; Hinds, 2012).

### 3.2 Urban aerosol effective densities: summary of direct measurements

Direct measurements of $\rho_{eff}$ in urban environments are limited. However, sufficient data is available in the literature to identify trends in $\rho_{eff}$ among geographical regions and intra-city site types (urban or traffic). Size-resolved urban aerosol $\rho_{eff}$ values were extracted from $n$=9 studies conducted in Denmark, China, United States, and Finland (Table 2). The studies report direct

measurements of $\rho_{eff}$, primarily in the sub-micron regime through use of various aerosol instrument configurations, such as those evaluating the mass-mobility relationship of an aerosol population through a Differential Mobility Analyzer (DMA)-Aerosol Particle Mass Analyzer (APM) system (e.g. McMurry et al. 2002). The measured $\rho_{eff}$ values and measurement information, including measurement technique, duration, and site (city, country), are summarized in Table 2.

As the PSD and direct $\rho_{eff}$ measurements were conducted in different cities and at different site types within the city, it is reasonable to apply $\rho_{eff}$ values which were measured under a condition consistent with a PSD measurement when converting number PSDs to mass PSDs and $D_a$ to $D_{em}$. In order to apply the most reasonable $\rho_{eff}$ to a PSD observation, the collected size-resolved $\rho_{eff}$ values were divided into three groups according to the geographical region where the direct measurement was conducted and the site type (urban or traffic). Direct measurements conducted in China in 'urban' environments were

incorporated into Group A. Direct measurements in the United States in 'urban' environments were incorporated into Group B. Direct measurements conducted in the United States, Finland, and Denmark in 'traffic' environments were incorporated into Group C. None of the direct $\rho_{eff}$ measurements in China were conducted in 'traffic' environments. The $\rho_{eff}$ values for the size range of 3200 to 5600 nm reported by Hu et al. (2012) were not included as the high $\rho_{eff}$ values associated with the abundance of minerals in coarse particles in Beijing might bias the analysis (Guo et al., 2010; Zhang et al., 2010).

Representative size-resolved $\rho_{eff}$ functions were estimated for Groups A, B, and C (Fig. 9). For the direct $\rho_{eff}$ measurements tabulated in Table 2, the collected values were often reported as a function of particle size discretely. In order to convert them to a continuous $\rho_{eff}$ function with respect to size, which can be easily applied when converting number PSDs to mass PSDs and $D_a$ to $D_{em}$, a few assumptions were made. For particles greater than 10 nm, if the particles at a certain diameter $D_{em,1}$ were

reported to have an effective density of $\rho_{eff,1}$, we assume the particles in the size range from ($D_{em,1}$-50 nm) to ($D_{em,1}$+50 nm) also have the same effective density of $\rho_{eff,1}$. If the particles at diameter $D_{em,2}$ ($D_{em,2}$>$D_{em,1}$) were reported to have an effective density of $\rho_{eff,2}$ in the same study and $D_{em,2}$ is within the size range from $D_{em,1}$ to ($D_{em,1}$+50 nm), we assume the particles with the size from ($D_{em,1}$-50 nm) to ($D_{em,1}$+$D_{em,2}$)/2 to have the effective density of $\rho_{eff,1}$, while the particles with the size from


$(D_{em,1}+D_{em,2})/2$ to $(D_{em,2}+50\ nm)$ have the effective density of $\rho_{eff,2}$. By doing this, we obtained several continuous size-resolved $\rho_{eff}$ functions ranging from approximately 10 nm to several hundred nanometers. We then took the mean of the size-resolved $\rho_{eff}$ values derived from direct measurements in each of the three groups, illustrated as blue lines in the gray regions of Fig. 9.

**3.3 Urban aerosol effective densities: considerations for sub-10 nm particles without direct measurements**

Direct measurements of $\rho_{eff}$ from $D_{em} = 3$ to 10 nm have not been previously reported in the literature. Particles in this size
range are typically formed by homogeneous or heterogeneous nucleation, which can involve sulfuric acid ($H_2SO_4$), amines, ammonia, and organic vapors (e.g. Kulmala et al., 2013). $H_2SO_4$ and highly oxygenated molecules (HOMs) or extremely low volatility organic compounds (ELVOCs) are often involved in the nucleation and initial growth of particles during an atmospheric new particle formation event. Previous studies have assumed nucleated particles formed in experimental chambers in the size range of 4 to 12 nm to have a density of 1.5 g cm$^{-3}$ (Wang et al., 2010). ELVOCs and HOMs are assumed to have a density of 1.5 g cm$^{-3}$ and 1.4 g cm$^{-3}$, respectively (Ehn et al., 2014; Tröstl et al., 2016). For simplicity, we used the
$\rho_p$ of condensed ELVOCs (Ehn et al., 2014) (1.5 g cm$^{-3}$) and the condensed phase density of $H_2SO_4$ (Xiao et al., 2015) (1.83 g cm$^{-3}$) to estimate the lower and upper limits of $\rho_{eff}$ for particles from $D_{em} = 3$ to 10 nm, assuming the particles adopt a spherical shape with $\chi = 1$. The mean value of the two limits was used as the representative $\rho_{eff}$ for all three groups (dark red lines in light blue regions of Fig. 9). Despite uncertainties in the assumed $\rho_{eff}$ values, particles from $D_{em} = 3$ to 10 nm contribute negligibly to particle mass concentrations and are seldom measured with aerodynamic-based techniques, thus conversion from
$D_a$ to $D_{em}$ is often unnecessary.

**3.4 Urban aerosol effective densities: considerations for accumulation mode particles without direct measurements**

There is a lack of direct measurements of $\rho_{eff}$ in Groups B and C for particles greater than approximately 400 nm in size. Pitz et al. (2008) conducted measurements of apparent particle density for particles with a $D_a \leq 2500$ nm (PM2.5) at an urban background site in Germany. The apparent particle density ranged from 1.05 to 2.36 g cm$^{-3}$, with a mean value of 1.65 g cm$^{-3}$.
As particles in the accumulation mode strongly contribute to urban aerosol mass concentrations (e.g. PM1, PM2.5), we assume the mean apparent particle density of 1.65 g cm$^{-3}$ to be the representative $\rho_{eff}$ for particles in the size range from $D_{em} = 400$ to 2500 nm in Group B (dark red line in yellow region of Fig. 9). Rissler et al. (2014) measured the $\rho_{eff}$ of urban aerosols from 50 to 400 nm in a street canyon in central Copenhagen and identified two different groups of aerosols with distinctive $\rho_{eff}$: loose chain-like soot aggregates and more dense particles. The $\rho_{eff}$ of the dense particles from $D_{em} = 50$ to 400 nm was in
the range of 1.3 to 1.65 g cm$^{-3}$, of which the main constituents were inorganic salts, such as sulfate, nitrate, and ammonium, along with organics. Previous studies that have investigated the chemical composition of near-road aerosols indicated that the mass fraction of black carbon (elemental carbon) to the total mass of particles from $D_{em} = 400$ to 1000 nm was 6.4 to 26.7% (Brüggemann et al., 2009; Daher et al., 2013; Fushimi et al., 2008; Massoli et al., 2012; Song et al., 2012). Here, we assume that loose chain-like soot aggregates do not contribute significantly to particle mass concentrations in this size range and
applied the mean value of $\rho_{eff}$ for 'dense' particles as measured by Rissler et al. (2014) as the representative $\rho_{eff}$ for particles





from $D_{em}$ = 400 to 1000 nm in Group C (dark red line in orange region of Fig. 9). Although soot particles, which have a $\rho_{eff}$ less than that of the dense particles, exist in the size range of $D_{em}$ = 400 to 1000 nm, denser components, such as organics, mineral dusts, and crustal materials, may also exist in this size range.

**3.5 Urban aerosol effective densities: considerations for coarse mode particles without direct measurements**

Direct measurements of urban aerosol $\rho_{eff}$ have been rarely conducted in the coarse mode, in part due to the difficulty of extending mass-mobility measurements (e.g. DMA-APM) to this size range. Coarse particles can be composed of organics, ions, dusts, crustal material, brake and tire wear, sea salt, black carbon, and other trace elements (Brüggemann et al., 2009; Cheung et al., 2011; Cozic et al., 2008; Daher et al., 2013; Kim et al., 2003; Koçak et al., 2007; Koulouri et al., 2008; Pakkanen et al., 2001; Song et al., 2012). These materials are associated with a wide range in $\rho_p$. Secondary organic aerosol (SOA) synthesized in the laboratory are reported to have $\rho_p$ ranging from 1 to 1.65 g cm$^{-3}$, depending on formation conditions and gas-phase precursors (Kostenidou et al., 2007; Malloy et al., 2009; Nakao et al., 2013; Zelenyuk et al., 2008). The $\rho_p$ of $(NH_4)_2SO_4$ and $NH_4NO_3$, which represent sulfate and nitrate compounds dominant in ionic mass, were reported to be approximately 1.7 g cm$^{-3}$ (Lide, 2005; Mikhailov et al., 2013; Neusüß et al., 2002; Tang, 1996). The $\rho_p$ of dust varies with the associated components, including amorphous silicon oxide (2.1 to 2.3 g cm$^{-3}$), illite/muscovite (2.7 to 3.1 g cm$^{-3}$), montmorillonite (2.2 to 2.7 g cm$^{-3}$), and quartz (2.65 g cm$^{-3}$) (Reid et al., 2003). The inherent material density of diesel soot was measured to be 1.77 g cm$^{-3}$ (Park et al., 2004).

The shape of coarse mode particles is often composition-dependent. The value of $\chi$ depends on the shape of the particle. For a sphere, $\chi$ = 1; for a cylinder with an axial ratio of 2, $\chi$ = 1.1; for a cylinder with an axial ratio of 5, $\chi$ = 1.35; and for a compact cluster of four spheres, $\chi$ = 1.17 (Hinds, 2012). Some studies indicate that SOA has a spherical shape, suggestive of $\chi \sim 1$ (Abramson et al., 2013; Pajunoja et al., 2014; Virtanen et al., 2010). Coarse mode dust particles often exhibit large values of $\chi$. The $\chi$ of Saharan mineral dusts measured in Morocco with a $D_{em}$ = 1200 nm is 1.25 (Kaaden et al., 2009). Davies (1979) reported $\chi$ to be 1.36 to 1.82 for quartz, 2.04 for talc, and 1.57 for sand. Soot particles typically exist as porous agglomerates, however, they can transform to spheres over several hours by condensation of $H_2SO_4$ (Happonen et al., 2010; Pagels et al., 2009; Rissler et al., 2013, 2014). The complicated mixing state of urban aerosols introduces additional uncertainties in estimating the $\rho_{eff}$ for coarse mode particles (Riemer et al., 2019).

$\rho_{eff}$ values were estimated for coarse mode particles in Group C, for particles larger than 2500 nm in Group B, and for particles larger than 3200 nm in Group A. The $\rho_p$ and $\chi$ of three types of aerosols, including inorganic aerosol, SOA, and mineral dust, were used to estimate a range of values for $\rho_{eff}$. $(NH4)_2SO_4$ and $NH_4NO_3$, with a $\rho_p$ of approximately 1.7 g cm$^{-3}$ and a $\chi$ of 1.01 (Hudson et al., 2007; Lide, 2005; Mikhailov et al., 2013; Neusüß et al., 2002; Tang, 1996), were selected as the representative inorganic aerosol to calculate $\rho_{eff}$. Nearly spherical SOA was assumed to adopt $\rho_{eff}$ of 1 to 1.65 g cm$^{-3}$ (Kostenidou et al., 2007; Malloy et al., 2009; Nakao et al., 2013; Zelenyuk et al., 2008). Illite, kaolinite, and montmorillonite were chosen to represent





mineral dust, with $\rho_p$ of 2.7 to 3.1 g cm$^{-3}$, 2.6 g cm$^{-3}$, and 2.2 to 2.7 g cm$^{-3}$, and with $\chi$ of 1.3, 1.05, and 1.11, respectively (Hudson et al., 2007, 2008; Reid et al., 2003).

Equation (4) presents the relationship between $D_{em}$ and $D_{ve}$. For coarse mode particles, the value of $\frac{C_C(D_{ve})}{C_C(D_{em})}$ is approximately unity, such that the Cunningham slip correction factors in Eq. (4) can be reasonably neglected. Therefore, Eq. (4) becomes $D_{em} = D_{ve}\chi$. Plugging this into Eq. (1), we arrive at Eq. (5):

$$\rho_{eff} = \frac{6m_p}{\pi\chi^3 D_{ve}^3},\tag{5}$$

Combining Eq. (3) and (5), we can derive Eq. (6), which describes the relationship between $\rho_{eff}$, $\rho_p$, and $\chi$ for coarse mode particles (an example is given in Fig. 2):

$$\rho_{eff} = \frac{\rho_p}{\chi^3},\tag{6}$$

With the different combinations of $\rho_p$ and $\chi$ described above, Eq. (6) was applied to calculate $\rho_{eff}$ values for the three types of aerosols. The values span from 1 to approximately 2 g cm$^{-3}$ (light green region of Fig. 9). The $\rho_{eff}$ values within this range are
35 used as the representative $\rho_{eff}$ for coarse mode particles in Group C, for particles larger than 2500 nm in Group B, and for particles larger than 3200 nm in Group A.

**3.6 Integration of size-resolved urban aerosol effective density functions with urban aerosol PSD observations**

The combination of directly measured and estimated $\rho_{eff}$ values between $D_{em} = 3$ to 10000 nm provides a basis to establish continuous, size-resolved urban aerosol $\rho_{eff}$ functions for Groups A ($\rho^A_{eff}$), B ($\rho^B_{eff}$), and C ($\rho^C_{eff}$), as illustrated in Fig. 9. When
converting number PSDs to mass PSDs and $D_a$ to $D_{em}$, $\rho^A_{eff}$ was applied to number PSDs measured in the 'urban' environment in cities in CSSA, WA, LA, AF, and China. $\rho^B_{eff}$ was applied to number PSDs measured in the 'urban' environment in cities in EU, NAAN, Japan, and Korea. $\rho^C_{eff}$ was applied to number PSDs measured in the 'traffic' areas in cities around the globe, excluding China.

$\rho^A_{eff}$ was applied to the number PSDs measured at both urban and traffic sites in China. The urban PSD measurements in China collected in this study were mainly from megacities, such as Beijing, Shanghai, and Guangzhou. Heavy-duty diesel trucks are prohibited to enter many urban areas during the daytime in these megacities (Wu et al., 2008). In addition, the fraction of diesel-powered cars in cities in China are much lower than that in Europe or North America. It is shown that gasoline-powered passenger cars contribute 91% of the total amount of vehicles in Beijing (Wu et al., 2008). Therefore, the relative fraction of
soot particles in the near-road region was expected to be lower than that in North America and Europe. Previous studies suggest that the contribution of black carbon to PM$_{2.5}$ mass concentrations was less than 4% in Shanghai (Cao et al., 2013; Feng et al., 2009). In addition, soot particles from gasoline engine vehicles might be more 'compact' due to the relatively sulfur 'rich' fuel used in China (Yin et al., 2015). Soot particles may also be heavily aged or internally mixed with other





condensable materials due to the higher pollution levels in megacities, resulting in a higher $\rho_{eff}$ than freshly emitted soot particles. Huang et al. (2013) indicated that the number fraction of pure black carbon was 1.9% in Shanghai during polluted periods. Therefore, the near-road size-resolved $\rho_{eff}$ was assumed to be closer to the values compiled for $\rho^A_{eff}$, rather than those for $\rho^C_{eff}$.

## 4 Methodology for analyzing urban aerosol PSD observations

### 4.1 Introduction to multi-modal lognormal fitting and transformations of urban aerosol PSDs

The urban aerosol PSDs were fit to the multi-modal lognormal distribution function and translated across number (cm$^{-3}$), surface area ($\mu$m$^2$ cm$^{-3}$), volume ($\mu$m$^3$ cm$^{-3}$), and mass ($\mu$g m$^{-3}$) domains following different strategies depending on the measurement technique and size range, as described in Sect. 4.1.1-4.1.4. Lognormal fitting parameters, including the geometric mean diameter, geometric standard deviation, and concentration for each mode, along with measurement information, for all $n$=793 PSDs are compiled in the Supplement (Tables S1-S5 and individual PSD figures). For a few selected studies where the measured PSDs were already fit to the multi-modal lognormal distribution function, the listed fitting parameters were used directly. The fitting parameters provide a basis to characterize the shape and magnitude of the PSDs, as well as a mathematical parameterization to re-produce the measured PSDs for subsequent analysis by the atmospheric aerosol research community.

### 4.1.1 Urban aerosol number PSDs in the sub-micron regime

Urban aerosol number PSDs that only included modes in the sub-micron regime (nucleation, Aitken, and accumulation) were typically measured via electrical mobility-based techniques as number-based concentrations (e.g. SMPS). The extracted measured data for these PSDs were fit to the multi-modal lognormal distribution function ($dN/dLogD_p$, cm$^{-3}$, Eq. 7) by using a lognormal fitting code in MATLAB based on the nonlinear least-squares curve fitting function, lsqcurvefit.m:

$$\frac{dN}{dLogD_p} = \sum_{i=1}^{n} \frac{N_i}{(2\pi)^{1/2} log\,(\sigma_i)} exp\left[-\frac{(log\,D_p - log\,\overline{D_{p,i}})^2}{2log^2(\sigma_i)}\right],\tag{7}$$

The geometric mean diameter ($\overline{D_{p,i}}$), geometric standard deviation ($\sigma_i$), and particle number concentration or amplitude ($N_i$) for each mode ($i$) were determined. The number of modes was based upon what was needed to achieve the best fit to the measured data. Fitting parameters and measurement information for urban aerosol number PSDs in the sub-micron regime are provided in Table S1, as well as in individual PSD figures presented in the Supplement, an example of which is shown in Fig. 3. The fitted number PSDs were converted to surface area PSDs ($dS/dLogD_p$, $\mu$m$^2$ cm$^{-3}$) and volume PSDs ($dV/dLogD_p$, $\mu$m$^3$ cm$^{-3}$) assuming spherical particles and converted to mass PSDs ($dM/dLogD_p$, $\mu$g m$^{-3}$) using the representative size-resolved $\rho_{eff}$ functions for Groups A, B, or C (Sect. 3, Fig. 9). Size-integrated concentrations were also calculated.




#### 4.1.2 Urban aerosol number PSDs that cover both the sub-micron and coarse regimes

Urban aerosol number PSDs that cover both the sub-micron and coarse regimes typically utilize a combination of different aerosol measurement techniques. Electrical mobility-based techniques (e.g. SMPS) are often used for the sub-micron regime and aerodynamic-based techniques (e.g. APS) or optical-based techniques (e.g. Optical Particle Counter (OPC) or Optical Particle Sizer (OPS)) are used for the coarse regime and a fraction of the accumulation mode. The amplitude of number PSDs can span one to three orders of magnitude over the sub-micron and coarse regimes. When both regimes are measured concurrently, the coarse mode is often present as the tail of the accumulation mode in the number PSDs. Thus, the lognormal fitting of the number PSDs often ignores the coarse mode particles, which can result in an inaccurate estimation of the volume and mass concentrations of large particles when converting the fitted number PSDs to volume and mass PSDs. Therefore, the PSDs that cover both the sub-micron and coarse regimes were separated into two segments, each of which was individually fitted to the multi-modal lognormal distribution function in order to reproduce the measured data more accurately.

Electrical mobility-based techniques typically cover size fractions that contribute significantly to number PSDs in the urban atmosphere (e.g. $D_{em} \leq 100$ nm). Thus, the segment of the number PSDs measured by such techniques were directly fitted with the multi-modal lognormal distribution function by Eq. (7), and the fitted number PSDs were converted to surface area, volume, and mass PSDs as described in Sect. 4.1.1. Size fractions measured by aerodynamic- or optical-based techniques often cover a fraction of the accumulation mode and the coarse mode, both of which contribute significantly to volume and mass PSDs. Therefore, to best reproduce the volume and mass PSDs, a different approach was employed to fit the PSDs measured by these two techniques.

For aerodynamic-based measurements, $D_a$ needs to be converted to $D_{em}$ so that a consistent particle size definition can be used. In some studies, the authors did this by converting the measured $D_a$-based PSD to a $D_{em}$-based PSD using the value for $\rho_{eff}$ that gave the best fit between the converted $D_{em}$-based PSD with the measured $D_{em}$-based PSD in an overlap region (often the accumulation mode) that was covered by both electrical mobility- and aerodynamic-based techniques (e.g. Pitz et al., 2008a). Most urban aerosol number PSDs measured with an APS were reported as $D_a$-based PSDs (e.g. Morawska et al., 1998) or converted to Stoke's or geometric diameter-based PSDs by assuming values for $\chi$ and $\rho_p$ (e.g. Babu et al., 2016; Bäumer et al., 2008; Wehner et al., 2004a; Wu et al., 2008; Yue et al., 2009, 2010). For the latter, such diameters were first converted back to $D_a$, and then to $D_{em}$. Equation (8) shows the relationship between $D_a$ and $D_{ve}$ (DeCarlo et al., 2004; Hinds, 2012), where $\rho_0$ is the standard density of 1 g cm⁻³:

$$D_{ve} = D_a \sqrt{\frac{\chi \rho_0 C_C(D_a)}{\rho_p C_C(D_{ve})}}, \tag{8}$$

For coarse mode particles, it is assumed that $\frac{C_C(D_{ve})}{C_C(D_a)}$ is approximately unity. As mentioned in Sect. 3.5, $D_{ve} = D_{em}/\chi$; combining this with Eq. (8), we arrive at Eq. (9):





$$\frac{\rho_p^{1/2}}{\chi^{3/2}\rho_0^{1/2}} = \frac{D_a}{D_{em}},$$ (9)

From Eq. (9) and (6), we can derive the relationship between $D_a$ and $D_{em}$ (Eq. 10), which is used to convert $D_a$ to $D_{em}$:

$$\rho_{eff} = \rho_0 (\frac{D_a}{D_{em}})^2,$$ (10)

As described in Sect. 3, a series of $\rho_{eff}$ values were generated from different combinations of $\chi$ and $\rho_p$ (Fig. 2 and light green region of Fig. 9) for a fraction of the coarse mode without direct $\rho_{eff}$ measurements in Groups A, B, and C. When converting

$D_a$ to $D_{em}$ for these particles, each $D_a$ was converted to multiple $D_{em}$ corresponding to a series of $\rho_{eff}$ to account for the uncertainty in $\rho_{eff}$. Therefore, for each of the $D_a$-based number PSDs in the size range where direct $\rho_{eff}$ data is lacking, a series of $D_{em}$-based number PSDs were determined. Figure 4 shows the ratio of $D_a$ to $D_{em}$ when translating between the two diameters for different values of $\chi$ and $\rho_p$ (assuming $\frac{C_C(D_{ve})}{C_C(D_a)} \approx 1$). In general, $D_a/D_{em}$ increases with decreasing $\chi$ and increasing $\rho_p$.

The converted $D_{em}$-based number PSDs were transformed to $D_{em}$-based volume PSDs. For the size fraction of the coarse mode where the $\rho_{eff}$ was estimated by different combinations of $\chi$ and $\rho_p$, multiple $D_{em}$-based volume PSDs were obtained from the series of $D_{em}$-based number PSDs. An example is shown in Fig. 5. We took the mean $D_{em}$-based volume PSD of that series of $D_{em}$-based volume PSDs, and then conducted the multi-modal lognormal fitting via Eq. (11):

$$\frac{dV}{dLogD_p} = \sum_{i=1}^{n} \frac{V_i}{(2\pi)^{1/2} \log(\sigma_i)} exp \left[ -\frac{(\log D_p - \log \overline{D_{p,i}})^2}{2\log^2(\sigma_i)} \right],$$ (11)

where $V_i$ is the volume concentration of mode $i$. The fitted $D_{em}$-based volume PSDs were then converted back to number and surface area $D_{em}$-based PSDs. This was done to prevent possible amplification in the difference between the measured and lognormally fitted PSDs when converting number PSDs to volume PSDs.

The converted $D_{em}$-based number PSDs were also transformed into $D_{em}$-based mass PSDs by applying the effective density

functions $\rho^A_{eff}$, $\rho^B_{eff}$, or $\rho^C_{eff}$, according to the measurement location and site type (Sect. 3, Fig. 9). For the size fraction of the coarse mode where the $\rho_{eff}$ was estimated by different combinations of $\chi$ and $\rho_p$, multiple $D_{em}$-based mass PSDs were obtained from a series of $D_{em}$-based number PSDs. Similar to the volume PSDs, we took the mean $D_{em}$-based mass PSD of that series of $D_{em}$-based mass PSDs, and then conducted the multi-modal lognormal fitting via Eq. (12):

$$\frac{dM}{dLogD_p} = \sum_{i=1}^{n} \frac{M_i}{(2\pi)^{1/2} \log(\sigma_i)} exp \left[ -\frac{(\log D_p - \log \overline{D_{p,i}})^2}{2\log^2(\sigma_i)} \right],$$ (12)

where $M_i$ is the mass concentration of mode $i$. Now the number, surface area, volume, and mass PSDs in the size range covered both the electrical mobility- and aerodynamic-based techniques can be reproduced. The goal of this stage is to apply the most appropriate size-resolved $\rho_{eff}$ to a given number PSD measurement, thereby accurately estimating its volume and mass PSD and taking into consideration the uncertainties of the unknown size-resolved $\rho_{eff}$ values for the coarse mode. Fitting parameters



and measurement information for urban aerosol PSD measurements made with both electrical mobility- and aerodynamic-based techniques covering the sub-micron and coarse regimes are provided in Table S2.

For optical-based measurements, we assume the optical diameter is equivalent to $D_{em}$ due to the lack of information needed to convert one to the other. The number PSDs measured by optical-based techniques were transformed to mass PSDs assuming a uniform apparent density of 1.65 g cm$^{-3}$ (Pitz et al., 2008b). The mass PSDs were then fitted with the multi-modal lognormal distribution function using Eq. (12). The fitted mass PSDs were converted back to number, surface area, and volume PSDs. Fitting parameters and measurement information for urban aerosol PSD measurements made with both electrical mobility- and optical-based techniques covering the sub-micron and coarse regimes are provided in Table S3.

### 4.1.3 Urban aerosol mass PSDs measured by gravimetric methods employing inertial impactors

For urban aerosol mass PSDs measured by gravimetric methods with inertial impactors, the $D_a$-based mass PSDs were converted to $D_{em}$-based mass PSDs to enable for comparison with the other electrical mobility-based measurements. According to the measurement location and site type, $D_a$ was converted to $D_{em}$ using the $\rho_{eff}$ functions for Groups A, B, or C, via Eq. (10). For the fraction of the coarse mode where a series of $\chi$ and $\rho_p$ were used to estimate $\rho_{eff}$, each $D_a$-based mass PSD was converted to multiple $D_{em}$-based mass PSDs, with each PSD corresponding to a particular value of $\rho_{eff}$. Then, the mean $D_{em}$-based PSD was taken from the series of $D_{em}$-based mass PSDs and merged with the rest of the $D_{em}$-based mass PSD determined via the effective density functions $\rho^A_{eff}$, $\rho^B_{eff}$, or $\rho^C_{eff}$. The multi-modal lognormal fitting was conducted for the $D_{em}$-based mass PSDs by using Eq. (12). An example is shown in Fig. 6.

The $D_a$-based mass PSDs were also converted to $D_{em}$-based volume PSDs by using the $\rho_{eff}$ functions for Groups A, B, or C. Similar to the conversion for mass PSDs, for the fraction of the coarse mode where a series of $\chi$ and $\rho_p$ were used to estimate $\rho_{eff}$, each $D_a$-based mass PSD was converted to multiple $D_{em}$-based volume PSDs. The mean $D_{em}$-based volume PSD was taken from this series of $D_{em}$-based volume PSDs and merged with the rest of the $D_{em}$-based volume PSD. The multi-modal lognormal fitting was conducted for the $D_{em}$-based volume PSDs by using Eq. (11). The fitted $D_{em}$-based volume PSDs were converted to $D_{em}$-based number and surface area PSDs. Fitting parameters and measurement information for urban aerosol PSD measurements made with gravimetric methods employing inertial impactors are provided in Table S4.

### 4.1.4 Urban aerosol columnar volume PSDs

Urban aerosol columnar volume PSDs measured by sun/sky radiometers were first converted to number PSDs assuming spherical particles. Then, the converted number PSDs and the columnar volume PSDs were separately fitted with multi-modal lognormal distribution functions, using Eq. (7) and (11), respectively. The fitted number PSDs were transformed to surface area PSDs and the fitted volume PSDs were transformed to mass PSDs assuming a uniform apparent density of 1.65 g cm$^{-3}$



(Pitz et al., 2008b). As the columnar volume PSDs do not present the absolute aerosol concentration, they were plotted in arbitrary units. Fitting parameters and measurement information for urban aerosol columnar volume PSD measurements made with sun/sky radiometers are provided in Table S5.

**4.1.5 Considerations for grouping urban aerosol PSD observations by geographical region**

The collection of urban aerosol PSD observations from around the globe offers a basis to identify geographical trends in
number and mass PSDs. A few considerations were made in grouping the PSDs by geographical region. For urban aerosol number PSDs, the prominent peak is most often present in the UFP regime, which is captured very well by electrical mobility-based techniques (e.g. SMPS). However, PSD measurements made with aerodynamic-based techniques (e.g. APS, inertial impactors), optical-based techniques (e.g. OPC, OPS), or sun/sky radiometers typically cannot accurately characterize number PSDs down to the UFP regime. Thus, when grouping urban aerosol number PSDs by geographical region, only measurements
made via electrical mobility-based techniques involving the UFP regime were used. More recent PSD observations of the sub-3 nm nanocluster aerosol mode made with diethylene glycol-based condensation particle counters, which can contribute significantly to sub-micron particle number concentrations, were not included in the global-scale analysis given the limited number of measurements that have been made (e.g. Rönkkö et al., 2017). For urban aerosol mass PSDs, the maximum value of the PSD typically exists in either the accumulation mode or the coarse mode. Therefore, urban aerosol PSD measurements
made via electrical mobility-based techniques that only cover the sub-micron regime were not used in the analysis of geographical trends in mass PSDs. Only PSD measurements made with inertial impactors and those combining both electrical mobility- and aerodynamic-/optical-based techniques to cover both the sub-micron and coarse regimes were incorporated into the global mass PSD analysis.

**4.2 Categorization and presentation of urban aerosol PSD observations in the Supplement**

The urban aerosol PSD observations were categorized by seven factors in order to better evaluate the shape and magnitude of an individual PSD and to provide a basis for historical interpretations of the PSDs: (1.) sampling location, (2.) sampling duration, (3.) time of day, (4.) event identification, (5.) target aerosol population, (6.) measurement type, and (7.) prominent mode. The categorization information is presented in the Supplement in Tables S1-S5 as acronyms, along with each individual PSD figure in the upper-left corner as: (1.) – (2.) – (3.) – (4.) – (5.) – (6.) – (7.) (e.g. Fig. 3, 5, 6). Intra-city sampling locations
00    include: traffic-influenced (TR), city center (CC), urban background (UB), sub-urban (SUB), and non-specific urban (NU). Non-specific urban represents an environment that is in the urban area, but all of the other indicators are not applicable. The sampling duration of the measurements were classified into several time-scales: long term (LT: > 6 months), moderate term (MT: 1 – 6 months), short term (ST: 1 week – 1 month), very short term (VST: 1 day – 1 week), and very, very short term (VVST: < 1 day). The time of the day includes: morning (M), afternoon (A), evening/night (E), daytime (D), and mean/median
05    of the day (ME). Event identification includes: non-specific event (NS), new particle formation (NPF), biomass burning (BB),



photochemical event (PC), combustion event (CB), dust storm (DS), and haze (HZ). The majority of PSDs were classified as NS as the reported observational period was not associated with a particular event. The target aerosol population indicates the size range covered by the measurement setup, including: UFP regime (UFP: $D_p \leq 100$ nm), fine regime (F: 100 nm $< D_p \leq$ 1000 nm), coarse regime (C: $D_p > 1000$ nm), UFP and fine regimes (UFP+F), fine and coarse regimes (F+C), and UFP regime

·10  with both fine and coarse regimes (UFP+F+C). The measurement type includes aerosol instruments based on electrical mobility classification (EM), aerodynamic sizing (A), optical sizing (O), sun/sky radiometers (RS), and gravimetric methods using inertial impactors (A-G). The prominent mode indicates the location of the prominent peak in the number PSDs, including the nucleation mode (NUC), Aitken mode (AIT), and accumulation mode (ACC). In some cases, the peak is present at the border of the nucleation and Aitken modes (NUC-AIT), or at the border of the Aitken and accumulation modes (AIT-

·15  ACC).

Individual PSD figures are compiled in the Supplement, examples of which are shown in Fig. 3, 5, 6. For number PSDs measured with a combination of electrical mobility- and aerodynamic-/optical-based techniques covering the sub-micron and coarse regimes (Tables S2, S3), the PSDs were divided into two segments as outlined in Sect. 4.1.2; thus, a separate figure

·20  was generated for each segment. Figure 3 is representative of sub-micron urban aerosol number PSDs measured with electrical mobility-based techniques that are included in the Supplement. The upper-left plot includes the measured/extracted data (black curve), which was fitted to the multi-modal lognormal distribution function. The lognormal fitting parameters are listed, with the colors corresponding to the individual mode, shown as dashed curves. The blue curve shows the sum of the individual modes. The size range over which the fitting is effective is listed. The fitted size range was not necessarily equivalent to the

·25  size range of the measurement. The estimated surface area, volume, and mass PSDs derived from the fitted number PSD are presented in the remaining plots. Size-integrated concentrations are also provided in the figure.

Figure 5 is representative of urban aerosol number PSDs spanning the accumulation and coarse modes measured with aerodynamic-based techniques that are included in the Supplement. The black dotted curve in the number PSD represents the

·30  measured data. The dotted line in the $D_a$-based volume PSD was converted from the measured $D_a$-based number PSD. The solid black lines in the volume and mass PSDs are $D_{em}$-based PSDs converted from the measured $D_a$-based number PSD. The rainbow color lines are the converted $D_{em}$-based volume and mass PSDs for a range of $\rho_{eff}$ to account for the uncertainties of $\rho_{eff}$ in the coarse regime. The relationship between the color and $\rho_{eff}$ is shown in the color bar. The dashed color lines are the lognormal fitting curves for each mode. The dashed black lines in the volume and mass PSDs are the sum of the fitted sub-

·35  modes. The fitted volume PSD was converted back to number and surface area PSDs, which are shown as the dashed black lines. The effective size range for the lognormal fitting is shown. A $D_{em}$-based mass PSD was also determined assuming a uniform apparent particle density of 1.65 g cm$^{-3}$ as the dotted black line (Pitz et al., 2008b). It should be noted that three modes were applied in this example; however, four modes might be used in the lognormal fitting for the volume or mass PSDs in the



Supplement. The size-resolved $\rho_{eff}$ function (A, B, or C) for the conversion is listed and the vertical dashed green lines
correspond to the different size ranges for the $\rho_{eff}$ functions (denoted in Fig. 9).

Figure 6 is representative of urban aerosol mass PSDs measured by gravimetric methods with inertial impactors that are
included in the Supplement. The black dotted curve in the mass PSD indicates the measured data. The solid black lines in the
volume and mass PSDs are $D_{em}$-based PSDs converted from the measured $D_a$-based mass PSD by using $\rho^B_{eff}$. The rainbow
color lines are the converted $D_{em}$-based volume and mass PSDs for a range of $\rho_{eff}$ to account for the uncertainties of $\rho_{eff}$ in the
coarse regime. The relationship between the color and $\rho_{eff}$ is shown in the color bar. The dotted color lines are the lognormal
fitting curves for each mode. The dashed black lines in the volume and mass PSDs are the sum of the fitted sub-modes. The
fitted volume PSD was converted back to number and surface area PSDs, which are shown as the dashed black lines. The
effective size range for the lognormal fitting is shown.

**5 Methodology for determining size-resolved urban aerosol respiratory tract deposited dose rates**

The compilation of urban aerosol number and mass PSDs from around the globe offers a basis to evaluate the implications of
geographical variations in the shape and magnitude of PSDs on human inhalation exposure. The respiratory tract deposited
dose rate (RTDDR), or inhaled deposited dose rate, is a valuable, yet underused, exposure metric that combines an aerosol
PSD with size-resolved respiratory tract deposition fractions to predict the number and mass of particles deposited in each
55 region of the human respiratory tract per unit time. Despite the utility of this metric in offering a useful link between air
pollution and human health outcomes, only a few studies have determined RTDDRs for outdoor or indoor aerosols (e.g.
Hussein et al., 2013, 2019; Löndahl et al., 2007, 2009; Wu et al., 2018). Here, the RTDDR is evaluated in the form of a
lognormal size distribution, $dRTDDR_{N/M}/dLogD_{em}$, to be consistent with the presentation of the urban aerosol PSDs. The
subscripts $N$ and $M$ denote the number and mass dose rate, respectively.

The $dRTDDR_N/dLogD_{em}$ (h$^{-1}$) and $dRTDDR_M/dLogD_{em}$ (µg h$^{-1}$) were calculated for an adult engaged in light physical
activity (e.g. walking) in the urban outdoor environment as follows:

$$dRTDDR_N/dLogD_{em} = dN/dLogD_{em} \times V_T \times f \times DF(D_p), \tag{13}$$

$$dRTDDR_M/dLogD_{em} = dM/dLogD_{em} \times V_T \times f \times DF(D_p), \tag{14}$$

where $dN/dLogD_{em}$ and $dM/dLogD_{em}$ are the urban aerosol number and mass PSDs, respectively; $V_T$ is the tidal volume
(mL); $f$ is the breathing frequency (min$^{-1}$); and $DF(D_p)$ is the size-resolved particle deposition fraction in the human
respiratory tract (–). The $V_T$ and $f$ were taken as the average for an adult female and male engaged in light activity: $V_T$ = 1083
mL and $f$ = 18 min$^{-1}$ (U.S. EPA, 2011). Total and regional (head airways, tracheobronchial region, and pulmonary region)
size-resolved particle deposition fractions for an adult were obtained using the age-specific symmetric single-path model from





·70    the open-source Multiple-Path Particle Dosimetry (MPPD) Model (v3.04, Applied Research Associates, Inc., Albuquerque, NM, USA) (Miller et al., 2016), as illustrated in Fig. 7. It is assumed that the body of the adult is upright, and the breathing route is nasal.

·75    Hygroscopic growth of the inhaled urban aerosol is not considered given the broad collection of PSDs analyzed in this study and the unknown chemical composition, mixing state, and hygroscopic growth factors associated with the aerosol populations. Hygroscopic growth of the inhaled aerosol can shift the size-resolved deposition fractions, and in turn, the estimated $dRTDDR_N/dLogD_{em}$ and $dRTDDR_M/dLogD_{em}$ (e.g. Löndahl et al., 2007). An inhaled dose study by Kodros et al. (2018) found that the deposited mass concentration of aerosols in urban areas are lower than those in rural areas due to the higher hydrophobic content of urban aerosols. As aerosols are transported from an urban to rural area, the hydrophobic content
·80    becomes hydrophilic under atmospheric ageing, and the $SO_2$ converts to particle-phase sulfate. Relatively hydrophilic particles can undergo hygroscopic growth under the high relative humidity conditions of the human respiratory tract. Hygroscopic growth of the inhaled urban aerosol may increase the RTDDR of accumulation mode particles as the growth shifts the mass PSD out of the minimum of the particle deposition fraction curve at approximately $D_{em}$ = 400 nm.

·85    The $dRTDDR_N/dLogD_{em}$ were only determined for urban aerosol number PSDs measured with electrical mobility-based techniques involving the UFP regime. The $dRTDDR_M/dLogD_{em}$ were determined for urban aerosol mass PSDs converted from number PSDs measured via one or more measurement techniques, including electrical mobility- and aerodynamic-/optical-based techniques, as well as mass PSDs measured directly via gravimetric methods with inertial impactors.

**6 Methodology for evaluating the urban aerosol PSD that penetrates through a building ventilation system filter**

·90    The compilation of urban aerosol number and mass PSDs from around the globe offers a basis to evaluate the implications of geographical variations in the shape and magnitude of PSDs on indoor air quality and aerosol filtration in buildings. One of the pathways by which urban aerosols can enter the indoor environment, where people spend approximately 90% of their time (Klepeis et al., 2001), is through the ventilation system of a commercial or residential building (Azimi et al., 2014). Heating, ventilation, and air conditioning (HVAC) filters are installed in building ventilation sytems to filter outdoor air and recirculated
·95    indoor air. Thus, HVAC filters serve as an important interface between the outdoor and indoor atmospheres.

    The relationship between the urban aerosol number PSD and a HVAC filter's size-resolved filtration efficiency determines the PSD of the urban aerosol that penetrates through the filter. The filter-transformed PSD is a useful indicator of the urban aerosol PSD to which occupants will be exposed to in buildings where the majority of outdoor air is provided via the ventilation system.
·00    Furthermore, the shape of the urban aerosol PSD that passes through the HVAC filter plays an important role in influencing the loading kinetics of the filter (Valmari et al., 2006). PSD-driven changes in the loading behavior can affect the evolution




of the filter's pressue drop over time and the associated HVAC blower energy consumption required to overcome this airflow resistance (He et al., 2016).

The urban aerosol PSD that penetrates through a HVAC filter installed in a single-pass building ventilation system was calculated as:

$$Penetrated\ (dN/dLogD_{em}) = dN/dLogD_{em} \times (1 - Filtration\ Efficiency), \qquad (15)$$

Penetrated urban aerosol PSDs were determined for filters with the Minimum Efficiency Reporting Value (MERV) rating of 8 and 14. Size-resolved filtration efficiencies for the MERV 8 and 14 filters were estimated from Hecker and Hofacre (2008), as illustrated in Fig. 8. The impact of aerosol hygroscopicity and electrostatic charge, filter ageing, filter bypass, and ventilation airflow parameters (e.g. face velocity) on changes in the size-resolved filtration efficiency were not considered. The transformation of the urban aerosol PSD due to penetration through the building envelope (e.g. infiltration) was not evaluated.

## 7 Summary of size-resolved urban aerosol effective densities

The size-resolved $\rho_{eff}$ functions for Groups A, B, and C are illustrated in Fig. 9. The blue lines indicate the mean $\rho_{eff}$ values derived from direct measurements of $\rho_{eff}$, such as through evaluation of the mass-mobility relationship of an aerosol population. The grey areas represent the maximum and minimum values of the directly measured $\rho_{eff}$ for each group. The associated measurement techniques for $\rho_{eff}$ are noted. The directly measured $\rho_{eff}$ in Groups A and B, both of which are in the 'urban' environment, present similar values and do not show much variation among the different studies conducted in China and the United States (Table 2). Furthermore, across the size range covered by direct $\rho_{eff}$ measurements in Groups A and B, there is no clear size-dependency of $\rho_{eff}$, in part because organics and secondary inorganic ions are dominant in this size range.

The directly measured $\rho_{eff}$ values collected from cities in the United States (Group B) are between 1.1 to 1.6 g cm$^{-3}$, while those measured in China (Group A) are slightly greater, with values between 1.3 to 1.9 g cm$^{-3}$, possibly due to a greater abundance of secondary inorganic species. The directly measured $\rho_{eff}$ in the 'traffic' environment (Group C) presents a decreasing trend with the increase in particle size for 50 nm $< D_{em} <$ 400 nm. This is largely due to primary emissions of soot particles from vehicle exhaust, which typically adopt a loose, chain-like agglomerated morphology with a fractal dimension (mass-mobility exponent) less than 3 (Barone et al., 2011; Pagels et al., 2009; Rawat et al., 2016; Rissler et al., 2013, 2014). Numerous studies have revealed the decrease of $\rho_{eff}$ with the increase in particle size for vehicle exhaust aerosol (Barone et al., 2011; Maricq et al., 2000; Olfert et al., 2007; Park et al., 2003; Rissler et al., 2013; Virtanen et al., 2006). For size ranges where direct measurement of $\rho_{eff}$ are unavailable, various assumptions were made to estimate a range of $\rho_{eff}$ values, as discussed in Sect. 3. The light blue, yellow, and light green areas in Fig. 9 represent the range of possible $\rho_{eff}$ values for the three size fractions, and the red lines represent the mean values. The light green area in the coarse regime shows estimated $\rho_{eff}$ values derived from a combination of different $\chi$ and $\rho_P$ (e.g. Fig. 2), which take a variety of particle morphologies and chemical





compositions into consideration. The wide variation in $\rho_{eff}$ for coarse mode particles has important implications for estimating urban aerosol mass PSDs from number PSDs measured via commonly deployed instruments (e.g. APS, OPS, OPC).

$\rho_{eff}$ is dependent on the chemical composition and morphological features ($\chi$ and $\rho_p$) of an urban aerosol population. Typically, direct $\rho_{eff}$ measurements are conducted between $D_{em}$ = 30 to 400 nm. Particles in this size range often consist of organics, secondary inorganic material, and black carbon. As discussed in Sect. 3.5, studies have found SOA to have $\rho_{eff}$ values between 1 to 1.65 g cm$^{-3}$ and secondary inorganic material, such as $H_2SO_4$, $(NH_4)_2SO_4$, and $NH_4NO_3$ to have $\rho_{eff}$ values between 1.7 and 1.83 g cm$^{-3}$ (Kostenidou et al., 2007; Lide, 2005; Malloy et al., 2009; Mikhailov et al., 2013; Nakao et al., 2013; Neusüß et al., 2002; Tang, 1996; Xiao et al., 2015; Zelenyuk et al., 2008). The $\rho_{eff}$ of soot particles can fall below 1 g cm$^{-3}$, with a decreasing trend with the increase in particle size. The relative fraction of various species in an urban air mass, such as organics, secondary inorganic materials, and loosely agglomerated soot particles, among others, will determine the size dependency of the $\rho_{eff}$ for the externally and internally mixed aerosol population. For example, previous direct $\rho_{eff}$ measurements conducted in Los Angeles, Copenhagen, and Beijing found $\rho_{eff}$ to be inversely proportional to particle size when the fraction of soot particles from vehicle emissions were relatively abundant due to elevated traffic intensity (Geller et al., 2006; Rissler et al., 2014; Qiao et al., 2018). Conversely, a study in Shanghai observed an increase in $\rho_{eff}$ with particle size (Table 2), which was attributed to an abundance of hygroscopic species, such as $(NH_4)_2SO_4$ and $NH_4NO_3$ (Ye et al., 2011; Yin et al., 2015).

Urban aerosol $\rho_{eff}$ is expected to be temporally variant at a given sampling location due to the transient nature of emission sources. For example, in the urban environment, diurnal patterns in traffic density can drive time-dependent shifts in $\rho_{eff}$ as the relative fraction of soot particles changes throughout the day. A low fraction of soot particles was observed in Copenhagen during the nighttime due to the low traffic density (Rissler et al., 2014). Two minima in $\rho_{eff}$ were found in Houston in the morning at 07:00 and in the evening at 19:00 to 20:00, likely due to increased emissions of soot particles during rush hours (Levy et al., 2013). In Beijing, one study found $\rho_{eff}$ to decrease during the nighttime due to an increase in the abundance of soot particles (Hu et al., 2012). This temporal shift in the urban aerosol $\rho_{eff}$ was found to be due to the emissions of heavy trucks, which are only allowed to enter the fifth ring road in Beijing during the night, as well as more intense coal burning for domestic heating in the night during the heating season.

Urban aerosol $\rho_{eff}$ is also influenced by air pollution events and air mass origins. Direct $\rho_{eff}$ measurements in Beijing observed higher $\rho_{eff}$ values during clean air quality episodes compared to polluted air quality episodes (Hu et al., 2012). This finding was attributed to the greater relative fraction of mineral dust during the clean episodes and abundance of organics and secondary inorganic ions in the particle-phase during the polluted episodes (Hu et al., 2012). The $\rho_{eff}$ can shift during atmospheric NPF events, depending on the dominant condensable vapor during the particle growth period. Qiao et al. (2018) found $\rho_{eff}$ to decrease during a NPF event in Beijing, indicating that the condensable vapors were dominated by organics, which corresponded to the increase in the fraction of organic material in sub-micron particle mass concentrations. In contrast, $\rho_{eff}$





measurements in Shanghai observed an increase in $\rho_{eff}$ during NPF events, suggesting that relatively heavier secondary inorganic materials were the primary driver for particle condensational growth (Xie et al., 2017; Yin et al., 2015). Wind direction can affect $\rho_{eff}$ by changing the air mass origin. Direct $\rho_{eff}$ measurements in a street canyon in central Copenhagen showed higher fractions of dense mode particles when the air mass traveled over more polluted regions than when the air mass come from clean sea/ocean areas (Rissler et al., 2014).

While the studies summarized in Fig. 9 and Table 2 provide valuable insights into variations of $\rho_{eff}$ with particle size, geographical location, and intra-city environments (urban vs. traffic), more measurements are clearly needed in many cities around the world to develop a more comprehensive understanding of the nature of urban aerosol morphology and the factors that drive changes in size-resolved $\rho_{eff}$. In particular, direct $\rho_{eff}$ measurements are needed in the accumulation and coarse modes given the variability identified in this study and the contribution of both modes to urban aerosol mass PSDs (Sect. 9). Doing so will provide a basis to better translate measured number PSDs to mass PSDs and $D_a$-based PSDs to $D_{em}$-based PSDs.

## 8 Urban aerosol PSD observations around the globe: an overview of existing data

Urban aerosol PSD observations made between 1998 and 2017 were collected and analyzed to evaluate geographical variations in the shape and magnitude of number and mass PSDs and to identify gaps in PSD measurements. In total, $n$=793 PSDs from $n$=125 cities in $n$=51 countries were collected. The measured PSDs were fit to the multi-modal lognormal distribution function and translated across number, surface area, volume, and mass domains following the methodologies outlined in Sect. 4. The PSD observations are summarized and categorized in the Supplement and are grouped by geographical region: AF, CSSA, EA, EU, LA, NAAN, and WA. Table 1 lists the number of PSDs in each city and country partitioned among the seven geographical regions. It is important to note that the sampling duration of the PSD observations presented in Table 1 are variable. Among all PSDs, 14.3% were long-term measurements (LT: > 6 months) and 33.3% were moderate-term measurements (MT: 1 – 6 months). The remaining PSDs represent observations made over periods less than one month through short-term field measurement campaigns.

Figure 1 illustrates the temporal and geographical distribution by year between 1998 and 2017 of the urban aerosol PSD references analyzed in this study. Between 1998 and 2007, there is a clear increase in the number of published studies reporting urban aerosol PSD observations. During this period, the majority of PSD measurements were conducted in cities in EU and NAAN. However, beginning in 2006 and continuing through 2017, a greater fraction of published PSD observations were collected in cities in EA and CSSA. Studies reporting PSD observations in NAAN cities appear to have declined between 2009 and 2017. Urban aerosol PSD observations in LA, WA, and AF remain sparse across the examined time period, however, the frequency of publications reporting PSD measurements in LA has been fairly stable between 2009 and 2017. Based upon the criteria employed in the literature search (Sect. 2), the year with the greatest number of published studies reporting urban



aerosol PSD measurements was 2009 (*n*=17), followed by 2014 (*n*=15). 1998 and 1999 were associated with the fewest number of published PSD measurements (*n*=1).

Figure 10 presents the global distribution of urban aerosol PSD measurement locations included in this study. It is apparent that there are regions were numerous observations have been made (e.g. EU) and others were measurements are scarce (e.g. WA, LA, AF). Among the *n*=793 urban aerosol PSD observations collected in this study, 39.8% of them are from EU, 15.2% are from North America, and 18.1% are from EA. Conversely, only 7.2% are from WA, 5.9% from LA, and 3.6% from AF. The three countries that contribute the most to the collection of urban aerosol PSD observations in this study are the United States (13.3%), China (12.2%), and Germany (8.8%).

The majority of PSD observations in EU have been collected in Germany (*n*=68), Finland (*n*=52), the United Kingdom (*n*=37), Italy (*n*=31), and Denmark (*n*=26). The top five EU cities with the greatest number of urban aerosol PSD observations include Helsinki, Finland (*n*=52), Leipzig, Germany (*n*=27), Copenhagen, Denmark (*n*=25), London, United Kingdom (*n*=16), and Milan, Italy (*n*=15). The *n*=101 PSD observations compiled from the United States represent *n*=16 cities, including Los Angeles, California (*n*=34), Riverside, California (*n*=8), Pittsburgh, Pennsylvania (*n*=7), and Buffalo, New York (*n*=6). A growing number of PSD measurements have been made in China (*n*=93), in cities such as Beijing (*n*=38), Guangzhou (*n*=14), and Shanghai (*n*=12). In India, *n*=48 PSD observations were analyzed from Kanpur (*n*=23) and New Delhi (*n*=13), among other cities.

A paucity of urban aerosol PSD measurements is clear throughout the entirety of AF, LA, and WA; CSSA excluding India; Canada, although a few measurements have been conducted in the Greater Toronto and Hamilton Area (total of *n*=15); Russia (*n*=1 from Tiksi); Australia (*n*=10 across Launceston, Brisbane, and Wollongong); and New Zealand (*n*=8 from Auckland). The few published PSD measurements in AF, CSSA excluding India, and WA were primarily reported as columnar volume PSDs by using sun/sky radiometers. Urban aerosol PSD observations have only been made in *n*=8 countries in AF, including Egypt (*n*=12), South Africa (*n*=6), Zambia (*n*=4), Botswana (*n*=3), Kenya (*n*=1), Cape Verde (*n*=1), Mali (*n*=1), and Senegal (*n*=1). Similarly, in LA, PSD measurements have been made in a few countries, the majority of which have been reported in Brazil (*n*=38, with *n*=35 from São Paulo), Chile (*n*=6), Mexico (*n*=3), and Cuba (*n*=1). *n*=58 urban aerosol PSD measurements have been reported in WA, many of which were made in Zanjan, Iran (*n*=23), followed by Istanbul, Turkey (*n*=13), Fahaheel, Kuwait (*n*=12), and Yanbu, Saudi Arabia (*n*=9). 68.6% of the PSD observations in CSSA have been reported in India, with comparatively less measurements coming from other countries in the region, including Pakistan (*n*=8), Singapore (*n*=6), Nepal (*n*=4), Thailand (*n*=2), and Vietnam (*n*=2).

The vast majority of urban aerosol PSD measurements analyzed in this study were made via electrical mobility-based techniques, as summarized in Table S1. Comparatively less direct measurements of mass PSDs were made via gravimetric





methods employing inertial impactors (Table S4). In total, 76.8% of the urban aerosol PSDs reported number PSDs down to the UFP fraction of the sub-micron regime. However, only six of the urban aerosol number PSDs involving the UFP regime are from Southeast Asia; only thirteen are from WA, and none are from AF. The lack of urban aerosol PSD measurements down to the UFP regime in many regions of the world makes it very challenging to accurately estimate urban aerosol inhalation exposures. This is of concern given the inhalation toxicity and adverse health effects associated with UFPs (Delfino et al., 2005; Li et al., 2016, 2017a; Oberdörster et al., 2004, 2005; Pietropaoli et al., 2004; Rychlik et al., 2019; Sioutas et al., 2005).

## 9 Urban aerosol PSDs: from number to mass

### 9.1 Geographical variations in the magnitude and shape of sub-micron urban aerosol number PSDs

Geographical variations in sub-micron urban aerosol number PSD measurements ($dN/dLogD_{em}$, cm$^{-3}$) are presented in Fig. 11. Each log-log plot incorporates number PSDs measured with electrical mobility-based techniques that cover the sub-micron regime ($n$=624, Sect. 4.1.1, Table S1). Each line represents an individual PSD observation compiled in the Supplement and the color indicates the occurrence frequency of the number PSDs at a given particle size ($D_{em}$) with a certain particle number concentration. Red, orange, and yellow curves indicate the number PSDs where the occurrence frequency is high among the analyzed studies. All number PSD observations are included in the 'Global' plot (top-left). Number PSDs for EA, CSSA, EU, LA, and NAAN are presented in the remaining plots; however, AF and WA are not included due to the lack of PSD measurements in the sub-micron regime in the two regions. The solid black lines indicate the median number PSDs for each group, which are also presented in Fig. 12 on a linear y-axis scale. It can be seen that among the geographical regions, the greatest amount of sub-micron number PSDs have been reported for cities in EU, NAAN, and EA; comparatively less have been reported in CSSA and LA.

The visualization of the global distribution in sub-micron urban aerosol number PSDs (Fig. 11, top-left) demonstrates that there exist significant variations in both the magnitude and shape of number PSDs measured across urban environments around the world. For a given particle size ($D_{em}$), there can exist over two orders of magnitude variation in the particle number concentration. This variation in the amplitude of the number PSDs is persistent across the considered size range, from $D_{em}$ = 3 to 1000 nm, which includes the nucleation, Aitken, and accumulation modes. The red-yellow region of the global plot surrounds the median number PSD (black line). Wide variability in the magnitude of the number PSDs above and below the median PSD is apparent. Thus, defining a globally representative urban aerosol number PSD is challenging given the vast array of factors that can influence the shape of a PSD at a particular sampling location within in a city. However, the red region suggests that on a global-basis, some trends do exist in regard to the shape and magnitude of urban aerosol number PSDs. Notably, number PSDs are often dominated by particles between $D_{em}$ = 10 to 100 nm, with varying contributions from the sub-10 nm fraction and accumulation mode, depending on the conditions that exist at the measurement site. Across this





65    size fraction, there is a high occurrence frequency of number PSDs with an amplitude between 1000 to 10000 cm$^{-3}$. In some cases, the amplitude can reach or exceed 50000 cm$^{-3}$, most commonly in the nucleation and Aitken modes. The global median number PSD demonstrates that number PSDs often drop off in magnitude by nearly a factor of a hundred across the width of the accumulation mode, from approximately 1000 cm$^{-3}$ at $D_{em} = 100$ nm to 10 cm$^{-3}$ as $D_{em}$ approaches 1000 nm.

70    The geographically-resolved collections of urban aerosol number PSDs presented in Fig. 11 and 12 indicate that there exists inter-region variability in the shape and magnitude of number PSDs. Number PSDs in NAAN and EU present similar structural characteristics; similarly, number PSDs in EA and CSSA are alike in both shape and magnitude. Number PSDs measured in cities in NAAN and EU tend to skew to the left and are often dominated by nucleation and Aitken mode particles, whereas number PSDs measured in cities in EA and CSSA tend to skew to the right and are often dominated by Aitken and accumulation mode particles. The magnitude of the number PSDs in the accumulation mode in EA and CSSA tends to be higher than that in NAAN and EU. This is apparent in the collection of individual PSDs in each of the geographical regions in Fig. 11, as well as in the median number PSDs presented in Fig. 12. Conversely, the magnitude of the number PSDs in the sub-50 nm fraction of the UFP regime in EA and CSSA tends to be lower than those measured in NAAN and EU. This is especially true for the nucleation mode, which is often much more pronounced in the urban atmospheres of NAAN and EU cities. The median number PSD for LA more closely resembles number PSDs measured in NAAN and EU as compared to those in EA and CSSA. However, the lack of PSD observations in LA makes it difficult to draw conclusions about the shape of PSDs in this region. The $D_{em}$ associated with the prominent peak for each of the median number PSDs presented in Fig. 12 are: $D_{em} \sim 20$ nm for EU, $D_{em} \sim 30$ nm for NAAN, $D_{em} \sim 35$ nm for LA, and $D_{em} \sim 60$ to 100 nm for EA and CSSA.

85    The variation in the magnitude of the number PSDs for EA, CSSA, EU, LA, and NAAN is generally consistent with that observed in the global distribution of PSDs presented in Fig. 11. The abundance of number PSD observations in EU provides a basis to more reliably identify a representative PSD for the region. The red-yellow-light green band for EU demonstrates that a large fraction of PSD measurements in EU cities tend to cluster around the median number PSD. Between $D_{em} = 10$ to 100 nm, the amplitude of this PSD band varies between 1000 and 10000 cm$^{-3}$. Less frequently, number PSDs with magnitudes exceeding 10000 cm$^{-3}$, or as low as 100 cm$^{-3}$, have been reported in EU cities. A faint band of moderate occurrence frequency can be observed in both EA and NAAN, however, the comparatively few PSD observations in CSSA and LA make it difficult to identify such trends in these two regions.

To better visualize differences in the shape of the urban aerosol number PSDs and to probe the relative fraction of particles in different modes, each number PSD was normalized by its maximum concentration such that variations in the magnitude of the number PSDs can be neglected (Fig. 13). The normalized urban aerosol number PSDs presented in Fig. 13 are grouped by country and geographical region (from top to bottom): WA, NAAN, LA, EU, EA, and CSSA. Many of the normalized number PSDs in EA and CSSA tend to show a peak (red-orange color) at around $D_{em} = 100$ nm and few show peaks at or near the





nucleation mode. Some of the normalized number PSDs measured in China and India present prominent peaks in the

'00 accumulation mode, between $D_{em}$ = 100 to 200 nm. However, it can be seen that particles greater than $D_{em}$ = 300 nm contribute negligibly to normalized number PSDs in EA and CSSA. Normalized number PSDs in NAAN and EU generally exhibit peaks at smaller particle sizes ($D_{em}$ = 10 to 50 nm), while a few observations made in Germany, Italy, and the United States present peaks near $D_{em}$ = 100 nm. The normalized number PSDs measured in LA, predominately in São Paulo, Brazil, closely resemble observations reported in NAAN and EU. The prominent nucleation mode in the WA normalized number PSDs is in part due

'05 to the few PSD observations collected from the region, which were made at a 'traffic' site in Fahaheel, Kuwait.

There are clear distinctions between urban aerosol number PSDs measured in NAAN/EU and EA/CSSA. Fig. 14 presents the relationship between total particle number concentration, integrated over the measured size range of a PSD measurement, and the count median diameter (CMD) for each of the sub-micron number PSDs presented in Fig. 11. Number PSDs in EA and

'10 CSSA tend to cluster to the right, from CMD = 50 to 100 nm, whereas number PSDs in NAAN and EU tend to cluster toward the left, from CMD = 20 to 60 nm. There is, however, outliers in each region, such as number PSDs in EA with prominent nucleation modes and CMDs of approximately 10 nm, and number PSDs in NAAN with prominent accumulation modes and CMDs approaching 100 nm. There are only a few number PSDs in CSSA that exhibit CMDs below 50 nm. In all geographical regions, there exists nearly two orders of magnitude variation in total particle number concentrations, which are often bounded

'15 by 1000 cm$^{-3}$ at the lower end and 100000 cm$^{-3}$ at the upper end. Number PSDs in EU, NAAN, and EA that have CMDs < 20 nm are associated with total particle number concentrations exceeding 10000 cm$^{-3}$. A clear inverse relationship between total particle number concentration and CMD does not appear to exist. Interestingly, numerous number PSDs in EA and CSSA with CMDs of approximately 100 nm have concentrations > 10000 cm$^{-3}$. The wide variation in the total particle number concentrations presented in Fig. 14 is consistent with the trends reported in a review of geographical variations in total particle

'20 number concentrations across forty urban roadside measurement sites around the world (Kumar et al., 2014).

It should be noted that many factors can influence the magnitude and shape of urban aerosol number PSDs, beyond geographical region, which is the focus of the global-scale analysis presented in Fig. 11-14. Country-wide PSD measurement campaigns have identified significant variations in number PSDs among different cities within the same country (Peng et al.,

'25 2014; Tuch et al., 2003) and at different measurement sites within the same city (Birmili et al., 2013; Costabile et al., 2009; Hussein et al., 2005; Ketzel et al., 2004; Tuch et al., 2006; Wehner et al., 2002). Regarding the latter, several studies conducted in EU cities have shown that total particle number concentrations can vary as high as a factor of roughly nine within the same city (Birmili et al., 2013; Buonanno et al., 2011; Mejía et al., 2008; Mishra et al., 2012). Localized spatial variations in urban aerosol PSDs and number concentrations are due in part to the nature of local emission sources near the measurement site and

'30 meteorological conditions, including wind speed and direction, temperature, and relative humidity (Baxla et al., 2009; Birmili et al., 2001; Charron and Harrison, 2003; Kaul et al., 2011; Nieto et al., 1994; Rose et al., 2010; Stanier et al., 2004; Swietlicki et al., 2008; Väkevä et al., 2000; Wehner and Wiedensohler, 2003; Weingartner et al., 1997; Yu et al., 2018). Physiochemical





processes that can transform an aerosol population over space and time are also very important, such as particle growth due to coagulation and condensation, particle shrinkage due to evaporation, reactive uptake, and wet and dry deposition, among others (Gaston et al., 2014; Limbeck et al., 2003; Lin et al., 2011; Moise and Rudich, 2002; Salma et al., 2011; Shi and Harrison, 1999; Tang et al., 2010; Zhu et al., 2002a, 2002b).

**9.2 Geographical variations in the magnitude and shape of urban aerosol mass PSDs**

Global variations in urban aerosol mass PSD measurements ($dM/dLogD_{em}$, $\mu$g m$^{-3}$) are presented in Fig. 15. The log-log plot incorporates mass PSDs measured by gravimetric methods with inertial impactors and measurements made with electrical mobility- and aerodynamic-/optical-based techniques that cover both the sub-micron and coarse modes ($n$=122, Sect. 4.1.2-4.1.3, Tables S2-S4). As discussed in Sect. 4, the size-resolved $\rho_{eff}$ functions for Groups A, B, and C (Fig. 9) were used in converting $D_a$-based PSDs to $D_{em}$-based PSDs and translating measured number PSDs to mass PSDs. Similar to Fig. 11, each line represents an individual PSD observation compiled in the Supplement and the color indicates the occurrence frequency of the mass PSDs at a given particle size ($D_{em}$) with a certain particle mass concentration. The solid black line indicates the median mass PSD among the global compilation of observations. In comparing Fig. 11 and 15, it is evident that sub-micron urban aerosol number PSDs are more commonly reported in the literature compared to mass PSDs or number PSDs spanning the sub-micron and coarse regimes.

The visualization of the global distribution in urban aerosol mass PSDs in Fig. 15 demonstrates that there exist significant variations in both the magnitude and shape of mass PSDs measured across urban environments around the world. While the limited amount of mass PSD observations makes it difficult to discern clear trends in the structure of mass PSDs, some trends are evident. Notably, urban aerosol mass PSDs are dominated by particles with $D_{em} > 100$ nm and are typically bi-modal, exhibiting peaks in both the accumulation and coarse modes, as indicated by the median mass PSD. The relative contribution of the two modes is variable among the PSD observations. In some cases, urban aerosol mass PSDs are dominated by accumulation mode particles, while other PSDs present a prominent coarse mode. Within the accumulation mode, the amplitude of the mass PSDs spans two orders of magnitude, from 1 $\mu$g m$^{-3}$ to 100 $\mu$g m$^{-3}$. The $D_{em}$ associated with the prominent peak in the accumulation mode is variable. The spread in the magnitude of the mass PSD in the coarse mode is consistent with that observed in the accumulation mode. Some mass PSDs exhibit amplitudes that exceed 100 $\mu$g m$^{-3}$, however, their occurrence frequency is very low. The magnitude of mass PSDs in the UFP regime is relatively insignificant and ranges from 0.01 to 1 $\mu$g m$^{-3}$. Unlike for the number PSDs in Fig. 11, a band of high occurrence frequency is not evident in Fig. 15. Some degree of clustering of mass PSDs around the median PSD is evident, however, there is clearly more variation in the structure of mass PSDs as compared to number PSDs. This may be due to the variety of measurement techniques employed and uncertainties in translating number PSDs to mass PSDs using the size-resolved $\rho_{eff}$ functions for Groups A, B, and C.



'65    As with the sub-micron urban aerosol number PSDs, the mass PSDs were normalized by their maximum concentrations such that variations in the magnitude of the mass PSDs can be neglected (Fig. 16). The normalized urban aerosol mass PSDs presented in Fig. 16 are grouped by country and geographical region (from top to bottom): WA, NAAN, LA, EU, EA, CSSA, and AF. The normalized mass PSDs demonstrate that a significant fraction of particle mass exists below $D_{em} = 1000$ nm in numerous cities in NAAN, EU, EA, and CSSA. For measurements that included the UFP regime, it is clear that sub-100 nm

'70    particles contribute little to urban aerosol mass PSDs. The majority of the normalized mass PSDs in NAAN and EU show a peak in the accumulation mode (red-orange color) between $D_{em} = 200$ to 600 nm, while some show peaks in both the accumulation and coarse modes. Most of the normalized mass PSDs in EA (predominately from China) are bi-modal with accumulation mode peaks that span $D_{em} = 300$ to 1000 nm and coarse mode peaks that span $D_{em} = 3000$ to 8000 nm. A few mass PSDs in EA (measured in Korea) are uni-modal with a prominent coarse mode that extend beyond $D_{em} = 10000$ nm. The

'75    normalized mass PSDs in CSSA are more variable in shape, with varying contributions from both modes. The $D_{em} = 100$ to 200 nm fraction of the accumulation mode in both EA and CSSA, which contributed meaningfully to number PSDs in the two regions, represents a minor component of sub-micron aerosol mass.

         The shape of normalized urban aerosol mass PSDs in WA and AF are uniquely different from the other geographical regions.

'80    In WA, the normalized mass PSDs are clearly dominated by coarse mode particles. Measurements made in Istanbul, Turkey show a prominent peak between $D_{em} = 6000$ to 10000 nm, with some displaying a second coarse mode peak between $D_{em} = 1000$ to 2000 nm. Normalized mass PSDs from Yanbu, Saudi Arabia show a strong peak near $D_{em} = 10000$ nm, with either a very weak or non-existent peak in the accumulation mode. The prominent coarse modes in WA cities are likely due to frequent dust events and enhanced dust resuspension in WA cities and the relatively large size of mineral dust particles (Al-Mahmodi,

'85    2011). The few PSD observations from AF display a dominant coarse mode, with peaks spanning $D_{em} = 1000$ to 5000 nm.

**9.3 Intra-city variations in urban aerosol number PSDs between urban background and traffic-influenced sites**

         Urban aerosol PSDs can exhibit intra-city spatial variations depending on the measurement location and its proximity to local emission sources, such as traffic. The urban aerosol PSD observations were categorized by intra-city sampling location, as documented in the Supplement. This provides a basis to compare the shape of PSDs collected at urban background (UB) and

'90    traffic-influenced (TR) sites from cities across the globe. Figure 17 presents normalized sub-micron urban aerosol number PSDs divided into UB (top) and TR (bottom) sites. UB represents urban areas that are far from direct emission sources and are not meaningfully affected by local traffic emissions, while TR indicates an environment that is strongly influenced by traffic emissions, such as an urban street canyon or roadside (Birmili et al., 2013). The aerosol populations measured at UB sites are typically transported from other urban microenvironments, undergoing various transformation and ageing processes

'95    during transport. The PSDs at UB and TR sites are grouped by country in Fig. 17. Only PSD observations with a measurement period greater than one week are presented. The majority of the measurements presented in Fig. 17 are from NAAN and EU cities, with only a few from EA, WA, and LA.





Normalized number PSDs measured at UB sites often show prominent peaks at larger particle sizes compared to those
measured at TR sites. UB measurements are typically dominated by Aitken mode particles, with peaks ranging from $D_{em} = 20$
to 90 nm. In contrast, many of the TR measurements exhibit prominent nucleation modes with peaks falling below $D_{em} = 30$
nm, and in some cases, below $D_{em} = 10$ nm. For both UB and TR, there exists variability in the shape of the PSDs among the
cities and countries. In some cases, the structure of the normalized number PSDs are similar at UB and TR sites as some TR
sites exhibit prominent Aitken mode. The larger particles observed at UB sites are due in part to various aerosol transformation
processes, such as particle growth due to coagulation and the uptake of condensable organic and inorganic vapors during short-
and long-range transport (Fine et al., 2004; Wehner et al., 2002). A few of the normalized number PSDs at UB sites show
meaningful contributions from the accumulation mode, between $D_{em} = 100$ to 200 nm; such particles tend to persist in the
urban atmosphere for longer periods of time due to their lower rates of coagulation compared to nucleation and Aitken mode
particles and lower deposition rates compared to coarse mode particles (Hinds, 2012; Seinfeld and Pandis, 2012).

Urban aerosol number PSD observations made at TR sites are strongly influenced by traffic emissions. Vehicle emissions are
a major source of UFPs in the urban atmospheric environment (Kumar et al., 2014; Morawska et al., 2008a; Pant and Harrison,
2013). Several studies have reviewed the characteristics of aerosol emissions from traffic, including urban SOA formation
associated with vehicle exhaust (Gentner et al., 2017; Kittelson et al., 2006; Morawska et al., 2008a; Pant and Harrison, 2013;
Thorpe and Harrison, 2008). Traffic emissions can be broadly classified as exhaust- and non-exhaust-related. Exhaust-related
vehicle emissions include soot particles from incomplete combustion and particles formed via the nucleation and condensation
of $H_2SO_4$ and hydrocarbons as the hot exhaust is cooled and diluted in the ambient atmosphere (Dallmann et al., 2014; Kleeman
et al., 2000; Meyer and Ristovski, 2007; Morawska et al., 2008a; Shi et al., 2001; Shi and Harrison, 1999; Wehner et al., 2002).

Vehicle exhaust aerosol PSDs are influenced by many factors, including vehicle type (Gupta et al., 2010; Harris and Maricq,
2001; Liang et al., 2013), vehicle/engine operational mode (Giechaskiel et al., 2005; Li et al., 2013; Shi and Harrison, 1999),
fuel type (Agarwal et al., 2013; Armas et al., 2012; Jones et al., 2012; Kittelson et al., 2002), and use of aftertreatment
technologies (Giechaskiel et al., 2010; Mayer et al., 2002). As illustrated in Fig. 18, which presents normalized number PSDs
for selected urban aerosol sources, vehicle exhaust PSDs are typically dominated by UFPs. Freshly nucleated particles in
vehicle exhaust are relatively small, with $D_{em} < 30$ nm. Near-road measurements at TR sites commonly present peaks in the
sub-30 nm size fraction (Fig. 17), and some TR sites can be dominated by sub-10 nm particles (Pedata et al., 2015; Rönkkö et
al., 2017). This indicates that the freshly nucleated particles can contribute significantly to number PSDs at TR sites (Buonanno
et al., 2009b; Fushimi et al., 2008; Ketzel et al., 2003; Shi et al., 1999; Zhu et al., 2002a). However, with the increase of
distance from the road, either horizontally or vertically, these particles can grow by coagulation and condensation during
transport (Agus et al., 2007; Hitchins et al., 2000; Li et al., 2007; Zhu et al., 2002a, 2002b), while some can shrink due to
evaporation (Dall'Osto et al., 2011b; Ning et al., 2010; Zhang et al., 2004). Non-exhaust-related traffic emissions include



brake wear, road-tire interactions, and road dust resuspension; the former is an important source of sub-micron particles. Brake wear aerosol PSDs are influenced by the material and operational temperature of the brake pad (Grigoratos and Martini, 2015; Kukutschová et al., 2011; Nosko et al., 2017; Timmers and Achten, 2016; Wahlström et al., 2010). As shown in Fig. 18, normalized sub-micron number PSDs of brake wear aerosol can span from the nucleation mode to the accumulation mode.

**9.4 Sub-micron urban aerosol number PSDs in Asia: factors contributing to the prominent accumulation mode**

The results presented in Fig. 11-14 indicate that urban aerosol number PSDs in EA and CSSA are more commonly associated with a significant fraction of accumulation mode particles and CMDs of approximately 100 nm compared to those reported in NAAN and EU. This indicates that sub-micron urban aerosol populations in EA and CSSA, and particularly in China and India, are relatively larger in size than those reported in other geographical regions. A multitude of factors are responsible for governing the shape of number PSDs in urban environments in EA and CSSA. The pronounced accumulation mode can be driven by the direct emissions of accumulation mode particles in both the urban area, as well as regional transport of such particles from rural and industrialized areas. Biomass burning is an important emission source in a number of countries in EA and CSSA. The PSDs of biomass burning aerosol depend on a variety of factors, including: the type of biomass, the condition of the flame, and atmospheric ageing processes (Janhäll et al., 2010; Reid and Hobbs, 1998; Rissler et al., 2006; Sakamoto et al., 2016; Zhang et al., 2011). As shown in Fig. 18, the burning of grass, corn straw, and rice straw produces normalized number PSDs with a significant fraction of accumulation mode particles and CMDs of approximately 100 nm (Janhäll et al., 2010; Reid et al., 2005; Sakamoto et al., 2016). It has been observed that residential biomass burning, possibly for cooking and heating, can contribute to high particle number concentrations of accumulation mode particles in the evening in New Delhi, India (Mönkkönen et al., 2005). A recent study using the GEOS-Chem-TOMAS model identified significant aerosol emissions from biomass burning in India and Indonesia from residential, agricultural, and wildfire sources (Kodros et al., 2018). Direct burning has been reported to be a common technique to eliminate agricultural residuals in both China and India (Bi et al., 2019). The contribution of biomass burning to urban aerosols was confirmed by the high content of water-soluble potassium in the particle-phase (Qi et al., 2015; Zheng et al., 2005). In addition to biomass burning, other urban sources may directly emit accumulation mode particles, such as vehicle exhaust, power plants, and industrial activities (Vu et al., 2015).

Another factor contributing to the abundance of accumulation mode particles in EA and CSSA are ageing processes that can grow nucleation and Aitken mode particles through coagulation and condensation of organic or inorganic vapors (Moffet et al., 2008; Yang et al., 2012). Back trajectories indicate that aerosols transported from industrialized regions south and west of Beijing, China can grow into larger sizes by condensation within slowly moving air masses, thereby contributing to the pronounced accumulation mode in urban areas (Wu et al., 2008). The abundance of condensable organic and inorganic vapors (e.g. $NO_x$, $SO_2$, and VOCs) in polluted areas can aid particle growth to larger sizes. It has been reported that the concentrations of condensable vapors are higher in urban areas in China and India compared to those in NAAN and EU due to heavier air pollution in the former (Gao et al., 2009; Huang et al., 2014; Kodros et al., 2018; Kulmala et al., 2005; Misra et al., 2014;



565  Mönkkönen et al., 2005; Shen et al., 2016a, 2016b; Wang et al., 2013; Wehner et al., 2004b).  High levels of gas-phase precursors in China can result in significant SOA formation, which can contribute to severe haze events that are often dominated by accumulation mode particles (Huang et al., 2014).  The elevated number concentrations of accumulation mode particles in EA and CSSA have a substantial surface area that can serve as a coagulation sink for nucleation and Aitken mode particles.  This suppression of UFPs can cause the number PSDs in EA and CSSA to further skew to larger particle sizes.

## 10 Urban aerosol PSDs: implications of human inhalation exposure assessment

### 10.1 Geographical variations in size-resolved urban aerosol number respiratory tract deposited dose rates

Geographical variations in size-resolved urban aerosol number respiratory tract deposited dose rates (RTDDRs), expressed in the form of lognormal size distributions ($dRTDDR_N/dLogD_{em}$, h$^{-1}$), are presented in Fig. 19.  Urban aerosol PSD measurements are a valuable tool for human inhalation exposure assessment given the strong size dependency of particle deposition in the respiratory system.  RTDDRs provide a basis to understand how variations in the shape and magnitude of urban aerosol number PSDs affect the rate at which particles deposit in each region of the human respiratory tract (Hussein et al., 2013, 2019; Löndahl et al., 2007, 2009; Wu et al., 2018).  As described in Sect. 5, number RTDDRs were estimated by integrating the urban aerosol number PSDs complied for each geographical region (Fig. 11) with size-resolved particle deposition fractions in the human respiratory tract (Fig. 7) and selected breathing parameters.

The urban aerosol number RTDDRs are presented in the form of a grid, with each row corresponding to a geographical region (CSSA, EA, EU, NAAN, and LA) and each column corresponding to a region (or summation across regions as the total) of the human respiratory tract (total, head airways, tracheobronchial, and pulmonary).  In a given number RTDDR log-log plot, each line represents a RTDDR (as $dRTDDR_N/dLogD_{em}$, used interchangeably herein) estimated from a sub-micron urban aerosol number PSD presented in Fig. 11 and the color indicates the occurrence frequency of the number RTDDRs at a given particle size ($D_{em}$) with a certain RTDDR value.  The solid black lines indicate the median number RTDDR for each geographical and respiratory tract region pair, which are illustrated in Fig. 20 on a linear y-axis scale.  It should be noted that the RTDDRs presented in Fig. 19 are estimates derived for an adult engaged in light physical activity (e.g. walking) in the urban outdoor environment.  As noted in Sect. 5, hygroscopic growth of the inhaled urban aerosol is not considered given the broad collection of PSDs analyzed in this study and the unknown chemical composition and hygroscopic growth factors associated with the aerosol populations.  Thus, care should be used in interpreting the number RTDDRs presented herein.  However, valuable insight can still be gleaned by the vast collection of urban aerosol number PSDs, and associated RTDDRs, analyzed in this study.





95   There exists significant geographical variation in number RTDDRs for each respiratory tract region, as shown in Fig. 19 and 20. Similar to the urban aerosol number PSDs, there can exist over two orders of magnitude variation in the number RTDDR for a given particle size ($D_{em}$). The amplitude of the total $dRTDDR_N/dLogD_{em}$ (left column) is typically between $10^8$ and $10^{10}$ h$^{-1}$ in the UFP regime, with values exceeding $10^{11}$ h$^{-1}$ in the nucleation mode and as low as $10^6$ h$^{-1}$ in the accumulation mode. Despite the variability in the magnitude, a clear red-orange-yellow band is present around the median number RTDDRs

00   in EA, EU, and NAAN, suggesting a general trend exists in each geographical region. Total number RTDDRs are dominated by sub-micron particles across all geographical regions. The pronounced differences observed between NAAN/EU and EA/CSSA urban aerosol number PSDs in Fig. 11 and 12 are evident in the total number RTDDRs. Similar to the median number PSDs presented in Fig. 12, the median total number RTDDRs in NAAN/EU (and LA) and EA/CSSA are similar in both shape and magnitude. In NAAN, EU, and LA cities, nucleation and Aitken mode particles contribute significantly to

05   total number RTDDRs, whereas in EA and CSSA cities, total number RTDDRs receive significant contributions from both Aitken and accumulation mode particles (Fig. 19 and 20). A prominent mode in the median total $dRTDDR_N/dLogD_{em}$ in NAAN/EU/LA is evident between $D_{em}$ = 20 to 30 nm, with a partial second mode occurring at $D_{em}$ < 10 nm for NAAN/EU. EA/CSSA exhibit a prominent mode between $D_{em}$ = 50 to 70 nm, with CSSA presenting a second mode beyond $D_{em}$ = 100 nm. The magnitude of the prominent modes of the median total $dRTDDR_N/dLogD_{em}$ in all geographical regions are on the order

of $10^9$ h$^{-1}$, and vary between 3 and 6 $x$ $10^9$ h$^{-1}$.

High number RTDDRs occur for size fractions where prominent modes of the urban aerosol number PSDs coincide with high deposition fractions. The deposition fraction curve for the pulmonary region exhibits a maximum at approximately $D_{em}$ = 30 nm (Fig. 7). In the UFP regime, the deposition fraction curve for the tracheobronchial region increases with decreasing particle

size from $D_{em}$ = 100 nm to approximately $D_{em}$ = 2 nm, where it obtains its maximum. Similarly, the head airways deposition curve increases with decreasing particle size from $D_{em}$ = 70 to 1 nm in the UFP regime. As a result, the sub-micron total deposition fraction curve increases with decreasing particle size from a minimum at approximately $D_{em}$ = 400 nm to nearly unity at $D_{em}$ = 1 nm (Fig. 7). Thus, particles in the UFP regime more efficiently deposit in the human respiratory tract compared to particles in the accumulation mode.

NAAN/EU number PSDs are often dominated by nucleation and Aitken mode particles, with CMDs between 20 and 60 nm, whereas EA/CSSA number PSDs are often dominated by Aitken and accumulation mode particles, with CMDs between 50 and 100 nm (Fig. 13 and 14). The strong contribution of the nucleation mode in NAAN/EU number PSDs, as compared to EA/CSSA cities, results in comparatively higher number RTDDRs for $D_{em}$ < 30 nm in both the tracheobronchial region and

head airways (Fig. 19 and 20). For NAAN/EU, the median $dRTDDR_N/dLogD_{em}$ for the tracheobronchial region and head airways reaches a peak at $D_{em}$ = 10 nm (Fig. 20), with some individual PSDs exhibiting a peak at $D_{em}$ < 10 nm (Fig. 19). In the pulmonary region, the prominent peak of the median $dRTDDR_N/dLogD_{em}$ for NAAN/EU is present near $D_{em}$ = 20 nm, which is the approximate location of the prominent mode of the median number PSDs in NAAN/EU and the maximum of the





pulmonary deposition fraction curve. Median number RTDDRs in the sub-30 nm fraction are greater in NAAN/EU compared to EA/CSSA for each of the three respiratory tract regions. Conversely, greater number RTDDRs are observed for particles with $D_{em}$ > 50 nm for all respiratory tract regions in EA/CSSA as compared to NAAN/EU. The magnitude of the median $RTDDR_N/dLogD_{em}$ for particles with $D_{em}$ > 50 nm are about a factor of two greater in EA/CSSA compared to NAAN/EU/LA. In both the tracheobronchial and pulmonary regions, the median number RTDDRs for EA/CSSA exhibit a prominent mode at approximately $D_{em}$ = 50 nm, while in the head airways, a peak is present near $D_{em}$ = 100 nm.

For a given geographical region, variations in the dose rate are observed among the three respiratory tract regions. As can be seen in both Fig. 19 and 20, the magnitude of the $dRTDDR_N/dLogD_{em}$ is greatest for the pulmonary region across all geographical regions, followed by the tracheobronchial region and head airways. For all geographical regions, the median $dRTDDR_N/dLogD_{em}$ in the pulmonary region is nearly one order of magnitude higher than that in the head airways, and two to three times higher than in the tracheobronchial region. This is due to the significant contribution of Aitken mode particles to urban aerosol number PSDs in most cities and the high deposition fractions observed in the Aitken mode for the pulmonary region. Thus, the pulmonary region experiences the greatest dose burden in regard to the number of particles that deposit per unit time.

### 10.2 Geographical variations in size-resolved urban aerosol mass respiratory tract deposited dose rates

Geographical variations in size-resolved urban aerosol mass RTDDRs, expressed in the form of lognormal size distributions ($dRTDDR_M/dLogD_{em}$, μg h$^{-1}$), are presented in Fig. 21. Mass RTDDRs were estimated for CSSA, EA, EU, NAAN, LA, and WA. In a given mass RTDDR log-log plot, each line represents a RTDDR (as $dRTDDR_M/dLogD_{em}$, used interchangeably herein) estimated from urban aerosol mass PSDs presented in Fig. 15 and the color indicates the occurrence frequency of the mass RTDDRs at a given particle size ($D_{em}$) with a certain RTDDR value. The solid black lines indicate the median mass RTDDR for each geographical and respiratory tract region pair, which are illustrated in Fig. 22 on a linear y-axis scale.

Similar to the urban aerosol number RTDDRs, there can exist over two orders of magnitude variation in the mass RTDDR for a given particle size ($D_{em}$). The amplitude of the total $dRTDDR_M/dLogD_{em}$ (left column) is typically between 10$^{-1}$ and 10$^{1}$μg h$^{-1}$ in the accumulation and coarse modes, with values as low as 10$^{-2}$ μg h$^{-1}$ in the UFP regime. Despite the variability in the magnitude, a clear red-orange-yellow band is present around the median mass RTDDRs in EA, EU, and NAAN, suggesting a general trend exists in each geographical region. In contrast to the number RTDDRs, most mass RTDDRs are bimodal and dominated by accumulation and coarse mode particles across all geographical and respiratory tract regions (Fig. 21 and 22). However, the relative contribution of the two modes to the mass RTDDRs varies among the urban aerosol mass PSD observations. As illustrated in Fig. 19 and 21, UFPs tend to dominate number RTDDRs in most urban environments, but contribute negligibly to mass RTDDRs. The magnitude of $dRTDDR_M/dLogD_{em}$ is greatest for cities in CSSA, followed by





EA, for all respiratory tract regions. This is consistent with the findings of Kodros et al. (2018), which reported high aerosol mass deposited concentrations in the human respiratory tract for India and Eastern China. Cities in NAAN and EU are associated with $dRTDDR_M/dLogD_{em}$ lower in magnitude compared to the other geographical regions. LA and WA lie in between mass RTDDRs for EA/CSSA and NAAN/EU. The magnitude of $dRTDDR_M/dLogD_{em}$ are fairly consistent across all respiratory tract regions.

Similar to the number RTDDRs, high mass RTDDRs occur for size fractions where prominent modes of the urban aerosol mass PSDs coincide with high deposition fractions. From $D_{em}$ = 100 to 10000 nm, the head airways deposition curve increases with increasing particle size, with local maxima observed for the pulmonary region at approximately $D_{em}$ = 3000 nm and for the tracheobronchial region at around $D_{em}$ = 7000 nm (Fig. 7). As a result, the total deposition fraction curve increases with increasing particle size from a minimum at approximately $D_{em}$ = 400 nm to nearly unity at $D_{em}$ = 10000 nm (Fig. 7). Thus, coarse mode particles more efficiently deposit in the human respiratory tract compared to particles in the accumulation mode.

Coarse mode particles tend dominate the total, tracheobronchial, and head airways $dRTDDR_M/dLogD_{em}$ for each geographical region (Fig. 21 and 22). This is due to high deposition fractions coinciding with coarse modes that exceed, or are nearly equal to, the accumulation mode in most cities (Fig. 16). The dominance of the coarse mode to the mass RTDDRs is especially strong in both the head airways and the tracheobronchial regions. In CSSA and EA, the prominent mode of the mass RTDDR in the coarse mode is located at approximately $D_{em}$ = 6000 nm for the head airways and the tracheobronchial regions, while in the pulmonary region it is located near $D_{em}$ = 3000 nm. For both the tracheobronchial and pulmonary regions, this peak occurs near the local maximum in the deposition curves for both regions (Fig. 7). Mass RTDDRs in NAAN and EU are similar in shape to those in CSSA and EA in the tracheobronchial region, however, in the head airways, a partial second mode is observed at around $D_{em}$ = 10000 nm. Mass RTDDRs in LA and WA exhibit a prominent coarse mode for the head airways and tracheobronchial region. However, the full extent of the mass RTDDRs for LA and WA in the coarse mode cannot be reliably characterized due to the lack of mass PSD observations in the two regions. As shown in Kodros et al. (2018), desert regions of WA and AF are associated with very high deposited mass concentrations in the human respiratory tract.

In contrast to the head airways and the tracheobronchial regions, mass RTDDRs in the pulmonary region receive near equal contributions from the accumulation and coarse modes in CSSA, EU, and NAAN. The location of the prominent mode of the mass RTDDRs in the accumulation mode varies among the geographical regions, but is typically between $D_{em}$ = 200 to 500 nm. As illustrated in Fig. 7, the highest deposition fractions in the accumulation mode are found in the pulmonary region. Thus, cities with mass PSDs with prominent accumulation modes can be associated with high mass dose rates in this size fraction for the deepest region of the human respiratory tract. Across all geographical regions, the pulmonary region contributes significantly to the total mass RTDDRs from $D_{em}$ = 100 to 1000 nm.



## 11 Urban aerosol PSDs: implications for indoor air quality & aerosol filtration in building ventilation systems

The compilation of urban aerosol PSDs from around the globe offers a basis to evaluate the implications of geographical variations in the shape and magnitude of PSDs on indoor air quality and aerosol filtration in buildings. Urban aerosols can be transported into the indoor environment though mechanical and natural ventilation or through infiltration via cracks and gaps in the building envelope. HVAC filters in mechanical ventilation systems of commercial or residential buildings can influence the transport of urban aerosols into the indoor atmosphere. Thus, understanding how HVAC filters modulate an urban aerosol PSD can provide insights on evaluation of human inhalation exposure to aerosols of outdoor origin.

Geographical variations in urban aerosol number PSDs that penetrate through a HVAC filter installed in a single-pass building ventilation system are presented in Fig. 23. The penetrated urban aerosol number PSDs are presented in the form of a grid, with each row corresponding to a geographical region (CSSA, EA, EU, NAAN, and LA) and the two columns corresponding to MERV 8 and 14 filters. The solid black lines indicate the median penetrated urban aerosol number PSD for each geographical region and MERV filter, which are illustrated in Fig. 24 on a linear y-axis scale. Similar to the urban aerosol number PSDs presented in Fig. 11, there are significant geographical variations in both the shape and magnitude of the filter-transformed PSDs. The sub-micron penetrated PSDs are unimodal for all geographical regions and for both filters. The magnitude of the penetrated PSDs are greater in CSSA and EA as compared to NAAN, EU, and LA for both the MERV 8 and 14 filters. However, the nucleation mode of the penetrated PSDs for a MERV 8 filter in NAAN/EU are slightly greater than those for EA/CSSA. The PSDs of the penetrated urban aerosol in NAAN, EU, and LA present a peak at approximately $D_{em} = $ 30 to 50 nm for the MERV 8 filter, while those in CSSA and EA show a peak between $D_{em} = $ 70 to 120 nm. The magnitude of the penetrated PSDs is lower for the MERV 14 filter given its higher filtration efficiency compared to the MERV 8 filter (Fig. 8). For NAAN, EU, and LA, the penetrated PSDs shift to the right when transitioning from a MERV 8 to 14 filter, while this shift is not as pronounced for CSSA and EA. The latter is due in part to the overlap between the minimum filtration efficiency at $D_{em} = $ 100 to 200 nm (Fig. 8) and the strong accumulation mode of the number PSDs in CSSA and EA (Fig. 12).

To further characterize the impact of geographical variations in the shape of urban aerosol number PSDs on indoor air quality and aerosol filtration in buildings, the relative fraction of particles that penetrate or deposit onto the filter were determined (Fig. 25). The total penetration and deposition fractions were determined as the ratio of the total number of particles that penetrate through the filter or deposit onto the filter to the total number of incoming particles (urban aerosol concentration). In doing so, this negates the impact of variations in the magnitude of the urban aerosol PSDs on filtration. The total penetration and deposition fractions were calculated for each urban aerosol number PSD in each geographical region (Fig. 11), with the mean and standard deviation for each geographical region shown in Fig. 25. Notably, a significant decrease of approximately 40% in the total penetration fraction is apparent for all geographical regions when switching from a MERV 8 to 14 filter.



Differences in the total penetration fractions among the five geographical regions with the same MERV-rated filter suggests that the shape of the urban aerosol number PSDs influences the overall filter performance in different regions. For both filters, the total penetration fractions are higher in EA/CSSA as compared to NAAN/EU by 5 to 9%. This is due to the prominent

accumulation modes of the number PSDs in EA/CSSA, which coincide with the particle size range ($D_{em}$ = 100 to 200 nm) where the filtration efficiency is at a minimum since the particles are too large to deposit by diffusion and too small to deposit by interception and inertial impaction (Fig. 8). Thus, the results presented in Fig. 25 suggests that the same filter may perform less effectively in EA/CSSA as compared to NAAN/EU due to variations in the shape of the number PSDs between the regions.

It is important to note that the analysis of the urban aerosol number PSDs that penetrate through HVAC filters is intended to serve as an illustrative example of how geographical variations in the shape and magnitude of urban PSDs can influence indoor air quality in mechanically ventilated buildings. The modeling approach simply integrates urban aerosol PSDs and fixed HVAC filtration efficiency curves; thus, it may not represent actual filtration processes occurring in real HVAC installations. The actual filtration efficiency can vary from filter to filter, even among those with the same MERV rating. In addition, the

filtration efficiency of a filter is not fixed during its service life and will evolve over time with the formation of solid dendrites and liquid films within the filter fiber matrix (Pei et al., 2019). A multitude of factors can affect the actual penetrated urban aerosol PSD downstream of a filter, including: the type of filter (e.g. electret filter) (Tang et al., 2018), relative humidity (Pei et al., 2019), the filter installation method (which influences the bypass of particles around the filter), face velocity (Leung et al., 2010), aerosol morphological features (Wang, 2012), and the mechanical ventilation recirculation ratio.

Beyond filtration of the urban aerosol as it enters the indoor environment, other physiochemical transformations can occur due to heating and cooling processes in mechanical ventilation systems. Heating of outdoor air can cause the volatile component of the urban aerosol population to evaporate from the particles, as well as to reduce the aerosol liquid water content and possibly transfer water-soluble species into the gas phase (Donahue et al., 2006; Huffman et al., 2009; Johnson et al., 2017).

Such evaporation processes can shift the urban aerosol PSDs to smaller sizes. Sudden elevations in relative humidity due to humidification in mechanical ventilation systems may induce hygroscopic growth of the urban aerosol as it enters the indoor environment, as well as increase the aerosol liquid water content, which could enhance the uptake of water-soluble species indoors.

## 12 Framing future research directions for urban aerosol PSD observations at a global-scale

This critical review has provided a comprehensive overview of urban aerosol number and mass PSD observations made in cities around the globe. Critical gaps in urban aerosol PSD observations were identified in many geographical regions and countries, with a severe lack of ground-based PSD data for cities in AF, LA, WA, and parts of CSSA (Fig. 1 and 10). Available PSD measurement data is often short in duration, with only 14.3% of the analyzed observations extending beyond 6 months.


Similarly, there have been few direct measurements of size-resolved urban aerosol effective densities, and existing data is
limited for many size fractions (Fig. 9, Table 2). A greater number of direct measurements of urban aerosol effective densities
will enable for accurate translation of urban aerosol number PSDs to mass PSDs in a given urban environment.

There exist significant geographical variations in the shape and magnitude of urban aerosol PSDs due to differences in primary
and secondary aerosol sources and meteorological conditions (Fig. 11 and 15). Such differences have important implications
for human exposure and health as they drive large changes in the rate at which particles deposit in each region of the human
respiratory tract (Fig. 19 and 21). The important contribution of sub-200 nm particles to urban aerosol number PSDs in all
regions reinforces the need for routine monitoring of the smallest particles in the urban atmosphere. Urban aerosol PSD
observations that span the UFP to coarse regimes are especially lacking, with only 14% of the analyzed PSDs measuring
particles across this wide size range. Coordinated global efforts are needed to build a continuous, long-term, wide size range,
and ground-based urban PSD observation network in cities across the world. Such a network is necessary for improving our
ability to link urban air pollution with human health and toxicological outcomes, understanding the atmospheric
transformations of urban aerosol populations, and supporting air quality legislation and policy decisions that address particles
both big and small (Kulmala, 2018).

Existing ground-based air quality monitoring stations are largely focused on measurements of size-integrated $PM_{2.5}$ mass
concentrations. Expansive observational datasets of $PM_{2.5}$ mass concentrations are now available. This has significantly
advanced knowledge of the impact of $PM_{2.5}$ on urban air pollution and human health in the past two decades (Apte et al., 2015;
Cheng et al., 2016; Van Donkelaar et al., 2010; Gelencsér et al., 2007; de Jesus et al., 2019; Thunis et al., 2017; West et al.,
2016). $PM_{2.5}$ measurement gaps do exist, with 141 of 243 countries lacking ground-based $PM_{2.5}$ monitoring stations (Martin
et al., 2019). The ubiquity of low-cost aerosol sensors (e.g. OPCs) are providing a foundation for large-scale deployment of
$PM_{2.5}$ monitoring networks (Motlagh et al., 2020). However, given the nature of urban aerosol number and mass PSDs, as
illustrated in Fig. 11 and 15, observations of $PM_{2.5}$ mass concentrations are insufficient to accurately characterize an urban
aerosol population. Of particular importance is the measurement of PSDs that include the UFP regime, given their significant
contribution to particle number concentrations (Fig. 11, 12, and 13). This is especially important given that UFP number
concentrations and $PM_{2.5}$ mass concentrations are not representative of each other, as particles that contribute to the two size-
integrated metrics often originate from different sources (de Jesus et al., 2019).

The compilation of urban aerosol PSD observations in this review demonstrates the need for a transition from size-integrated
$PM_{2.5}$ mass concentration measurement to broader size range PSD measurements that include the nucleation, Aitken,
accumulation, and coarse modes. CMDs of urban aerosol number PSDs often fall between $D_{em}$ = 10 to 100 nm (Fig. 14); such
particles contribute negligibly to urban aerosol mass PSDs (Fig. 15 and 16). Many urban aerosol sources, such as biomass
burning, traffic emissions (exhaust and non-exhaust), industrial and domestic combustion, cooking, and atmospheric new



particle formation events, produce particles in the UFP regime (Fig. 18) (Brines et al., 2015; Kumar et al., 2014; Venecek et al., 2019; Vu et al., 2015). Urban aerosol PSDs provide more detailed information of emission sources than do size-integrated concentrations. Several PSD-based models have been developed using characteristic emission profiles of different sources to identify and apportion the emission sources (Beddows et al., 2009, 2014; Charron et al., 2008; Dall'Osto et al., 2011a; Friend et al., 2012; Gu et al., 2011; Harrison et al., 2011b; Kasumba et al., 2009; Kim et al., 2004; Ogulei et al., 2007; Thimmaiah et al., 2009; Tunved et al., 2004; Yue et al., 2013). A detailed review of such models was given by Vu et al. (2015).

PM$_{2.5}$ measurements are motivated in part by existing air quality legislation and exposure guideline values, whereas there are no existing nationwide regulations based on UFP or total particle number concentrations (Kumar et al., 2014). However, emission standards in the European Union now regulate particle number emissions for diesel passenger cars and light commercial vehicles in Euro 5, and for both gasoline and diesel passenger cars, light commercial vehicles, and heavy-duty diesel engines in Euro 6 (European Parliament and the Council of the European Union, 2007). A transition from PM$_{2.5}$ measurement to routine urban aerosol PSD monitoring down the UFP regime can help support new legislation based upon UFP or total particle number concentrations (Morawska et al., 2008b).

Future urban aerosol PSD measurements should span the entire UFP regime, including sub-3 nm nanocluster aerosol. The nanocluster aerosol mode is especially prominent during atmospheric new particle formation events and has been shown to dominate number PSDs measured at both urban background and traffic sites (Kangasniemi et al., 2019; Kontkanen et al., 2017; Olin et al., 2020; Rönkkö et al., 2017). Number concentrations of nanocluster aerosol can exceed 10000 cm$^{-3}$ in polluted megacities, such as Shanghai and Nanjing, China (Kontkanen et al., 2017); while roadside measurements have reported concentrations in excess of 100000 cm$^{-3}$ (Rönkkö et al., 2017). Achieving continuous urban aerosol number PSD observations from 1 to 10000 nm at the global-scale remains a challenge given the cost of sensitive aerosol instrumentation required for the detection of UFPs down to 1 nm and the collection of different measurement techniques needed to detect particles across such a wide size range. While advancements in low-cost optical particle sensing for detection of aerosols down to approximately $D_{em}$ = 300 to 500 nm have been made in recent years, efforts are still needed to develop low-cost condensation particle counters, differential mobility analyzers, and diffusion chargers for measurement of PSDs down to the UFP regime.

A future urban aerosol PSD observation network will improve our ability to more fully understand the health implications of urban aerosols. Measurement of PSDs incorporating the UFP regime are needed given the importance of UFPs on human health and the size-dependency of deposition in the human respiratory tract (Fig. 7). The health effects of UFPs are increasingly receiving more attention due to their high number concentrations in the urban environment, high surface area to mass ratio, and higher oxidative stress compared to larger particles (Allen et al., 2017; Burnett et al., 2018; Delfino et al., 2005; Li et al., 2016, 2017a; Oberdörster et al., 2004, 2005; Pieters et al., 2015; Pietropaoli et al., 2004; Rychlik et al., 2019). Human exposure to UFPs has been associated with the development of cardiopulmonary and cardiovascular diseases, lung cancer, and





asthma (Anderson et al., 2012; Delfino et al., 2005; Li et al., 2016; Meldrum et al., 2017; Oberdörster et al., 2005; Pietropaoli et al., 2004; Rychlik et al., 2019; Tsiouri et al., 2015; Valavanidis et al., 2008). As illustrated in Fig. 19 and 20, inhaled deposited dose rates (RTDDRs) on a number-basis are dominated by UFPs in nearly all geographical and respiratory tract

regions. The deposition fraction in the pulmonary region, which is often assumed to be more relevant for respiratory diseases, shows a maximum at approximately $D_{em}$ = 30 nm (Fig. 7), which overlaps with the prominent modes of many urban aerosol number PSDs in NAAN, EU, and LA (Fig. 11 and 12). UFPs can penetrate deep into the lung, can be transported into the bloodstream, and can translocate to different organs (Li et al., 2017b; Oberdörster et al., 2004).



*Supplement.* The Supplement includes a summary of the urban aerosol PSD database. Measurement information and lognormal fitting parameters for each PSD are summarized in Tables S1-S5 (pg. 2-25). Individual PSD figures present the measured and fitted PSDs, translated across number, surface area, volume, and mass domains (pg. 26-854). References used in compiling the urban aerosol PSD database are also included (pg. 855-865).

*Author contributions.* BEB conceived, planned, and secured funding for the study. TW conducted the literature search and performed the data collection and analysis under the supervision of BEB. TW and BEB wrote the manuscript.

*Competing interests.* The authors declare that they have no conflict of interest.

*Acknowledgements.* This work was supported by the American Society of Heating, Refrigeration and Air Conditioning Engineers (ASHRAE RP-1734). The authors are grateful for the support of the ASHRAE RP-1734 Project Monitoring Subcommittee: Paolo Tronville, Geoff Crosby, Bruce McDonald, Tom Justice, and Brian Krafthefer and the assistance of undergraduate student researchers Geordi Jose and Jihang Liu.

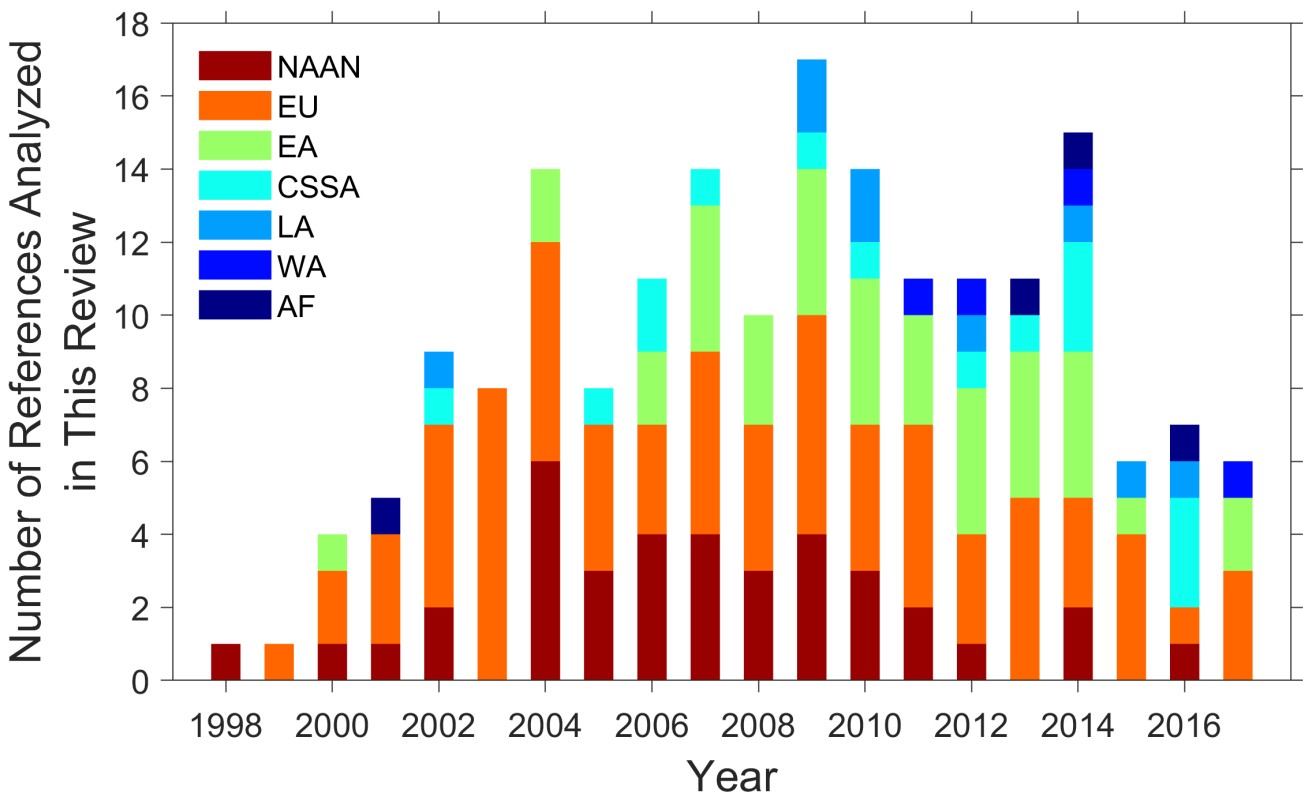

**Figure 1**. Temporal and geographical distribution by year of the urban PSD references analyzed in this study (1998-2017).





**Figure 2.** Effective densities ($\rho_{eff}$) as derived from different values of dynamic shape factors ($\chi$) and particle densities ($\rho_p$), assuming the value of $\frac{C_C(D_{ve})}{C_C(D_{em})}$ is approximately unity for coarse particles.





**Figure 3.** Example of urban aerosol number, surface area, volume, and mass PSDs measured by electrical mobility techniques. The upper left plot shows an urban aerosol number PSD with lognormal fitting parameters listed. The black curve indicates the measured data and the blue curve was reproduced with the multi-modal lognormal fitting parameters. The dashed curves represent each individual mode of the lognormal fitting. PN, PS, PV, and PM represent the size-integrated particle number, surface area, volume, and mass concentrations, respectively (the subscripts indicate the size range).







**Figure 4**. The ratio of aerodynamic diameter ($D_a$) to electrical mobility diameter ($D_{em}$) derived from different values of the dynamic shape factor ($\chi$) and particle density ($\rho_p$).



**Figure 5.** An example of urban aerosol number, surface area, volume, and mass PSDs measured by aerodynamic techniques. The black dotted curve in the number PSD indicates the measured data. The dotted line in the $D_a$-based volume PSD was converted from the measured $D_a$-based number PSD. The solid black lines in the volume and mass PSDs are $D_{em}$-based PSDs converted from the measured $D_a$-based number PSD. The rainbow color lines are the converted $D_{em}$-based volume and mass PSDs for a range of $\rho_{eff}$ to account for the uncertainties in $\rho_{eff}$ in the coarse regime. The relationship between the color and $\rho_{eff}$ is shown in the color bar. The dotted color lines are the lognormal fitting curves for each mode. The dashed black lines in the volume and mass PSDs are the sum of the fitted sub-modes. The fitted volume PSD was converted back to number and surface area PSDs, which are shown as the dashed black lines. The effective size range for the lognormal fitting is indicated. The mass $D_{em}$-based PSD was also converted assuming a uniform apparent $\rho_{eff}$ of 1.65 g cm$^{-3}$ as the dotted black line (Pitz et al., 2008a).





**Figure 6.** An example of urban aerosol number, surface area, volume, and mass PSDs measured by gravimetric methods employing inertial impactors. The black dotted curve in the mass PSD indicates the measured data. The solid black lines in the volume and mass PSDs are $D_{em}$-based PSDs converted from the measured $D_a$-based mass PSD by using $\rho^B_{eff}$. The rainbow color lines are the converted $D_{em}$-based volume and mass PSDs for a range of $\rho_{eff}$ to account for the uncertainties in $\rho_{eff}$ in the coarse regime. The relationship between the color and $\rho_{eff}$ is shown in the color bar. The dotted color lines are the lognormal fitting curves for each mode. The dashed black lines in the volume and mass PSDs are the sum of the fitted sub-modes. The fitted volume PSD was converted to number and surface area PSDs, which are shown as the dashed black lines. The effective size range for the lognormal fitting is indicated.





**Figure 7.** Size-resolved particle deposition fractions in the human respiratory tract, estimated by using the symmetric single-path model from the open-source Multiple-Path Particle Dosimetry (MPPD) Model (v3.04, Applied Research Associates, Inc., Albuquerque, NM, USA) (Miller et al., 2016) for an adult with an upright upper body.

85



**Figure 8.** Size-resolved filtration efficiency curves for MERV 8 and MERV 14 filters for estimating the number PSDs of the penetrated urban aerosol (Hecker and Hofacre, 2008).









**Figure 9.** Size-resolved urban aerosol effective density functions ($\rho_{eff}$) for Group A ('urban'; obtained from measurements in China), Group B ('urban'; obtained from measurements in the United States), and Group C ('traffic'; obtained from measurements in the United States, Finland, and Denmark). Details of the $\rho_{eff}$ measurements are summarized in Table 2. $\rho_{eff}$ values for different combinations of $\chi$ and $\rho_p$ are illustrated in Fig. 2. Measurement technique nomenclature: DMA: Differential Mobility Analyzer, APM: Aerosol Particle Mass Analyzer, SMPS: Scanning Mobility Particle Sizer, APS: Aerodynamic Particle Sizer, MOUDI: Micro-Orifice Uniform Deposit Impactor.





**Figure 10.** Global distribution of urban aerosol PSD measurement locations included in this study.







**Figure 11.** Urban aerosol number PSDs analyzed in this study, grouped by geographical region. The figure incorporates all sub-micron number PSDs measured by electrical mobility-based techniques (*n*=624). The color represents the occurrence frequency of the number PSDs at a given particle size with a certain concentration. The black lines indicate the median number PSDs in each group.

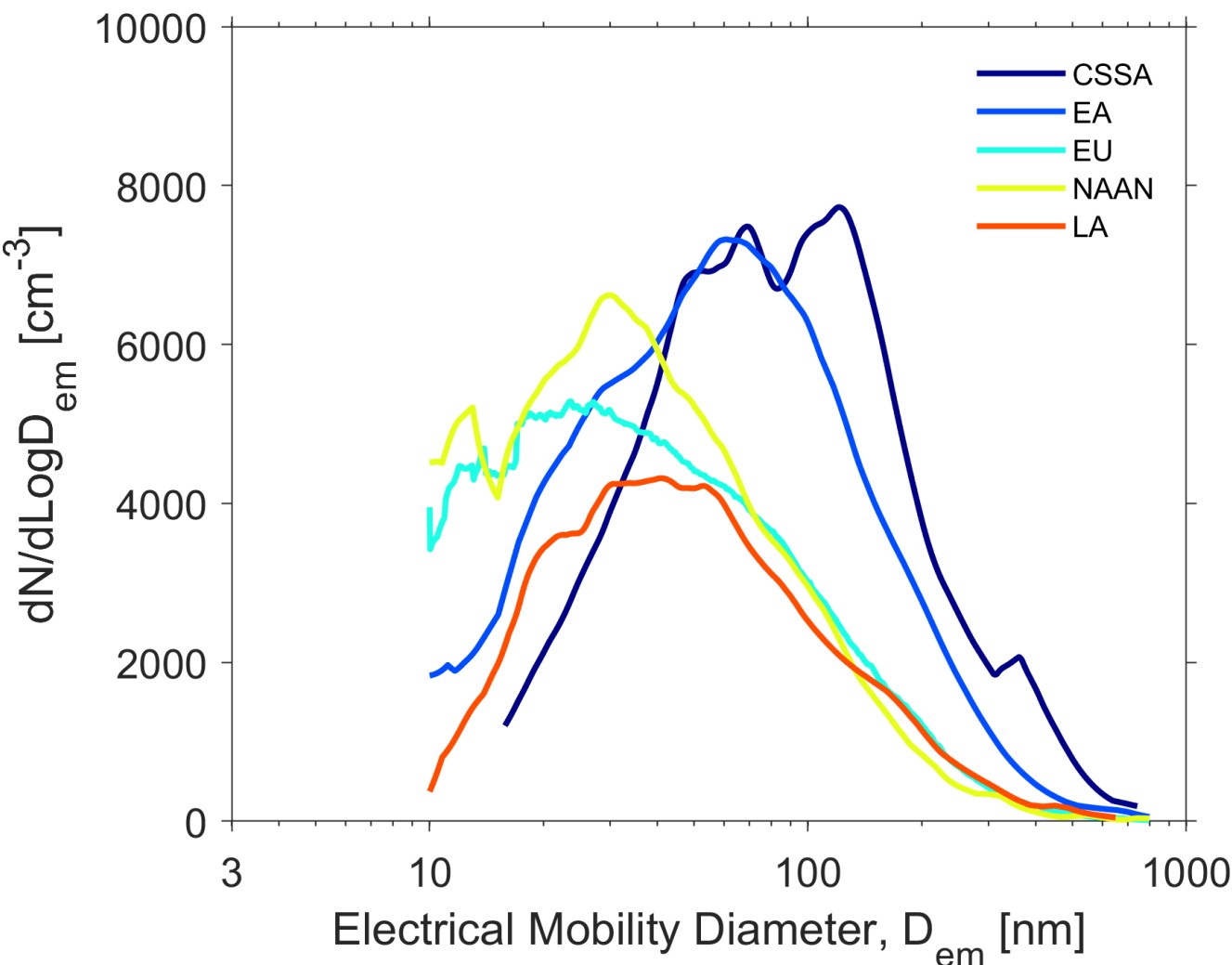

**Figure 12.** Median number PSDs for each geographical region.





10    **Figure 13.** Normalized urban aerosol number PSDs analyzed in this study from around the globe. The country codes are listed on the left and the region codes are listed on the right.

**Figure 14.** Relationship between the total particle number concentration, integrated over the measured size range, and the count median diameter (CMD), determined for each sub-micron number PSD measured by electrical mobility-based techniques (*n*=624) and grouped by geographical region.







**Figure 15.** Urban aerosol mass PSDs analyzed in this study from around the globe (*n*=122). The figure incorporates mass PSDs measured by gravimetric methods with inertial impactors and measurements made with electrical mobility-based and aerodynamic-/optical-based techniques that cover both the sub-micron and coarse modes. The color represents the occurrence frequency of the mass PSDs at a given particle size with a certain concentration. The black line indicates the median mass PSD.





**Figure 16.** Normalized urban aerosol mass PSDs analyzed in this study from around the globe. The country codes are listed on the left and the region codes are listed on the right.





**Figure 17.** Comparison between normalized urban aerosol number PSDs measured at urban background (UB) and traffic-influenced (TR) sites. Only the number PSDs with a measurement period greater than one week are presented. The country codes are listed on the left and the site type is listed on the right.





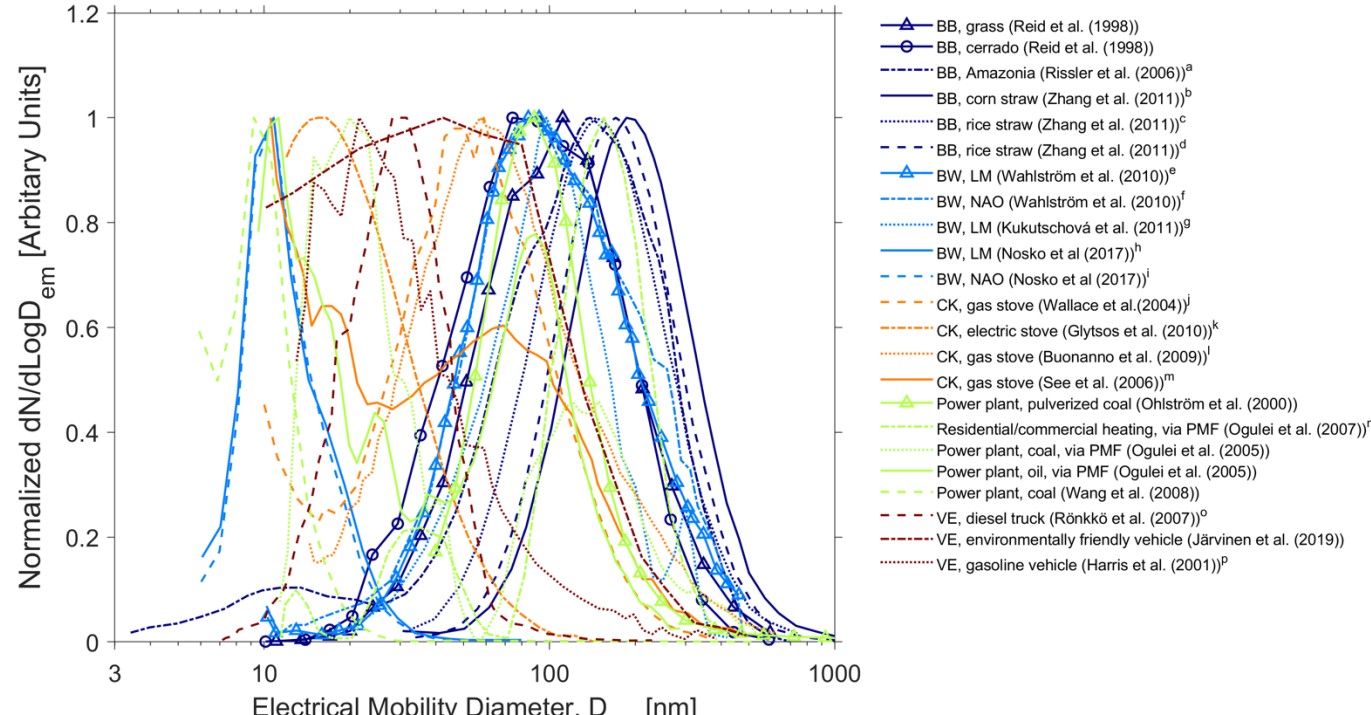

[a] Dry period in Fig. 4a in Rissler et al. (2006).
[bcd] PSD after ageing of 30 min in Fig. 6 in Zhang et al. (2011).
[e] LM1 in Fig. 7 in Wahlström et al. (2010). Brake pad material: low metallic (LM).
[f] NAO1 in Fig. 7 in Wahlström et al. (2010). Brake pad material: non-asbestos organic (NAO).
[g] t0+30 min in Fig. 6a in Kukutschová et al. (2011). Brake pad material: LM.
[h] LM 175 °C in Fig. 7 in Nosko et al. (2017). Brake pad material: LM.
[i] NAO 175 °C in Fig. 7 in Nosko et al. (2017). Brake pad material: NAO.
[j] DMA-CPC measurement in Fig. 4 in Wallace et al. (2004).
[k] Frying onion, t=7 min in Fig. 13 in Glytsos et al. (2010).
[l] Grilling bacon with maximum power in Fig. 6a in Buonanno et al. (2009a).
[m] Boiling in Fig. 2a in See and Balasubramanian (2006).
[n] Residential/commercial heating in winter in Fig. 6 in Ogulei et al. (2007b).
[o] Test condition 8 without thermodenuder in Fig. 2a in Rönkkö et al. (2007).
[p] Gasoline engine #2 at 96 km h$^{-1}$ in Fig. 4 in Harris and Maricq (2001).

**Figure 18.** Normalized number PSDs of selected urban aerosol sources, including biomass burning (BB), brake wear (BW),
cooking (CK), coal and oil burning for energy and heating, and vehicle exhaust (VE). Each number PSD is normalized by its
maximum concentration.





**Figure 19.** The total and regional $dRTDDR_N/dLogD_{em}$ in the respiratory tract in different geographical regions determined using the urban number PSDs analyzed in this study. The color represents the occurrence frequency of the $dRTDDR_N/dLogD_{em}$ at a given particle size with a certain dose rate. The black lines indicate the median $dRTDDR_N/dLogD_{em}$ in each group.







**Figure 20.** Median $dRTDDR_N/dLogD_{em}$ in each geographical region (black lines in Fig. 19).





**Figure 21.** The total and regional $dRTDDR_M/dLogD_{em}$ in the respiratory tract in different geographical regions determined using the urban mass PSDs analyzed in this study. The color represents the occurrence frequency of the $dRTDDR_M/dLogD_{em}$ at a given particle size with a certain dose rate. The black lines indicate the median $dRTDDR_M/dLogD_{em}$ in each group.





'55

**Figure 22.** Median $dRTDDR_M/dLogD_{em}$ in each geographical region (black lines in Fig. 21).





**Figure 23**. Urban aerosol number PSDs that penetrate through a MERV 8 and 14 filter in a building ventilation system for each geographical region, estimated using the number PSDs collected in this study (Fig. 11) and assuming single-pass MERV 8 and MERV 14 filters. The color represents the occurrence frequency of the penetrated number PSDs at a given particle size with a certain concentration. The black lines indicate the median penetrated number PSDs in each group.





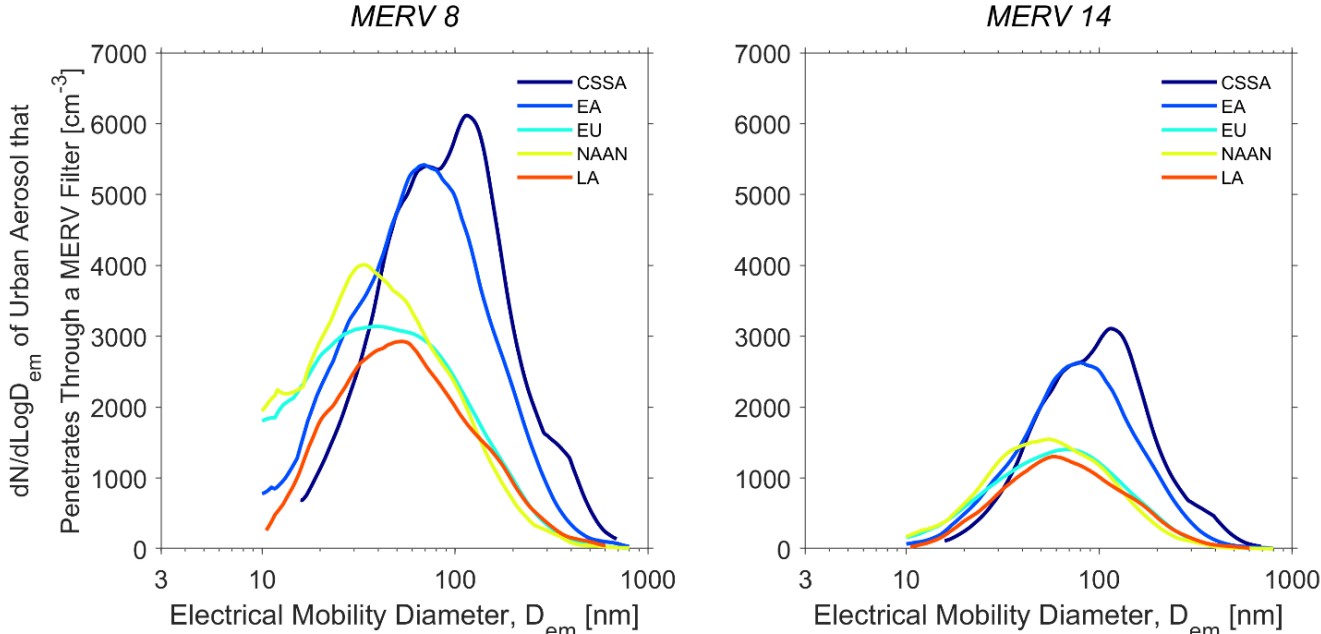

**Figure 24.** Median urban aerosol number PSDs that penetrate through a MERV 8 (left) and a MERV 14 (right) filter in a
building ventilation system for each geographical region (black lines in Fig. 23).





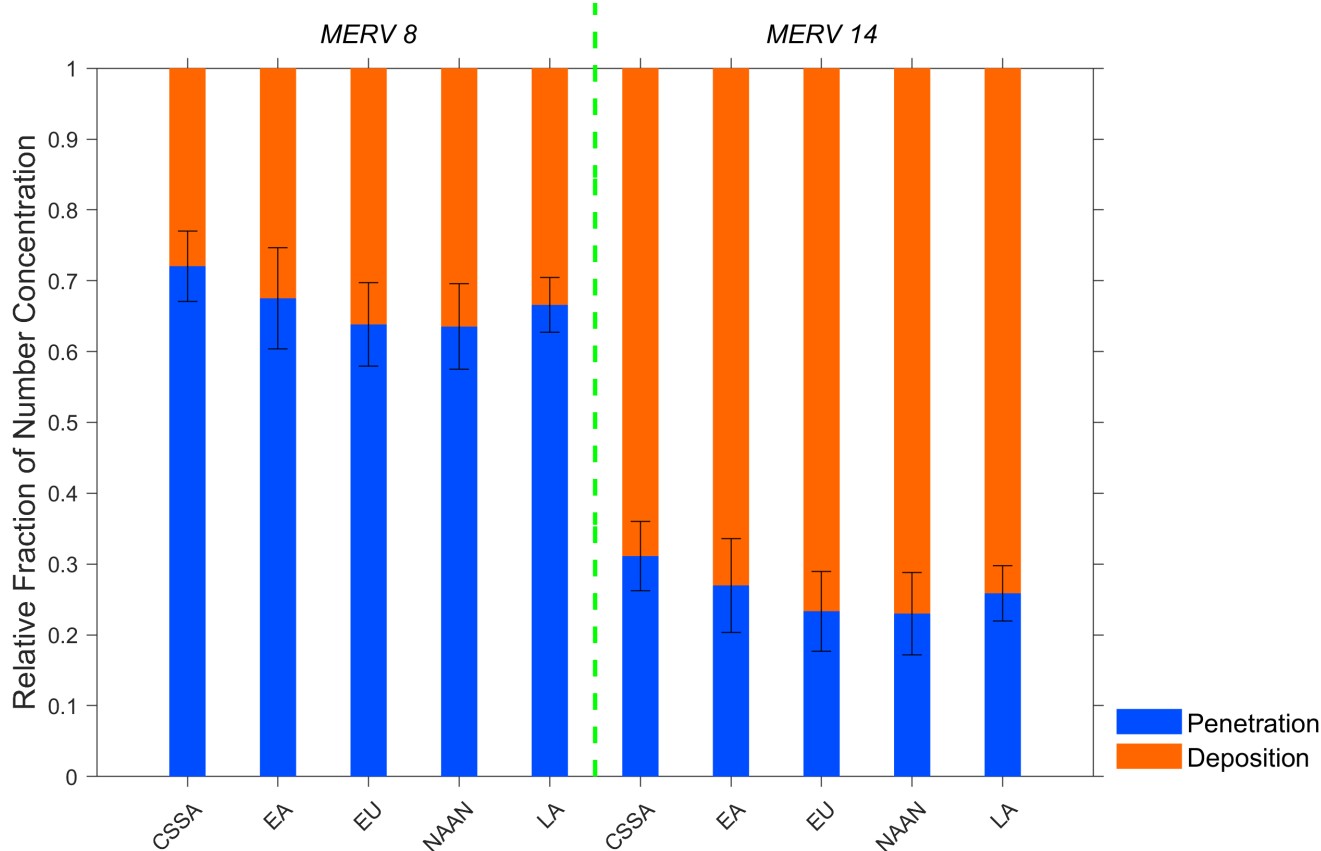

**Figure 25.** Mean HVAC filter penetration and deposition fractions for each geographical region, assuming single-pass MERV 8 and MERV 14 filters.



:70

**Table 1.** List of the number of urban PSDs that have been extracted and analyzed in each region, country, and city.

| Region | Country | City |
|---|---|---|
| Africa (*n*=29) | Botswana (3) | Gaborone (3) |
| | Cape Verde (1) | Praia City (1) |
| | Egypt (12) | Cairo (12) |
| | Kenya (1) | Nairobi (1) |
| | Mali (1) | Bamako (1) |
| | Senegal (1) | Dakar (1) |
| | South Africa (6) | Johannesburg (2) |
| | | Pretoria (4) |
| | Zambia (4) | Mongu (4) |
| Central, South, and Southeast Asia (*n*=70) | India (48) | Durg (4) |
| | | Kanpur (23) |
| | | Mumbai (2) |
| | | New Delhi (13) |
| | | Pune (2) |
| | | Trivandrum (4) |
| | Nepal (4) | Lalitpur (1) |
| | | Dhulikhel (3) |
| | Pakistan (8) | Karachi (3) |
| | | Lahore (3) |
| | | Peshawar (1) |
| | | Rawalpindi (1) |
| | Singapore (6) | Singapore City (6) |
| | Thailand (2) | Chiang Mai (1) |
| | | Silpakorn (1) |
| | Vietnam (2) | Ho Chi Minh (2) |
| East Asia (*n*=138) | China (93) | Beijing (38) |
| | | Guangzhou (14) |
| | | Hong Kong (6) |
| | | Jinan (1) |
| | | Jinhua (1) |
| | | Lanzhou (11) |
| | | Nanjing (2) |
| | | Shanghai (12) |
| | | Shenzhen (3) |
| | | Zhengzhou (2) |
| | | Urumchi (1) |
| | | Wuxi (2) |
| | Japan (17) | Kawasaki (10) |
| | | Sapporo (6) |
| | | Tokyo (1) |
| | South Korea (24) | Gwangju (16) |



| | | Seoul (4) |
|---|---|---|
| | | Ulsan (4) |
| | China: Taiwan (4) | Taipei (3) |
| | | Taichung (1) |
| Europe (*n*=316) | Austria (7) | Vienna (7) |
| | Belgium (1) | Gent (1) |
| | Switzerland (4) | Zurich (4) |
| | Czech Republic (4) | Prague (4) |
| | Germany (68) | Aachen (1) |
| | | Augsburg (2) |
| | | Braunschweig (12) |
| | | Dresden (4) |
| | | Duisburg (1) |
| | | Erfurt (9) |
| | | Essen (5) |
| | | Heidelberg (2) |
| | | Karlsruhe (5) |
| | | Leipzig (27) |
| | Denmark (26) | Copenhagen (25) |
| | | Odense (1) |
| | Spain (17) | Barcelona (7) |
| | | Ciudad Real (2) |
| | | Madrid (8) |
| | Finland (52) | Helsinki (52) |
| | France (10) | Marseilles (5) |
| | | Paris (3) |
| | | Dunkirk (2) |
| | United Kingdom (37) | Birmingham (1) |
| | | Bloomsbury (3) |
| | | Cambridge (7) |
| | | Leeds (3) |
| | | Leicester (2) |
| | | London (16) |
| | | Manchester (5) |
| | Greece (7) | Athens (3) |
| | | Chania (4) |
| | Hungary (9) | Budapest (9) |
| | Italy (31) | Bologna (2) |
| | | Cagliari (1) |
| | | Cassino (2) |
| | | Ispra (4) |
| | | Milan (15) |
| | | Rome (7) |





| | Lithuania (9) | Vilnius (9) |
|---|---|---|
| | Netherland (6) | Amsterdam (4) |
| | | Rotterdam (2) |
| | Norway (3) | Oslo (3) |
| | Portugal (12) | Oporto (10) |
| | | Lisbon (2) |
| | Russia (1) | Tiksi (1) |
| | Sweden (11) | Gothenburg (3) |
| | | Stockholm (8) |
| | Moldova (1) | Chisinau (1) |
| Latin America (*n*=48) | Brazil (38) | Porto Alegre (3) |
| | | São Paulo (35) |
| | Chile (6) | Santiago (6) |
| | Mexico (3) | Mexico City (3) |
| | Cuba (1) | Camagüey (1) |
| North America, Australia and New Zealand (*n*=134) | Australia (10) | Brisbane (2) |
| | | Launceston (7) |
| | | Wollongong (1) |
| | New Zealand (8) | Auckland (8) |
| | Canada (15) | Hamilton (2) |
| | | Toronto (13) |
| | United States (U.S.) (101) | Atlanta (4) |
| | | Boulder (1) |
| | | Buffalo (6) |
| | | Claremont (5) |
| | | Corpus Christi (4) |
| | | Detroit (6) |
| | | Downey (4) |
| | | Fresno (12) |
| | | Houston (1) |
| | | Los Angeles (34) |
| | | New York (5) |
| | | Newark (2) |
| | | Pittsburgh (7) |
| | | Raleigh (1) |
| | | Riverside (8) |
| | | Rochester (1) |
| West Asia (*n*=58) | Iran (23) | Zanjan (23) |
| | Jordan (1) | NA |
| | Kuwait (12) | Fahaheel (12) |
| | Saudi Arabia (9) | Yanbu (9) |
| | Turkey (13) | Istanbul (13) |





**Table 2.** Summary of direct measurements of size-resolved urban aerosol effective densities.

| Country | City | Site Type | Measurement Duration | Measurement Technique | Size (nm) | Effective Density, $\rho_{eff}$ (g cm$^{-3}$) | Reference |
|---|---|---|---|---|---|---|---|
| Denmark | Copenhagen | Traffic (near road) | January - February 2012 | DMA-APM | 75 | 0.934 | Rissler et al. (2014)[a] |
| | | | | | 100 | 0.8078 | |
| | | | | | 150 | 0.4636 | |
| | | | | | 250 | 0.6432 | |
| | | | | | 350 | 0.276 | |
| U.S. | Los Angeles[b] | Urban | September - October 2005 | DMA-APM | 50 | 1.14 | Geller et al. (2006) |
| | | | | | 118 | 1.12 | |
| | | | | | 146 | 1.21 | |
| | | | | | 202 | 1.14 | |
| | | | | | 322 | 0.86 | |
| | | | | | 414 | 0.73 | |
| | Downey[c] | Traffic (near road) | | | 50 | 1.13 | |
| | | | | | 118 | 1 | |
| | | | | | 146 | 0.94 | |
| | | | | | 202 | 0.78 | |
| | | | | | 322 | 0.49 | |
| | | | | | 414 | 0.31 | |
| | Riverside | Urban | | | 50 | 1.4 | |
| | | | | | 118 | 1.4 | |
| | | | | | 146 | 1.29 | |
| | | | | | 202 | 1.06 | |
| China | Beijing | Urban | January 22 - 27 2007 | SMPS/APS-MOUDI | 56-100 | 2.085 | Hu et al. (2012)[d] |
| | | | | | 100-180 | 1.32 | |
| | | | | | 180-320 | 1.336 | |
| | | | | | 320-560 | 1.885 | |
| | | | | | 560-1000 | 1.75 | |
| | | | | | 1000-1800 | 1.655 | |
| | | | | | 1800-3200 | 1.38 | |
| | | | | | 3200-5600 | 2.23 | |
| | | | | | 5600-10000 | - | |
| China | Shanghai | Urban | December 6 - January 12 2012 | TDMA-APM | 50 | 1.36 | Yin et al. (2015) |
| | | | | | 100 | 1.45 | |
| | | | | | 200 | 1.52 | |
| | | | | | 300 | 1.53 | |
| | | | | | 400 | 1.55 | |
| China | Shanghai | Urban | | DMA-APM | 40 | 1.37 | Xie et al. (2017) |
| | | | | | 100 | 1.4 | |





| Country | City | Type | Date | Method | | | Reference |
|---------|------|------|------|--------|---|---|-----------|
| | | | December 21 2014 - January 13 2015 | | 220 | 1.47 | |
| | | | | | 300 | 1.5 | |
| U.S. | Houston | Urban | April 15 - May 31 2009 | DMA-APM | 46 | 1.55 | Levy et al. (2013)[e] |
| | | | | | 81 | 1.55 | |
| | | | | | 151 | 1.54 | |
| | | | | | 240 | 1.54 | |
| | | | | | 350 | 1.5 | |
| Finland | Helsinki | Traffic (near road) | February 10 - 26 2003, January 28 - February 12 2004 August 12 - 27 2003, August 6 - 20 2004 | ELPI-SMPS | 20.3 | 1.04 | Virtanen et al. (2006) |
| | | | | | 72 | 1.5 | |
| | | | | | 18.9 | 0.96 | |
| | | | | | 75.1 | 1.8 | |
| China | Beijing | Semi-Urban | May - June 2016 | DMA-CPMA | 50 | 1.56 | Qiao et al. (2018) |
| | | | | | 80 | 1.55 | |
| | | | | | 100 | 1.51 | |
| | | | | | 150 | 1.5 | |
| | | | | | 240 | 1.47 | |
| | | | | | 350 | 1.42 | |
| U.S. | Atlanta | Urban | August 2009 | TDMA-APM | 107 | 1.58 | McMurry et al. (2002)[f] |
| | | | | | 309 | 1.62 | |

[a]Average of the soot and dense mode during Period I in Rissler et al. (2014)
[b]Referred to as USC in Geller et al. (2006)
[c]Referred to as 710-freeway in Geller et al. (2006)
[d]Average of the polluted and clean episodes in Fig. 4 in Hu et al. (2012)
[e]Average of the values in Fig. 11 in Levy et al. (2013)
[f]Average of $\rho_{eff,2}$ in Table 3 in McMurry et al. (2002)
DMA: Differential Mobility Analyzer
TDMA: Tandem Differential Mobility Analyzer
CPMA: Centrifugal Particle Mass Analyzer
APM: Aerosol Particle Mass Analyzer
ELPI: Electrical Low Pressure Impactor
SMPS: Scanning Mobility Particle Sizer
APS: Aerodynamic Particle Sizer
MOUDI: Micro-Orifice Uniform Deposit Impactor





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
