# Peer review of "Urban Aerosol Size Distributions: A Global Perspective"

_Atmospheric Chemistry and Physics, 2020_

## Referee Comment (RC1) · Anonymous Referee #1 · 28 Jun 2020

This paper presents a compilation of observed particle size distributions in urban environments for many locations around the globe from the existing literature over the last 20 years. An important contribution of this paper is that the authors unified the many size distributions measured by different instruments to mobility diameter-based size distribution and fitted lognormal functions, providing the lognormal function parameters to be used for future studies. The paper also evaluates the implications for human inhalation exposure.

I think this paper will be a very useful reference for the community, since aerosol size distributions are so fundamental to the many science questions. To my knowledge, a compilation like this does not exist yet, so this paper fills an important gap. I commend the authors for their work and recommend publication after the following comments are

addressed:

Major comments: 1. The analysis focuses on the spatial variation, but since the data includes measurements over 20 years, there is temporal variation as well (i.e. particle numbers have decreased over the years in certain areas and increased in others, plus there are seasonal variations within a year as well). In other words, it is reasonable to assume that there is a lot of temporal variation within the data from a single site on various time scales. The paper doesn't mention these issues at all, but it needs to be discussed.

2. Another question is how representative the compiled observations are for urban environments in the different regions. One way to check this (at least for the US region) would be to construct a histogram of PM2.5 concentrations from the studies included here and compare these to EPA long term data.

3. Rearrange figures so that they are referenced in sequential order. For example, Figure 9 is referred to as the first figure in the paper, which is strange. This happens because the method description is separated out from the results, which is the usual way of structuring a manuscript. However, in this case, I think it would work to have one section (Section 3) about the "Geographical trends in size-resolved urban aerosol effective densities" which would include the methods and the results. Similar for the other topics covered. This would make the paper more cohesive and easier to read.

Minor comments: 1. Introduction: Mention the relevance of particle size for aerosol climate impacts. As global and regional climate models move to higher and higher spatial resolution, a compilation of size distributions in urban areas such as presented in this paper will be very useful.

2. A list of symbols and abbreviations would be helpful.

3. Section 3.1, eq. 1-4: What is the relationship of rho_eff and rho_p. Is to some extent mentioned later in eq. (6), but would be good to get this out of the way here.

4. p. 7, l. 12: The shape parameter is usually not used to characterize soot, but rather fractal dimension.

5. p. 7, l. 15: Complications in estimating rho_eff for coarse mode because of mixing state – I think the same can be said for the accumulation mode.

6. p. 7, l. 16: Avoid starting a sentence with a mathematical symbol. (This happens several times throughout the manuscript).

7. p. 9, l. 72: Notation: Is log referring to log10 or natural log?

8. p. 9, equation (7): Determining the number of modes to achieve best fit – how many modes are usually needed to achieve best fit? (i.e. is it mostly 3 modes or are sometimes more needed?)

9. Figure 1: Include abbreviation of different regions from the legend in the figure caption.

10. Tables 1 & 2: Include sampling frequency for the various studies or any information if the size distribution is an instantaneous snapshot or the result of a time-average.

11. Figure 5: Considering that the Aitken mode is only incompletely captured, the fit is doing some extrapolation. Mathematically, this is fine, but given that there is some noise, the fitted size distribution parameters could be way off. How reliable is the fit in such a case?

12. p. 17, l. 10: It would be helpful to include some references to papers where the issues mentioned here are considered and also give the reader some context about how important these issues might be.

13. Figure 17: Are the PSD assigned to UB and to TR significantly different (in a technical sense)? In other words, does the grouping into UB and TR make sense? Did you try to do a cluster analysis based on the PSD shapes to see what different types in terms of shape emerge (=size distribution regimes), and if they tend to include

size distributions from the same geographical region? Given the many factors that influence the shape of size distributions (as mentioned on p. 23), I question if the a priori separation into geographical regions is the best way to look at the data.

14 Section 10 and 11: It is not clear to me what the take home message is of these sections. This needs to be more clearly formulated and presented. To better gauge the differences in particle exposure for different regions, I recommend introducing a couple of integrated metrics, e.g. number concentrations of particles smaller than 100 nm/1 micron etc.

15. Data availability statement: Suggest posting an electronic version of the supplement to some databank (with doi) so that it can be easily used by the community for future studies.

---

## Referee Comment (RC2) · Ulrich Pöschl (Referee) · 19 Oct 2020

Dear Colleagues,

In the access review prior to publication of the preprint/discussion paper in ACPD, three reviewers rated the manuscript as good or excellent in all review criteria. Only one of the reviewers, however, followed up with a full review and public comments. In spite of numerous reminders and calls for additional reviewers, it was unfortunately not possibly to obtain another review and public comments in due time.

Thus, I recommend to revise and optimize the manuscript in response to the thorough review and constructive comments of Anonymous Referee Point 1. Additional reviewers may be consulted upon peer review completion for the revised manuscript.

[Figure]

With best wishes,

Ulrich Pöschl (handling editor)
* * *

---

## Editor Comment (EC1) · Ulrich Pöschl (Editor) · 2 Nov 2020

Dear Colleagues,

Please take the following comments from Anonymous Referee #2 into account when revising the manuscript.

Many thanks and best wishes, Uli Pöschl (handling editor)

Comments on Wu and Boor (2020), submitted to ACP

I commend the authors for attempting this expansive review, which will be very helpful to the aerosol community. I apologize for not submitting a review in time, and even now my work is incomplete. Normally, I spend a couple of hours on a review; I have spent

well over that on this manuscript and I am still less than a third of the way through it. Some of that is apparently because of ACP's format requirements - with 25 figures at the end, reading and relating discussions to figures becomes very difficult. For what it's worth, the comments I have so far are mostly grammatical or ways to frame it better.

Some general comments:

1. This review attempts to not only review worldwide PSDs, but also applications in exposure and filtration. The latter two are fields in their own right for good reason and should perhaps be separated as a companion paper(or papers) that can be reviewed by experts in those fields. 2. Perhaps this expansive work is better as a monograph that can be fact-checked, verified, and copy-edited by paid staff, or at least split into three or four papers, with measurement experts, atmospheric chemists, inhalation modelers, and building/indoor air quality experts reviewing the different parts. The latter might also make it easier for an individual reviewer to spend a few hours and do a satisfactory job within their expertise. 3. One of my comments was whether 1 nm species can even be called aerosol particles, and that seems a definitional rabbit hole. If one were to focus on urban air quality and exposure/HVAC applications, the sub-10 nm aerosols may not even be practically relevant or can be left for another paper.

Specific comments: Abstract, line 3: replace "enable for characterization" with "enable characterization"

Introduction, line 2/para 1: "spanning in size from a single nanometer to tens of micrometers" - can 1 nm actually be considered an aerosol particle? Seems like an issue of definition - has previous work established a lower size limit for the transition from molecules to aerosol particles? (The authors limit their own review to aerosol sizes above 3 nm.)

Introduction, para 2: "Measurement of urban aerosol PSDs provides..." and "UFPs...are associated with various deleterious human health outcomes." These statements should be backed up by references to epidemiological and toxicological studies,

especially as the paragraph criticizes current air quality standards that are based on PM mass (and which have been quite effective at improving human health). A problem could very well be (as the authors state) that most PSD measurements are not long-term, which is necessary for epidemiological studies; but that surely is the end product of the current manuscript (e.g. "we recommend more long-term studies") rather than a declarative statement at the outset. Toxicological studies do not require long-term studies; the authors should be able to cite studies that back up their claims.

Introduction/last paragraph: replace "enables for identification of gaps" with "enables identification of gaps" (Please check the manuscript for similar language corrections, I will stop now.) Also, the authors should probably state in the introduction that their 3 nm to 10 micron size range refers to electrical mobility diameter.

Methodology: What is the time duration of an individual PSD? Typical SMPS scans are 2 minutes; optical sizing can be at 1 Hz. Or do they mean 793 measurement campaigns or days of data?

Abbreviations for geographical regions: I had a hard time remembering what each stood for, e.g. CSSA or WA. Suggest the use of Africa, EAsia, WAsia, CSSAsia, etc. Why is the EU separated from North America/Australia/New Zealand - these are all OECD regions with greater similarities in emissions control regimes compared to Asia or Africa.

Figure 9 is introduced and discussed before Figure 2. Suggest re-ordering the figures in the order they are discussed.

Section 3.4: what did Rissler et al. report as density for soot aggregates - that seems relevant for group C (traffic environments) if EC can be a quarter of particle mass. A statement about the material density of soot appears at the end of the first paragraph of Sec 3.5 - which is supposed to be about coarse particles that likely don't have a lot of EC. Perhaps that should be moved to Sec 3.4. The last sentence seems unclear - if both soot particles and denser(?) components can exist in the 400-1000 nm size

range, why pick one over the other unless one component dominates the mass?

---

## Author Comment (AC1) · 15 Nov 2020

The comment was uploaded in the form of a supplement:
https://acp.copernicus.org/preprints/acp-2020-92/acp-2020-92-AC1-supplement.pdf

---

## Author Comment (AC2) · 16 Nov 2020

**Journal:** *Atmospheric Chemistry and Physics*

**MS Title:** Urban Aerosol Size Distributions: A Global Perspective

**MS Authors:** Tianren Wu and Brandon E. Boor

**MS No.:** acp-2020-92

**MS Type:** Review Article

We appreciate the editor's efforts to help us obtain reviews and comments for our lengthy review manuscript. We have addressed RC1 and EC1 as per the editor's recommendation.

---

## Author Comment (AC3) · 16 Nov 2020

- 1 Journal: Atmospheric Chemistry and Physics
- 2 MS Title: Urban Aerosol Size Distributions: A Global Perspective
- 3 MS Authors: Tianren Wu and Brandon E. Boor
- 4 MS No.: acp-2020-92
- 5 MS Type: Review Article

We would like to thank the two reviewers for their insightful comments and helpful feedback. We have revised our manuscript,
as outlined below. Comments from the reviewers are in italic black font. Our responses are in normal black font. Revised text
are in normal red font. Revised figures and tables are included.

**9 Response to Anonymous Referee #2**

10 I commend the authors for attempting this expansive review, which will be very helpful to the aerosol community. I apologize

11 for not submitting a review in time, and even now my work is incomplete. Normally, I spend a couple of hours on a review; I

12 have spent well over that on this manuscript and I am still less than a third of the way through it.

Some of that is apparently because of ACP's format requirements - with 25 figures at the end, reading and relating discussions to figures becomes very difficult. For what it's worth, the comments I have so far are mostly grammatical or ways to frame it better.

16 Some general comments:

17 1. This review attempts to not only review worldwide PSDs, but also applications in exposure and filtration. The latter two

18 are fields in their own right for good reason and should perhaps be separated as a companion paper (or papers) that can be

19 reviewed by experts in those fields. 2. Perhaps this expansive work is better as a monograph that can be fact-checked, verified,

20 and copy-edited by paid staff, or at least split into three or four papers, with measurement experts, atmospheric chemists,

21 inhalation modelers, and building/indoor air quality experts reviewing the different parts. The latter might also make it easier

22 for an individual reviewer to spend a few hours and do a satisfactory job within their expertise.

23 Response: We understand that this review is very long with nearly 30 figures. One of the most important reasons that we 24 chose to submit our review to ACP is that it is a high-quality, open-access, interdisciplinary journal without word/page limits. 25 It has a broad audience interested in atmospheric science, air quality, climate, and human health impacts. This long review 26 allows readers with different backgrounds to retrieve the relevant information they need, but not necessarily having to read the 27 entire paper. We feel that the geographical variations in urban PSDs provides important implications for human exposure and 28 indoor air quality, which may be of interest to many readers. Therefore, we think it is logical to discuss these topics in one 29 paper. In addition, expansive and lengthy reviews have been published in ACP in the past, such as Fuzzi et al. (2015), Lawrence and Lelieveld (2010), Kukkonen et al. (2012), Bartels-Rausch et al. (2014), Vihma et al. (2014), and Tang et al. (2019). 30

3. One of my comments was whether 1 nm species can even be called aerosol particles, and that seems a definitional rabbit 32 hole. If one were to focus on urban air quality and exposure/HVAC applications, the sub-10 nm aerosols may not even be

33 *practically relevant or can be left for another paper.*

Response: Although previous papers referred to the species between 1-3 nm as nanocluster aerosols, here we would not like to debate if they are aerosol particles. 'a diverse mixture of liquid and solid particles spanning in size from a single nanometer

- 36 to tens of micrometers' has been changed to 'a diverse mixture of liquid and solid particles spanning in size from several 37 nanometers to tens of micrometers'.
- 38 Specific comments: Abstract, line 3: replace "enable for characterization" with "enable characterization"
- 39 **Response:** This has been changed.

40 Introduction, line 2/para 1: "spanning in size from a single nanometer to tens of micrometers" - can 1 nm actually be

41 considered an aerosol particle? Seems like an issue of definition - has previous work established a lower size limit for the

42 transition from molecules to aerosol particles? (The authors limit their own review to aerosol sizes above 3 nm.)

43 Response: 'spanning in size from a single nanometer to tens of micrometers' has been changed to 'spanning in size from 44 several nanometers to tens of micrometers'.

- Introduction, para 2: "Measurement of urban aerosol PSDs provides..." and "UFPs...are associated with various deleterious human health outcomes." These statements should be backed up by references to epidemiological and toxicological studies, especially as the paragraph criticizes current air quality standards that are based on PM mass (and which have been quite effective at improving human health). A problem could very well be (as the authors state) that most PSD measurements are not longterm, which is necessary for epidemiological studies; but that surely is the end product of the current manuscript (e.g. "we recommend more long-term studies") rather than a declarative statement at the outset. Toxicological studies do not
- 51 *require long-term studies; the authors should be able to cite studies that back up their claims.*
- 52 **Response:** References have been added to para. 2 to support the claims.
- 53 'Of particular importance are measurements of urban aerosol particle size distributions (PSDs). Measurement of urban aerosol 54 PSDs provides a basis for in-depth evaluation of size-resolved aerosol transport and transformation processes in the urban 55 atmosphere (e.g. Hussein et al., 2004; Peng et al., 2014; Salma et al., 2011; Wehner et al., 2008; Wu et al., 2008), air pollution 56 source apportionment (e.g. Harrison et al., 2011; Sowlat et al., 2016; Wang et al., 2013b), aerosol deposition in the human 57 respiratory system (e.g. Hussein et al., 2019, 2020; Kodros et al., 2018; Zwozdziak et al., 2017), and associated toxicological effects on the human body (e.g. Bentayeb et al., 2015; Burnett et al., 2014; Oberdürster, 2000; Shiraiwa et al., 2017; Tseng et 58 59 al., 2015; Wong et al., 2015). In addition, the measurement of aerosol PSD is important for evaluating the global climate 60 change, since aerosol size strongly affects the interaction of aerosol with radiation and its ability to form fog and cloud droplets (Mahowald, 2011; Seinfeld and Pandis, 2012; Zhang et al., 2012). Despite the atmospheric and health relevance of urban 61 62 PSDs, long-term aerosol measurements are often focused on size-integrated concentration metrics, such as PM2.5, that lack essential size-resolved information. While urban aerosol PSD measurements have been conducted in cities around the globe, 63 64 they are often short in duration and not performed as part of routine air quality monitoring. Urban PSDs provide a more 65 complete assessment of an aerosol population, beyond what can be achieved with size-integrated metrics. Of particular 66 importance are urban PSDs that capture the ultrafine particle regime (UFP, 1 to 100 nm). UFPs tend to dominate number 67 PSDs, penetrate deep into the lung, and are associated with various deleterious human health outcomes (Allen et al., 2017; 68 Delfino et al., 2005; Jiang et al., 2009; Li et al., 2016, 2017a; Oberdörster, 2001; Oberdörster et al., 2004, 2005; Rychlik et al., 69 2019; Sioutas et al., 2005; Weichenthal et al., 2017).'
- 70

Introduction/last paragraph: replace "enables for identification of gaps" with "enables identification of gaps" (Please check the manuscript for similar language corrections, I will stop now.) Also, the authors should probably state in the introduction that their 3 nm to 10 micron size range refers to electrical mobility diameter.

74 **Response:** The language has been corrected. Clarification has been made: '3 to 10,000 nm as electrical mobility diameter'.

Methodology: What is the time duration of an individual PSD? Typical SMPS scans are 2 minutes; optical sizing can be at 1
 Hz. Or do they mean 793 measurement campaigns or days of data?

**Response:** The time duration of each individual PSD has been listed in the SI. Given the great uncertainty of instantaneous snapshots, the PSDs and size-resolved  $\rho_{eff}$  collected in this study are all time-average results over a given sampling period.

79 The following sentences have been added to Section 2 and Section 3.2:

80 'These articles presented n=793 individual PSDs (182 of which covered both the sub-micron and coarse regime), which have

81 been reported in previous peer-review journal articles in the form of figures or fitting parameters'; 'All the PSDs are the results

82 of a time-average over certain sampling periods'.

Abbreviations for geographical regions: I had a hard time remembering what each stood for, e.g. CSSA or WA. Suggest the
 use of Africa, EAsia, WAsia, CSSAsia, etc. Why is the EU separated from North America/Australia/New Zealand - these are
 all OECD regions with greater similarities in emissions control regimes compared to Asia or Africa.

86 **Response:** A list of symbols and abbreviations has been added to Appendix A, which could help the readers to identify the 87 abbreviations for geographical regions more easily.

**88 Appendix A: List of symbols and abbreviations.**

**89 **Table A1.** List of symbols and abbreviations.**

[revised manuscript text omitted]

91 We would like to group the PSDs according to geographical regions to investigate the spatial variations in PSDs, therefore we 92 separated EU from North America. In the early stages of our analysis, we grouped Australia and New Zealand as Oceania, 93 separately from North America. However, the sample size of Oceania is too small for conducting a meaningful analysis. 94 Therefore, we grouped Australia and New Zealand with North America, rather than Europe, since their population density is 95 closer to North America.

96

97 Figure 9 is introduced and discussed before Figure 2. Suggest re-ordering the figures in the order they are discussed.

98 **Response:** The figures have been re-ordered.

99 Section 3.4: what did Rissler et al. report as density for soot aggregates - that seems relevant for group C (traffic environments)

00 *if EC can be a quarter of particle mass. A statement about the material density of soot appears at the end of the first paragraph*

01 of Sec 3.5 - which is supposed to be about coarse particles that likely don't have a lot of EC. Perhaps that should be moved to

02 Sec 3.4. The last sentence seems unclear - if both soot particles and denser(?) components can exist in the 400-1000 nm size

03 range, why pick one over the other unless one component dominates the mass?

**Response:** Direct measurements of effective density are missing for the particles between 400-1000 nm. The measurements in Rissler et al. (2014) are from 50 to 350 nm. Since at 350 nm, dense particles contributed more than 80% to the total particle mass in Rissler et al. (2014), the effective density of dense mode particles (1.46 g cm-3) was chosen for the size range of 400-1000 nm. In addition, the HR-AMS analysis in Rissler et al. (2014) indicated that the dense mode particles were made of ammonium nitrate, ammonium sulfate, and organic carbon. Although soot particles also exist in the size range of 400-1000 nm, the fraction of particles with greater effective densities than the dense mode particles, such as crustal materials, could be

10 higher (Daher et al., 2013) than in the size range of 50-350 nm, which to some extent offsets the low density of soot particles.

11 The sentence, 'The inherent material density of diesel soot was measured to be 1.77 g cm-3 (Park et al., 2004)', has been 12 removed.

 been corrected.

- 39 Table 1. List of the number of urban PSDs that have been extracted and analyzed in each region, country, and city.
- 40

| Region                 | Country          | City             |
|------------------------|------------------|------------------|
|                        | Botswana (3)     | Gaborone (3)     |
|                        | Cape Verde (1)   | Praia City (1)   |
|                        | Egypt (12)       | Cairo (12)       |
| Africa ( n =29) | Kenya (1)        | Nairobi (1)      |
|                        | Mali (1)         | Bamako (1)       |
|                        | Senegal (1)      | Dakar (1)        |
|                        | South Africa (6) | Johannesburg (2) |
|                        |                  | Pretoria (4)     |
|                        | Zambia (4)       | Mongu (4)        |
|                        | India (48)       | Durg (4)         |

|                      |                    | Kanpur (23)        |
|----------------------|--------------------|--------------------|
|                      |                    | Mumbai (2)         |
|                      |                    | New Delhi (13)     |
|                      |                    | Pune (2)           |
|                      |                    | Trivandrum (4)     |
|                      | Nepal (4)          | Lalitpur (1)       |
| Central, South,      |                    | Dhulikhel (3)      |
| and Southeast        | Pakistan (8)       | Karachi (3)        |
| Asia ( n =70) |                    | Lahore (3)         |
|                      |                    | Peshawar (1)       |
|                      |                    | Rawalpindi (1)     |
|                      | Singapore (6)      | Singapore City (6) |
|                      | Theiland (2)       | Chiang Mai (1)     |
|                      | I natiand $(2)$    | Silpakorn (1)      |
|                      | Vietnam (2)        | Ho Chi Minh (2)    |
|                      |                    | Beijing (38)       |
|                      |                    | Guangzhou (14)     |
|                      |                    | Hong Kong (6)      |
|                      |                    | Jinan (1)          |
|                      |                    | Jinhua (1)         |
|                      |                    | Lanzhou (11)       |
|                      | China (95)         | Nanjing (2)        |
|                      |                    | Shanghai (12)      |
|                      |                    | Shenzhen (3)       |
| East Asia            |                    | Zhengzhou (2)      |
| ( n =138)     |                    | Urumchi (1)        |
|                      |                    | Wuxi (2)           |
|                      |                    | Kawasaki (10)      |
|                      | Japan (17)         | Sapporo (6)        |
|                      |                    | Tokyo (1)          |
|                      |                    | Gwangju (16)       |
|                      | South Korea (24)   | Seoul (4)          |
|                      |                    | Ulsan (4)          |
|                      | Chine: Taiwan (1)  | Taipei (3)         |
|                      | China: Laiwan (4)  | Taichung (1)       |
|                      | Austria (7)        | Vienna (7)         |
|                      | Belgium (1)        | Gent (1)           |
| Europe $(-216)$      | Switzerland (4)    | Zurich (4)         |
| Europe ( $n=510$ )   | Czech Republic (4) | Prague (4)         |
|                      | Germany (68)       | Aachen (1)         |
|                      |                    | Augsburg (2)       |
|                      |                    |                    |

|                         |                        | Braunschweig (12) |
|-------------------------|------------------------|-------------------|
|                         |                        | Dresden (4)       |
|                         |                        | Duisburg (1)      |
|                         |                        | Erfurt (9)        |
|                         |                        | Essen (5)         |
|                         |                        | Heidelberg (2)    |
|                         |                        | Karlsruhe (5)     |
|                         |                        | Leipzig (27)      |
|                         | Denmark (26)           | Copenhagen (25)   |
|                         | Definitiar (20)        | Odense (1)        |
|                         | Spain (17)             | Barcelona (7)     |
|                         |                        | Ciudad Real (2)   |
|                         |                        | Madrid (8)        |
|                         | Finland (52)           | Helsinki (52)     |
|                         |                        | Marseilles (5)    |
|                         | France (10)            | Paris (3)         |
|                         |                        | Dunkirk (2)       |
|                         |                        | Birmingham (1)    |
|                         |                        | Cambridge (7)     |
|                         | United Kingdom (27)    | Leeds (3)         |
|                         | Officed Kingdolfi (37) | Leicester (2)     |
|                         |                        | London (19)       |
|                         |                        | Manchester (5)    |
|                         | Greece (7)             | Athens (3)        |
|                         |                        | Chania (4)        |
|                         | Hungary (9)            | Budapest (9)      |
|                         |                        | Bologna (2)       |
|                         |                        | Cagliari (1)      |
|                         | Italy (31)             | Cassino (2)       |
|                         |                        | Ispra (4)         |
|                         |                        | Milan (15)        |
|                         |                        | Rome (7)          |
| Europe ( n =316) | Lithuania (9)          | Vilnius (9)       |
|                         | Netherland (6)         | Amsterdam (4)     |
|                         |                        | Rotterdam (2)     |
|                         | Norway (3)             | Oslo (3)          |
|                         | Portugal (12)          | Oporto (10)       |
|                         | 1 ortugur (12)         | Lisbon (2)        |
|                         | Russia (1)             | Tiksi (1)         |
|                         | Sweden (11)            | Gothenburg (3)    |

|                       |                            | Stockholm (8)      |
|-----------------------|----------------------------|--------------------|
|                       | Moldova (1)                | Chisinau (1)       |
|                       | Drazil (28)                | Porto Alegre (3)   |
| Tatin America         | Brazii (38)                | São Paulo (35)     |
| Laun America $(n-48)$ | Chile (6)                  | Santiago (6)       |
| (n-40)                | Mexico (3)                 | Mexico City (3)    |
|                       | Cuba (1)                   | Camagüey (1)       |
|                       | Australia (10)             | Brisbane (2)       |
| North America,        |                            | Launceston (7)     |
| Australia and         |                            | Wollongong (1)     |
| New Zealand           | New Zealand (8)            | Auckland (8)       |
| ( n =134)      | Canada (15)                | Hamilton (2)       |
|                       | Callada (15)               | Toronto (13)       |
|                       |                            | Atlanta (4)        |
|                       |                            | Boulder (1)        |
|                       |                            | Buffalo (6)        |
|                       |                            | Claremont (5)      |
|                       |                            | Corpus Christi (4) |
|                       | United States (U.S.) (101) | Detroit (6)        |
| North America,        |                            | Downey (4)         |
| Australia and         |                            | Fresno (12)        |
| New Zealand           |                            | Houston (1)        |
| ( n =134)      |                            | Los Angeles (34)   |
|                       |                            | New York (5)       |
|                       |                            | Newark (2)         |
|                       |                            | Pittsburgh (7)     |
|                       |                            | Raleigh (1)        |
|                       |                            | Riverside (8)      |
|                       |                            | Rochester (1)      |
|                       | Iran (23)                  | Zanjan (23)        |
| West Asia             | Jordan (1)                 | NA                 |
| (n=58)                | Kuwait (12)                | Fahaheel (12)      |
| (11 30)               | Saudi Arabia (9)           | Yanbu (9)          |
|                       | Turkey (13)                | Istanbul (13)      |

43 2. A paragraph of discussion on the influence of environmental factors in different geographical locations on filter performance
 44 has been added to Section 10.

45 'In addition, different geographical locations may result in different environmental factors, which can influence the 46 performance of a filter. For example, the dendrites of the deposited particles may experience more frequent collapse in

- 47 geographical regions with annual rainy seasons when the relative humidity increases significantly. The dendrites may 48 restructure to adopt a compact morphology under elevated humidity due to the capillary effect of condensed water, which will 49 lower the filtration efficiency and pressure drop across the filter (Pei et al., 2019). In addition, the hygroscopic content of the 49 dendrites may transition to form a liquid film on the filter fiber, and decrease the filtration efficiency before eventually reaching 49 an equilibrium (Li et al., 2014; Walsh et al., 1996). This is attributes to the low viscosity of the liquid film, which allows the 49 liquid particles to coalesce, relocate, and drain along the filter fiber.'
- 53
- 54 3. Errors in some figures have been corrected. Order of the figures has been rearranged.

56 **Figure 1.** Effective densities ( $\rho_{eff}$ ) as derived from different values of dynamic shape factors ( $\chi$ ) and particle densities ( $\rho_p$ ), 57 assuming the value of  $\frac{c_C(D_{ve})}{c_c(D_{em})}$  is approximately unity for coarse particles.

---

## Author Response (AR1)

- 1 Journal: Atmospheric Chemistry and Physics
- 2 MS Title: Urban Aerosol Size Distributions: A Global Perspective
- 3 MS Authors: Tianren Wu and Brandon E. Boor
- 4 MS No.: acp-2020-92
- 5 MS Type: Review Article

We would like to thank the two reviewers for their insightful comments and helpful feedback. We have revised our manuscript,
as outlined below. Comments from the reviewers are in italic black font. Our responses are in normal black font. Revised text
are in normal red font. Revised figures and tables are included.

- 9 Response to Anonymous Referee #1
- 10 This paper presents a compilation of observed particle size distributions in urban environments for many locations around the

11 globe from the existing literature over the last 20 years. An important contribution of this paper is that the authors unified the

12 many size distributions measured by different instruments to mobility diameter-based size distribution and fitted lognormal

13 *functions, providing the lognormal function parameters to be used for future studies. The paper also evaluates the implications*

14 for human inhalation exposure.

15 I think this paper will be a very useful reference for the community, since aerosol size distributions are so fundamental to the

16 many science questions. To my knowledge, a compilation like this does not exist yet, so this paper fills an important gap. I

17 commend the authors for their work and recommend publication after the following comments are addressed:

Major comments: 1. The analysis focuses on the spatial variation, but since the data includes measurements over 20 years, there is temporal variation as well (i.e. particle numbers have decreased over the years in certain areas and increased in others, plus there are seasonal variations within a year as well). In other words, it is reasonable to assume that there is a lot of temporal variation within the data from a single site on various time scales. The paper doesn't mention these issues at all, but it needs to be discussed.

Response: We agree that temporal variations exist for the data collected from a single site. Section 8.5 Temporal variations of urban aerosol PSDs has been added to discuss this in detail:

**25 '8.5 Temporal variations of urban aerosol PSDs**

26 Since the PSD observations collected in this study span over 20 years, temporal variations are expected for the PSDs measured 27 at the same sampling site, although the analysis of this study focuses on the geographical variations. The temporal variations 28 may exist on different time scales, from short-term hourly variations to long-term yearly changes. The diurnal, weekly and 29 seasonal variations in urban aerosol number PSDs, due to the changes in meteorological conditions and emission sources, 30 have been discussed in many studies (Babu et al., 2016; Baxla et al., 2009; Gómez-Moreno et al., 2011; Hussein et al., 2004; 31 Kanawade et al., 2014; Ketzel et al., 2003, 2004; Kim et al., 2002; Mönkkönen et al., 2005; Peng et al., 2014; Salma et al., 32 2011, 2014; Sowlat et al., 2016; Stanier et al., 2004a; Wang et al., 2014, 2013b; Wehner et al., 2004a, 2008; Wehner and 33 Wiedensohler, 2003; Wiedensohler et al., 2009; Wu et al., 2008; Yue et al., 2013, 2008; Zhang et al., 2017).

34

In addition, long-term trends in the urban aerosol number PSDs, UFP concentrations, and size-integrated metrics (e.g. PM2.5, PM10) were reported in several studies, and they have been associated with changes in economic conditions and emission sources, adaption of control strategies, and implementation of regulations. The long-term measurements of urban aerosol

38 PSDs and gaseous pollutants at Rochester, NY showed significant emission reductions in PM2.5, UFP concentrations, and 39 major gaseous pollutants (e.g. SO2, NOx, CO) over the last two decades (Masiol et al., 2018; Squizzato et al., 2018, 2019). A 40 downward trend in particle number concentrations was observed from 2005 to 2011 and it was attributed to reductions in fuel 41 sulfur content and economic conditions. The annual decrease was estimated to be -323 particles/cm3/year, with the main 42 contribution from the particles in the range of 11-50 nm. Subsequently, a small upward trend in particle number concentrations was observed between 2011 and 2017 due to the increase of vehicular traffic. A decrease in secondary sulfate and nitrate in 43 44 PM2.5 was observed from 2005-2016, likely due to the implementation of mitigation measures, closure of coal-fired power 45 plants, and improvements of the fuel quality (Masiol et al., 2019; Squizzato et al., 2018). Long-term measurements over the 46 last two decades in Australia showed slight declines in PM2.5 and PM10 due to the effectiveness of emission reduction policies 47 for vehicle emissions and power generations (de Jesus et al., 2020). Long-term measurements by German Ultrafine Aerosol 48 Network present statistically significant decreasing trends of particle number concentration and equivalent black carbon in 49 urban sites in Germany, which are mainly resulted from the mitigation policies for road transport and residential emissions (Sun et al., 2020). With the air pollution prevention and control action plan in Beijing, its annual PM2.5 decreased from ~90 50 51 μg m-3 in 2013 to ~60 μg m-3 in 2017 (Cheng et al., 2019; Zhang et al., 2019). The analysis showed that coal-fired boiler 52 control, clean fuels in the residential sector, and optimize industrial structure were the most effective strategies for emission 53 reduction and pollution control. An important implication from these long-term monitors is that the combination of emission 54 policies and mitigation techniques can effectively reduce aerosol concentrations and improve the urban air quality.'

55

2. Another question is how representative the compiled observations are for urban environments in the different regions. One
way to check this (at least for the US region) would be to construct a histogram of PM2.5 concentrations from the studies
included here and compare these to EPA long term data.

59 Response: The length of the measurements may somehow indicate how representative the PSDs are. Some of the long-term 60 PSD observations collected in this study may be representative of a given urban environment. However, we would like to carefully present the observations we collected and discuss them as aggregates, but not to conclude any representative or 61 62 characteristic urban PSDs in a given city or geographical region, as we understand the profile and magnitude of PSDs could 63 be site-specific, even within the same city. In addition, the urban PSDs at a given sampling site exhibit temporal variations 64 across varying time scales. By nature, it is difficult to use a single PSD to represent a city or geographical region. To more 65 confidently identify the characteristic PSDs in urban environments, long-term and high spatial-resolution measurements are 66 needed, as we discussed in the last section of the paper.

Figure 7 was added to compare the PM2.5 derived from the PSDs collected in this study with those measured by local monitoring stations in the same city over the same sampling periods.

- 70 aPSDs from Ding et al. (2017), PM2.5 from Shanghai Environment Monitoring Center at Hongkou Liangcheng Station,
- 71 including four different sampling periods.
- 72 bPSDs from Cabada et al. (2004), PM2.5 from EPA monitoring station (42-003-0021), including four different sampling
- 73 periods.
- cPSD from Watson et al. (2002), PM2.5 from EPA monitoring station (06-019-0008).
- 75 dPSDs from Harrison et al. (2012), PM2.5 from UK Automatic Urban and Rural Monitoring Network at London Marylebone
- 76 Road. Three PSDs were measured at three different sampling sites over the same period.
- 77 Figure 7. Comparison of the PM2.5 derived from the PSDs collected in this study (blue circular markers) with those measured
- 78 by local monitoring stations in the same city over the same sampling periods (green diamond markers and error bars). The
- 79 green diamond markers represent the mean values of the PM2.5 from local sampling stations over the sampling period of the
- 80 corresponding PSD and the error bars represent the standard deviations.
- 81

- 82 The following discussions were added to Section 4.1.5:
- 83 'To validate the PSDs compiled in this study, we selected several cities to compare the PM2.5 derived from the compiled PSDs
- 84 (Fig.7; blue circular markers) with those measured by local monitoring stations over the same sampling periods (green diamond

- markers and error bars). The green diamond markers in Fig. 7 represent the mean values of the PM2.5 from local sampling
  stations measured over the sampling period of the corresponding PSD and the error bars represent the standard deviations.
  The blue circular markers represent the derived PM2.5 from mass PSDs. The PSDs from four studies (Cabada et al., 2004;
  Ding et al., 2017; Harrison et al., 2012; Watson et al., 2002) were selected since long-term PM2.5 data from a monitoring station
  near the PSD sampling site are available. The PM2.5 derived from the PSDs exhibit a good agreement with those measured at
  local sampling stations, indicating the validity of the collected PSDs. Small discrepancies exist due to the difference in
  sampling locations between the PSD and local PM2.5 measurements.'
- 92 3. Rearrange figures so that they are referenced in sequential order. For example, Figure 9 is referred to as the first figure in 93 the paper, which is strange. This happens because the method description is separated out from the results, which is the usual 94 way of structuring a manuscript. However, in this case, I think it would work to have one section (Section 3) about the 95 "Geographical trends in size-resolved urban aerosol effective densities" which would include the methods and the results. 96 Similar for the other topics covered. This would make the paper more cohesive and easier to read.
- **Response:** The order of figures has been rearranged. The results and discussions of the urban aerosol effective densities have
   been move to Section 3.7.

[revised manuscript text omitted]

67 *Minor comments: 1. Introduction: Mention the relevance of particle size for aerosol climate impacts. As global and regional* 68 *climate models move to higher and higher spatial resolution, a compilation of size distributions in urban areas such as* 69 *presented in this paper will be very useful.*

- 70 **Response:** The following texts have been added to Introduction:
- 71 'In addition, the measurement of aerosol PSD is important for evaluating the global climate change, since aerosol size strongly
- affects the interaction of aerosols with radiation and its ability to form fog and cloud droplets (Mahowald, 2011; Seinfeld and
- 73 Pandis, 2012; Zhang et al., 2012).' 'As the climate models have been improved significantly in terms of spatial resolution, a
- compilation of urban PSDs can also serve as useful inputs for models to estimate the direct and indirect influence of aerosol
- 75 on regional and global climates.'
- 76 2. A list of symbols and abbreviations would be helpful.
- 77 **Response:** A list of symbols and abbreviations has been added to Appendix A.

**78 Appendix A: List of symbols and abbreviations.**

79 **Table A1.** List of symbols and abbreviations.

[revised manuscript text omitted]

- 92  $\rho_{eff} = \frac{\rho_p}{\chi^3},\tag{6}$
- 93 *4. p. 7, l. 12: The shape parameter is usually not used to characterize soot, but rather fractal dimension.*
- 94 **Response:** The sentence about soot particles has been removed.

Soot particles typically exist as porous agglomerates, however, they can transform to spheres over several hours by
 condensation of H2SO4 (Happonen et al., 2010; Pagels et al., 2009; Rissler et al., 2013, 2014).

- 97 5. p. 7, l. 15: Complications in estimating rho\_eff for coarse mode because of mixing state I think the same can be said for
  98 the accumulation mode.
- 99 **Response:** The sentence has been modified to:

00 'The complicated mixing state of urban aerosols introduces additional uncertainties in estimating the  $\rho_{eff}$  for both accumulation 01 and coarse mode particles'.

- 02 6. p. 7, l. 16: Avoid starting a sentence with a mathematical symbol. (This happens several times throughout the manuscript).
- 03 **Response:** This issue has been corrected for the manuscript.
- 04 7. p. 9, l. 72: Notation: Is log referring to log10 or natural log?

05 **Response:** 'logarithm base 10' has been added.

06 'The extracted measured data for these PSDs were fit to the multi-modal lognormal distribution function ( $dN/dLogD_p$ , cm-3,

Eq. 7; logarithm base 10) by using a lognormal fitting code in MATLAB based on the nonlinear least-squares curve fitting
 function'

8. p. 9, equation (7): Determining the number of modes to achieve best fit – how many modes are usually needed to achieve
best fit? (i.e. is it mostly 3 modes or are sometimes more needed?)

- 11 **Response:** For number PSDs, two or three modes can achieve the best fit.
- 12 'Two or three modes can achieve the best fit.'

23

- For the volume and mass PSDs described in Section 4.1.2 and 4.1.3, typically three modes can achieve the best fit, but sometimes it requires four modes.
- 15 'Typically three modes are enough to achieve the best fit, but sometimes four modes are needed.'
- 'For the volume and mass PSDs measured by inertial impactors, typically three modes can achieve the best fit, but sometimesthey need four modes.'
- 18 9. Figure 1: Include abbreviation of different regions from the legend in the figure caption.
- 19 **Response:** The name and abbreviation of geographical regions have been added to the figure caption.

Figure 10. Temporal and geographical distribution by year of the urban PSD references analyzed in this study (1998-2017).
The geographical regions include North America, Australia, and New Zealand (NAAN), Europe (EU), East Asia (EA), Central,
South, and Southeast Asia (CSSA), Latin America (LA), West Asia (WA), and Africa (AF).

10. Tables 1 & 2: Include sampling frequency for the various studies or any information if the size distribution is an instantaneous snapshot or the result of a time-average.

**Response:** Given the great uncertainty of instantaneous snapshots, the PSDs and size-resolved  $\rho_{eff}$  collected in this study are all time-average results over a given sampling period. They were all collected from peer-review journal articles in the form of figures or fitting parameters. The sampling duration for individual PSD has been listed in the SI.

- 29 The following two sentences have been added to Section 2 and Section 3.2:
- 'All the PSDs are the results of a time-average over certain sampling periods'; 'All the results are the averages over givensampling periods'.

*Figure 5: Considering that the Aitken mode is only incompletely captured, the fit is doing some extrapolation. Mathematically, this is fine, but given that there is some noise, the fitted size distribution parameters could be way off. How reliable is the fit in such a case?*

- 35 **Response:** The following texts were added to Section 4.2:
- 36 'The size range over which the fitting is effective is listed in each fitted PSD. To reproduce the urban aerosol PSDs by using
- 37 the fitting parameters, the size range of PSDs should not exceed the listed effective size range. The fitted size range was not necessarily equivalent to the size range of the measurement.'
- The effective size range was defined as the range where the fitted curve visually matches the extracted PSD data points. The effective size range can never exceed the original measurement size range, which avoids extrapolation.
- Figure 5 shows the measurement spanning from the accumulation and coarse modes. The listed effective size range for fitting is 660-10000 nm. Therefore it will not be extrapolated to Aitken or nucleation mode when being reproduced.
- 43 12. p. 17, l. 10: It would be helpful to include some references to papers where the issues mentioned here are considered and
  44 also give the reader some context about how important these issues might be.
- 45 **Response:** Several references have been added.
- 46 'Penetrated urban aerosol PSDs were determined for filters with the Minimum Efficiency Reporting Value (MERV) rating of 47 8 and 14. Size-resolved filtration efficiencies for the MERV 8 and 14 filters were estimated from Hecker and Hofacre (2008), 48 as illustrated in Fig. 9. The impact of aerosol hygroscopicity and electrostatic charge, filter ageing, filter bypass, and ventilation 49 airflow parameters (e.g. face velocity) on changes in the size-resolved filtration efficiency were not considered (Chang et al., 50 2016; Montgomery et al., 2015; Podgorski et al., 2006; Wang and Otani, 2013; Wang et al., 2006). The transformation of the 51 urban aerosol PSD due to penetration through the building envelope (e.g. infiltration) was not evaluated.'
- 52

13. Figure 17: Are the PSD assigned to UB and to TR significantly different (in a technical sense)? In other words, does the grouping into UB and TR make sense? Did you try to do a cluster analysis based on the PSD shapes to see what different types in terms of shape emerge (=size distribution regimes), and if they tend to include size distributions from the same geographical region? Given the many factors that influence the shape of size distributions (as mentioned on p. 23), I question if the a priori separation into geographical regions is the best way to look at the data.

- 58 Response: Although the difference of the PSDs between UB and TR sites is not the main focus of this paper, we feel that it is 59 worth it to discuss since traffic emission is an important pollution source in urban environments, and it is the main source of 60 UFPs in high-isolation developed world cities (Brines et al., 2015). The evolution of the PSDs of traffic-related emissions has 61 been discussed in many papers, and it is worthwhile to summarize in this section.
- We did not try to do any cluster or statistical analysis for the PSDs at UB and TR sites. Instead, we added two metrics to show the difference between the two sites: mean CMD and fraction of the particles < 20 nm in total number concentration. The following text has been modified:
- 65 'UB measurements are typically dominated by Aitken mode particles, with peaks ranging from Dem = 20 to 90 nm, with the 66 mean CMD of 45 nm. In contrast, many of the TR measurements exhibit prominent nucleation modes with peaks falling 67 below Dem = 30 nm, and in some cases, below Dem = 10 nm, with the mean CMD of 33 nm. The mean concentration fraction 68 of the particles smaller than 20 nm is 35% and 19% for the PSDs measured at TR and UB sites, respectively.'
- 69 We think the geographical separation provides a novel perspective to analyze the data. We would not like to conclude that the 70 geographical separation is the best way to look at the PSDs we collected, but we would like to present the results and differences
- 71 we found, and provide possible explanations.

14. Section 10 and 11: It is not clear to me what the take home message is of these sections. This needs to be more clearly
 formulated and presented. To better gauge the differences in particle exposure for different regions, I recommend introducing
 a couple of integrated metrics, e.g. number concentrations of particles smaller than 100 nm/1 micron etc.

**Response:** Size-integrated metrics on aerosol inhalation exposure and penetrated urban aerosol through an air filter have been added (Figures 24 & 27). The metrics include mean size-integrated  $RTDDR_N$  (10-100 nm) and  $RTDDR_M$  (100-2500 nm) in the respiratory tract and mean size-integrated number concentration (10-100 nm) of penetrated urban aerosol through a MERV filter in different geographical regions. The discussions were also added to the two sections:

79 'The size-integrated metrics were utilized to better illustrate the difference in aerosol inhalation exposure in different 80 geographical regions. The mean size-integrated (10-100 nm) RTDDRN (Fig. 24, left) shows the highest value in CSSA, 81 following by EA, EU, NAAN, and LA. The mean size-integrated (100-2500 nm)  $RTDDR_M$  (Fig. 24, right) also shows the 82 highest value in CSSA, which is twice higher than the second highest in EA and third highest in WA. This is attributable to 83 the abundant accumulation mode particles in CSSA due to strong air pollutions. EU and NAAN present much lower values 84 due to the better urban air quality.'

85

\*Fig. 27 shows the mean size-integrated number concentration (10-100 nm) of urban aerosol that penetrates through a MERV
8 (left) and MERV 14 (right) in different regions. This metric reflects the impact of geographical variations in urban PSDs on
indoor air quality. For both grades of filters, the poor urban air quality in CSSA results in the highest penetrated aerosol
concentration, following by EA, EU, NAAN, and LA.'

90

91 15. Data availability statement: Suggest posting an electronic version of the supplement to some databank (with doi) so that
92 it can be easily used by the community for future studies.

Response: The Supplement has been uploaded to the Purdue University Research Repository (PURR) DOI: 10.4231/MWM5 JQ30.

**95 Additional changes:**

1. The table of "List of the number of urban PSDs that have been extracted and analyzed in each region, country, and city" has
 been corrected.

| 98 Table 1. List of the number of urban PSDs that have been extracted and analyzed in each region, count | ry, and city. |
|----------------------------------------------------------------------------------------------------------|---------------|
|----------------------------------------------------------------------------------------------------------|---------------|

.99

| Region        | Country          | City                             |
|---------------|------------------|----------------------------------|
|               | Botswana (3)     | Gaborone (3)                     |
|               | Cape Verde (1)   | Praia City (1)                   |
|               | Egypt (12)       | Cairo (12)                       |
|               | Kenya (1)        | Nairobi (1)                      |
| Africa (n=29) | Mali (1)         | Bamako (1)                       |
|               | Senegal (1)      | Dakar (1)                        |
|               | South Africa (6) | Johannesburg (2)                 |
|               | South Africa (0) | Johannesburg (2)
Pretoria (4) |
|               | Zambia (4)       | Mongu (4)                        |
|               | India (48)       | Durg (4)                         |

|                      |                    | Kanpur (23)        |
|----------------------|--------------------|--------------------|
|                      |                    | Mumbai (2)         |
|                      |                    | New Delhi (13)     |
|                      |                    | Pune (2)           |
|                      |                    | Trivandrum (4)     |
|                      | Nepal (4)          | Lalitpur (1)       |
| Central, South,      |                    | Dhulikhel (3)      |
| and Southeast        | D           | Karachi (3)        |
| Asia ( n =70) |                    | Lahore (3)         |
|                      | rakistali (0)      | Peshawar (1)       |
|                      |                    | Rawalpindi (1)     |
|                      | Singapore (6)      | Singapore City (6) |
|                      | Theiland (2)       | Chiang Mai (1)     |
|                      | Thananu (2)        | Silpakorn (1)      |
|                      | Vietnam (2)        | Ho Chi Minh (2)    |
|                      |                    | Beijing (38)       |
|                      |                    | Guangzhou (14)     |
|                      |                    | Hong Kong (6)      |
|                      |                    | Jinan (1)          |
|                      |                    | Jinhua (1)         |
|                      |                    | Lanzhou (11)       |
|                      | China (95)         | Nanjing (2)        |
|                      |                    | Shanghai (12)      |
|                      |                    | Shenzhen (3)       |
| East Asia            |                    | Zhengzhou (2)      |
| ( n =138)     |                    | Urumchi (1)        |
|                      |                    | Wuxi (2)           |
|                      | Japan (17)         | Kawasaki (10)      |
|                      |                    | Sapporo (6)        |
|                      |                    | Tokyo (1)          |
|                      |                    | Gwangju (16)       |
|                      | South Korea (24)   | Seoul (4)          |
|                      |                    | Ulsan (4)          |
|                      | Chines Taimer (4)  | Taipei (3)         |
|                      | China. Taiwan (4)  | Taichung (1)       |
|                      | Austria (7)        | Vienna (7)         |
|                      | Belgium (1)        | Gent (1)           |
| Furane (n-216)       | Switzerland (4)    | Zurich (4)         |
| Europe (n-310)       | Czech Republic (4) | Prague (4)         |
|                      | Germany (68)       | Aachen (1)         |
|                      |                    | Augsburg (2)       |

|                         |                        | Braunschweig (12) |
|-------------------------|------------------------|-------------------|
|                         |                        | Dresden (4)       |
|                         |                        | Duisburg (1)      |
|                         |                        | Erfurt (9)        |
|                         |                        | Essen (5)         |
|                         |                        | Heidelberg (2)    |
|                         |                        | Karlsruhe (5)     |
|                         |                        | Leipzig (27)      |
|                         | Denmark (26)           | Copenhagen (25)   |
|                         | Definitiar (20)        | Odense (1)        |
|                         |                        | Barcelona (7)     |
|                         | Spain (17)             | Ciudad Real (2)   |
|                         |                        | Madrid (8)        |
|                         | Finland (52)           | Helsinki (52)     |
|                         |                        | Marseilles (5)    |
|                         | France (10)            | Paris (3)         |
|                         |                        | Dunkirk (2)       |
|                         |                        | Birmingham (1)    |
|                         |                        | Cambridge (7)     |
|                         | United Kingdom (27)    | Leeds (3)         |
|                         | Officed Kingdolfi (37) | Leicester (2)     |
|                         |                        | London (19)       |
|                         |                        | Manchester (5)    |
|                         | Graece (7)             | Athens (3)        |
|                         |                        | Chania (4)        |
|                         | Hungary (9)            | Budapest (9)      |
|                         |                        | Bologna (2)       |
|                         |                        | Cagliari (1)      |
|                         | Italy (31)             | Cassino (2)       |
|                         |                        | Ispra (4)         |
|                         |                        | Milan (15)        |
|                         |                        | Rome (7)          |
| Europe ( n =316) | Lithuania (9)          | Vilnius (9)       |
|                         | Netherland (6)         | Amsterdam (4)     |
|                         |                        | Rotterdam (2)     |
|                         | Norway (3)             | Oslo (3)          |
|                         | Portugal (12)          | Oporto (10)       |
|                         |                        | Lisbon (2)        |
|                         | Russia (1)             | Tiksi (1)         |
|                         | Sweden (11)            | Gothenburg (3)    |

|                       |                            | Stockholm (8)      |
|-----------------------|----------------------------|--------------------|
|                       | Moldova (1)                | Chisinau (1)       |
|                       | Drazil (28)                | Porto Alegre (3)   |
| Tatin America         | Brazii (38)                | São Paulo (35)     |
| Laun America $(n-48)$ | Chile (6)                  | Santiago (6)       |
| (n-40)                | Mexico (3)                 | Mexico City (3)    |
|                       | Cuba (1)                   | Camagüey (1)       |
|                       |                            | Brisbane (2)       |
| North America,        | Australia (10)             | Launceston (7)     |
| Australia and         |                            | Wollongong (1)     |
| New Zealand           | New Zealand (8)            | Auckland (8)       |
| ( n =134)      | Canada (15)                | Hamilton (2)       |
|                       | Callada (15)               | Toronto (13)       |
|                       |                            | Atlanta (4)        |
|                       |                            | Boulder (1)        |
|                       |                            | Buffalo (6)        |
|                       |                            | Claremont (5)      |
|                       |                            | Corpus Christi (4) |
|                       | United States (U.S.) (101) | Detroit (6)        |
| North America,        |                            | Downey (4)         |
| Australia and         |                            | Fresno (12)        |
| New Zealand           |                            | Houston (1)        |
| ( n =134)      |                            | Los Angeles (34)   |
|                       |                            | New York (5)       |
|                       |                            | Newark (2)         |
|                       |                            | Pittsburgh (7)     |
|                       |                            | Raleigh (1)        |
|                       |                            | Riverside (8)      |
|                       |                            | Rochester (1)      |
|                       | Iran (23)                  | Zanjan (23)        |
| West Asia             | Jordan (1)                 | NA                 |
| (n=58)                | Kuwait (12)                | Fahaheel (12)      |
| (11-30)               | Saudi Arabia (9)           | Yanbu (9)          |
|                       | Turkey (13)                | Istanbul (13)      |

02 2. A paragraph of discussion on the influence of environmental factors in different geographical locations on filter performance03 has been added to Section 10.

6 'In addition, different geographical locations may result in different environmental factors, which can influence the
 6 performance of a filter. For example, the dendrites of the deposited particles may experience more frequent collapse in

- 96 geographical regions with annual rainy seasons when the relative humidity increases significantly. The dendrites may 97 restructure to adopt a compact morphology under elevated humidity due to the capillary effect of condensed water, which will 98 lower the filtration efficiency and pressure drop across the filter (Pei et al., 2019). In addition, the hygroscopic content of the 99 dendrites may transition to form a liquid film on the filter fiber, and decrease the filtration efficiency before eventually reaching 91 an equilibrium (Li et al., 2014; Walsh et al., 1996). This is attributes to the low viscosity of the liquid film, which allows the 91 liquid particles to coalesce, relocate, and drain along the filter fiber.'
- 12
- 13 3. Errors in some figures have been corrected. Order of the figures has been rearranged.

---

## Author Response (AR2)

**Response to the Reviewers**

The authors wish to thank the three anonymous reviewers and the executive editor of *Atmospheric Chemistry and Physics* for thoroughly reviewing our manuscript and for providing very thoughtful suggestions. We have carefully reviewed each of the comments received and have rigorously revised our manuscript. This document presents a point-by-point response to our reviewers and includes all changes, corrections, additions, and subtractions. All comments from the reviewers are presented in italic black font. Our responses are presented in normal black font. The revised text is presented in normal blue font. These changes are also reflected within the revised manuscript's word document (main text and supplement) via track changes.

**Anonymous Referee #6**

*In their manuscript "Urban Aerosol Size Distributions: A Global Perspective", Wu and Boor use almost 800 particle size distributions from 125 cities around the globe, taken from the literature from approximately the last 20 years, to calculate typical number and mass particle size distributions for individual geographical regions. For this purpose, particle size distributions are extracted from figures or other information in the publications and fitted with 3-4 modal log-normal distributions. Size-resolved effective densities were compiled from information from the literature and used to convert particle number size distributions into mass distributions.*

*Using these distributions and size-dependent deposition efficiencies for the respiratory tract as well as size-dependent filtration efficiencies for two common types of filters of building ventilation systems, differences in ambient, respiratory tract-deposited and indoor particle size distributions for different regions around the globe were discussed.*

*In my eyes, the major effort of this work and the main contribution to the aerosol community is the consistent conversion of a very large number of particle size distribution measurement into a common format (i.e. fitted multi-modal log-normal distributions) that can be used for collective investigation of dependencies of PSDs on external influences. Here, the main focus was laid on the investigation of geographical differences of these particle size distributions.*

*I think this work can be published in Atmospheric Chemistry and Physics, however in a quite different format as it is currently attempted. In the current form, the manuscript is way to long for the type and extent of information which is conveyed. I do not think that this manuscript qualifies as a "critical review" as the authors state in their manuscript. According to ACPs rules a review article "summarize[s] the status of knowledge and outline[s] future directions of research ...". Even though, for this work, size distributions from many publications have been used, this manuscript does not provide an overview on the status of knowledge of, e.g. particle size distributions or their measurements. This manuscript rather describes a research project which uses data from the literature in order to obtain typical size distributions from different regions of the world.*

*Therefore, I suggest re-structuring the manuscript along the following lines: Overall, the manuscript can be shortened by at least 50% by avoiding repetitions or lengthy discussion of the literature for rather special side-effects and by moving detailed analysis information into the supplement. Suggestions for the individual sections are provided below. Furthermore, the manuscript should be focused more on the main findings of this work, instead of attempting to provide a broad coverage of various applications (i.e. respiratory tract deposition and filtration efficiency). The effect of differences in particle size distributions in general on such size-dependent effects can be shortly discussed in the manuscript, however, it should not cover such a large fraction of the whole paper and repetitive figures for each of these types of applications of size distributions are not necessary in the main paper. I think it is way more important to provide a thorough and critical discussion of the uncertainties and variabilities of the initial and the converted size distributions and how this affects*

*significance of differences between different geographical areas. Actually, with all the uncertainties and variabilities (see comments below for details), I doubt that the observed differences in size distributions for different geographical areas are significant. If this is the case, I do not see that it makes a lot of sense to discuss differences in respiratory tract deposition and filtration efficiencies (which have even larger uncertainty than the PSDs) for the different regions and to present multiple figures for this analysis which strongly resemble those presented in earlier sections.*

*With focusing the manuscript onto the main findings and a thorough consideration of uncertainties and variabilities, I think this manuscript provides a valuable contribution for the aerosol community and deserves publication in ACP.*

> **Response:** We feel that this study is closer to a critical review rather than an original research article. However, if the executive editor suggests to change the format to a research article, we are happy to follow their suggestion.
>
> Upon the reviewer's request, we have shortened the article by ~40% (30 pages). We removed the discussions on respiratory dose rates and building filtration, and focus more on the main findings on this work. The tables and numerous figures have been moved to the Supplement. A section regarding the uncertainties on the PSD extraction and lognormal fitting has been added. The variability and the limitations about the size-resolved urban aerosol $\rho_{eff}$ function were added.

*Detailed comments:*

*As mentioned by reviewer #1, the measurements are taken from a period of 20 years. Over this time, e.g. due to air quality measures, a change in particle size distributions at individual locations or areas could be expected. It would be interesting if the authors would use their unique data set to investigate this. Unfortunately, as a response to the reviewer's suggestion, they only included a general discussion on typical temporal variability of particle size distributions, but did not use their data to extract such information. As long as this kind of information is not used from the data, I do not see the need for an extensive discussion on temporal variations, beyond the point, what is needed to explain uncertainty and variability due to these effects. Unfortunately, the latter was not done, yet.*

> **Response:** Thank you for the comment regarding the temporal variations. It is an interesting topic, however, it is not the focus of this paper. There are difficulties in investigating temporal variations in PSDs. Long-term PSD data at the same sampling sites are generally not available for most of the cities included in this study. Since the paper mainly focus on the spatial variations, this section has been removed.
>
> A section discussing the uncertainties of the PSDs induced by data extraction and lognormal fitting has been added:
>
> **7 Uncertainties in the extraction and lognormal fitting of urban aerosol PSDs**
>
> The urban aerosol PSDs analyzed in this study are the extracted and fitted PSDs from previously reported measurements. The data extraction and lognormal fitting process introduced some uncertainties compared with the original data. Typically, the extraction process can obtain accurate data for the dominant peak of the PSD, where the concentrations are high. However, as the PSDs were primarily reported in the form of figures with a linear y-axis scale, relatively large uncertainties exist for the size ranges with low concentrations due to the limited resolution of the extraction process by using pixel picking in the figures. Similarly, the lognormal fitting process can accurately capture the dominant mode of the PSD, as the high concentrations are weighted greater in the nonlinear least-squares curve fitting (the fitting quality of individual PSDs can be visually checked in the Supplement). However, relatively large uncertainties exist

between the originally reported data and the fitted PSDs for the size ranges where the concentrations are relatively low compared with the dominant peak.

These uncertainties, to some extent, affect the comparison of the PSDs in regard to their magnitudes. For example, most of the number PSDs exhibit a dominant mode in the UFP regime, while the accumulation mode often appears as a tail of the dominant mode. Although the number PSDs are fitted well in the UFP regime, the fitted number PSDs in the accumulation mode exhibit relatively large uncertainties. Therefore, we mainly compared the dominant mode in the discussion of geographical variations in urban aerosol PSDs. The uncertainties are expected to affect the comparison of the normalized PSDs to a lesser extent. Since the magnitude of individual PSDs are normalized, only the shape of the main mode is emphasized and compared, while the size ranges with relatively large uncertainties become less important. Therefore, the uncertainties from the data extraction and lognormal fitting would not significantly influence the findings on the geographical variations of the urban aerosol PSDs. The uncertainties potentially affect the absolute concentrations estimated in Figure 8. Since the fitting quality of the main mode is typically good and the size range with relatively larger uncertainties do not contribute substantially to the total number concentrations, this influence may not be significant. The mass PSDs include measurements with gravimetric methods employing inertial impactors. These PSDs exhibit low size resolution due to the limited number of size bins of the impactors. The fitted mass PSDs have much greater uncertainties in their shape compared to the number PSDs since the fitted curves were interpolated among the limited data points.

This study constructed size-resolved urban aerosol $\rho_{eff}$ functions by using measured and assumed $\rho_{eff}$ in a step-wise manner. This resulted in some sharp changes at the borders of the assumed size ranges. Using the $\rho_{eff}$ functions in the PSD conversions caused some step-like artifacts. In reality, the $\rho_{eff}$ and converted PSDs should transition smoothly with particle size. With the construction method and the previously measured data detailed in the paper, readers can re-construct the $\rho_{eff}$ functions in an interpolated manner to obtain smooth functions.

*Abstract: L14-15: What do the authors want to say with "To provide guidance for the evolution of urban aerosol observations"?*

**Response:** The sentence has been rephrased as:

"To inform future measurements of urban aerosol observations,"

*L16-17: I think using the variable "n" repeatedly if it has no real meaning is rather confusing. It can be just omitted from the text in locations like "n=793 PDS observations ... in n=125 cities ... in n=51 countries."*

**Response:** The variable "n" has been removed.

*Section 1: Introduction*

*The whole introduction does not mention at all, how ambient aerosol particle size distributions are generated, i.e. which physical processes and particle generation processes generate the typical particle size distributions. In order to discuss differences of particle size distributions, e.g. at different locations, it would be good to lay a solid basis on how they are generated.*

**Response:** We agree with the reviewer that the introduction of the formation of ambient aerosol PSDs is important. A short paragraph has been added:

Of particular importance are measurements of urban aerosol particle size distributions (PSDs). The ambient aerosol PSD is the result of direct particle emissions, in-situ formation processes, atmospheric interactions between particles or between particles and gaseous compounds, and deposition processes.

Typically, nucleation mode particles (3 to ~20 nm) are freshly formed via the nucleation of gaseous molecules and ions (Brines et al., 2015; Charron and Harrison, 2003; Zhu et al., 2002a). Aitken (~20 to 100 nm) and accumulation (100 to 1000 nm) mode particles are often associated with primary emissions from combustion sources and condensation of secondary materials (Yue et al., 2009). Coarse mode particles (>1000 nm) generally result from mechanical processes, such as aerodynamic resuspension and abrasion. Nucleation mode particles can be removed relatively quickly via coagulation due to their high diffusivity (Hinds, 2012). They can also grow into the Aitken mode during new particle formation events (Cai et al., 2017; Xiao et al., 2015). Aitken mode particles may further form accumulation mode particles via coagulation and condensation. Accumulation mode particles can have a long lifetime due to their low gravitational settling velocity and slow coagulation rate among themselves. In the urban environment, nucleation and Aitken mode particles generally dominate number PSDs, due to the abundance of primary emission sources, such as power generation, traffic, and industrial activities. Their concentrations are high close to emission sources, while decreasing rapidly with distance from the source (Zhu et al., 2002a). Particle size can grow during transport by condensation of secondary materials. Coarse mode particles in the urban environment often contain road dust (Almeida et al., 2006), tire debris (Adachi and Tainosho, 2004; Rogge et al., 1993) and biological particles (e.g. pollen) (Saari et al., 2015). Due to their high gravitational settling velocities, their number concentrations can be 2-4 orders of magnitudes lower than other modes.

*L50-66: Many fields where particle size or particle size distribution plays an important role are mentioned here, e.g. transport, transformation, source apportionment, health effects and deposition in respiratory tract, climate effects. However, nothing is said about how particle size affects processes in all these fields. In order to motivate the importance of particle size distribution measurements, this would be important.*

**Response:** A short paragraph has been added.

Of particular importance are measurements of urban aerosol particle size distributions (PSDs). The ambient aerosol PSD is the result of direct particle emissions, in-situ formation processes, atmospheric interactions between particles or between particles and gaseous compounds, and deposition processes. Typically, nucleation mode particles (3 to ~20 nm) are freshly formed via the nucleation of gaseous molecules and ions (Brines et al., 2015; Charron and Harrison, 2003; Zhu et al., 2002a). Aitken (~20 to 100 nm) and accumulation (100 to 1000 nm) mode particles are often associated with primary emissions from combustion sources and condensation of secondary materials (Yue et al., 2009). Coarse mode particles (>1000 nm) generally result from mechanical processes, such as aerodynamic resuspension and abrasion. Nucleation mode particles can be removed relatively quickly via coagulation due to their high diffusivity (Hinds, 2012). They can also grow into the Aitken mode during new particle formation events (Cai et al., 2017; Xiao et al., 2015). Aitken mode particles may further form accumulation mode particles via coagulation and condensation. Accumulation mode particles can have a long lifetime due to their low gravitational settling velocity and slow coagulation rate among themselves. In the urban environment, nucleation and Aitken mode particles generally dominate number PSDs, due to the abundance of primary emission sources, such as power generation, traffic, and industrial activities. Their concentrations are high close to emission sources, while decreasing rapidly with distance from the source (Zhu et al., 2002a). Particle size can grow during transport by condensation of secondary materials. Coarse mode particles in the urban environment often contain road dust (Almeida et al., 2006), tire debris (Adachi and Tainosho, 2004; Rogge et al., 1993) and biological particles (e.g. pollen) (Saari et al., 2015). Due to their high gravitational settling velocities, their number concentrations can be 2-4 orders of magnitudes lower than other modes.

*L57: Do not use the term "aerosol" if the individual particle is meant. Rather use "particle" in such cases.*

**Response:** The term "aerosol" has been changed.

In addition, the measurement of aerosol PSDs is important for evaluating global climate change as particle size strongly affects the interaction of particles with solar radiation and their ability to form fog and cloud droplets (Mahowald, 2011; Seinfeld and Pandis, 2012; Zhang et al., 2012).

*Section 2: General methodology*

*L101: For coarse particles, mainly Aerodynamic Particle Sizer (APS) is mentioned as an instrument. Many measurements are based on OPS or OPC measurements, i.e. on optical particle size measurements. Why are these measurements not included here?*

**Response:** We did include many measurements made with an OPS or OPC (Table S5). The following sentence has been changed.

Most PSDs reported number-based concentrations (e.g. measured with a Scanning Mobility Particle Sizer (SMPS), Aerodynamic Particle Sizer (APS), or Optical Particle Sizer (OPS)), while some report mass-based concentrations (e.g. measured by inertial impactors).

*Section 3: Size-resolved urban aerosol effective densities*

*The authors develop size-resolved urban aerosol effective densities for three different groups, which are associated with three different geographical locations, over almost 7 pages. At the end of this effort, three different effective density functions for particles from 3 nm up to 10 µm electrical mobility diameter are available for the three groups of locations.*

*Since all three effective density functions are based on different sources for different particle size ranges, they all show substantial steps at the borders between these size ranges, which are physically not meaningful and not realistic and will affect the particle mass distributions, calculated with these functions. In addition, as also stated by the authors, the actual effective density values will strongly depend on e.g. location and environment of the measurement site (e.g. traffic, residential, industrial environment, city center, city outskirts, city at marine location, city near desert, etc.), meteorological conditions (air mass history – e.g. dust advection events, ambient temperature, solar radiation, precipitation, etc.), and time (time of the day, e.g. rush hour times, weekday – weekend, season). All this uncertainty comes on top of the uncertainty of the measurements of effective densities (which is not provided) and assumptions made to calculate the three effective density functions. With this massive overall uncertainty, I doubt that the differences between the density functions of the three groups are significant and therefore they might just introduce additional ambiguity to the calculated mass distributions. In addition, the only differences between these three effective density functions are based on a total of nine measurements. This is not a representative data base for typical effective density functions in these three area groups, even not for the small fraction of the particle size range, covered with these measurements.*

*With all these limitations and uncertainties, I do not think it makes sense to calculate more than a single effective density function, which is then applied to all data sets to convert number size distributions to mass size distributions. Furthermore, I do not feel that the determination of such a single effective density function, which should not include large steps between different size ranges, should be more than a small part of the methodology section of this paper. 90-95% of this text with all the details on how the effective density function was determined should be moved into the supplement.*

**Response:** Thank you for your detailed comments on the size-resolved aerosol effective density. We feel this section is one of the important outcomes of this study and we would like to keep this section in the main text. The need to convert urban number PSDs to mass PSDs will always exist, and measured data

on the size-resolved effective density has not yet been summarized, to our knowledge, in prior studies. Even with the uncertainties mentioned, the constructed effective density functions are still useful. The construction of these effective density functions is for an informative purpose. We do realize that there is a limited number of direct measurements and it is an important knowledge gap that we emphasized in the paper.

*Atmospheric Chemistry and Physics* has broad readers. However, not all of them are familiar with the topic of effective density. The method provides basic knowledge to understand the aerosol effective density parameter and the compiled functions. We believe it is important for the readers to follow the steps of how we construct the effective density functions, so the readers can adjust them easily according to their own use. Therefore, we would like to keep the method of constructing the effective density functions in the main text.

We acknowledge the dynamic features of the urban aerosol effective density. Discussions have been added regarding the limitations of using the compiled effective density functions for PSD conversions. However, to reduce the uncertainties, more direct measurements are needed. We agree that the sharp change at the borders are the artifacts resulting from the construction method. However, with the construction method and extracted data detailed in the paper, the readers can modify the function by simply switching to an interpolated method rather than using a step-wise function, which we used in this paper, if a smooth function is needed.

We think that separating the data into three groups is reasonable. The difference between the effective density measured in a typical urban background environment and a traffic-influenced environment has been discussed in previous studies. The comparisons between the urban number PSDs in EA, NAAN, and EU indicates potential differences in emission sources. Therefore, we would like to apply the effective density accordingly when doing the PSD conversions.

There are limitations of the effective density function. We still feel they are very informative and useful for the aerosol community, although the available data compiled here are limited. In the textbook, "ATMOSPHERIC CHEMISTRY AND PHYSICS: From Air Pollution to Climate Change" (Seinfeld and Pandis, 2016), the number PSDs from one single study (Hussein et al., 2004) have been used as the "representative" urban aerosol number PSD to compare with PSDs from other environments, while the authors acknowledge the strong dependence of PSDs on many factors. Therefore, we do not think the analyses of the size-resolved urban aerosol effective density functions is inappropriate.

The constructed size-resolved urban aerosol $\rho_{eff}$ functions represent the first attempt to compile previous measured data and extend those to unmeasured size ranges to obtain a continuous function that can be applied in PSD conversions when direct measurements of the size-resolved $\rho_{eff}$ are unavailable. With the method and measured data detailed in this paper, one can adjust or re-construct the $\rho_{eff}$ functions according to their own use. It is acknowledged that the number of compiled data is limited, and uncertainties could be significant for size ranges where direct measurements are unavailable. To reduce uncertainties, more direct measurements are needed in the future. The effective density can be affected by different atmospheric processes and is often site-specific and temporally variant. Caution should be taken when applying the $\rho_{eff}$ functions. In addition, we compiled the data in a step-wise manner, which resulted in sudden changes at the border of certain size ranges. Other interpolation methods can be used to smooth the $\rho_{eff}$ functions.

*L119: I suggest using the more commonly symbol "Dmob" for the electrical mobility diameter throughout the manuscript.*

**Response:** We feel $D_{em}$ can be understood by the readers. The symbol is adopted from the textbook "ATMOSPHERIC CHEMISTRY AND PHYSICS: From Air Pollution to Climate Change".

*L173: Why is Figure 2 mentioned in the text before Figure 1?*

**Response:** The order has been changed.

*L199-215: The whole determination of effective density for accumulation mode particles without direct measurements seems to be not more than a very rough estimate with all the unknowns mentioned in this text. E.g. assuming a constant r_eff of 1.65 g cm-3 for particles in a size range from 400 to 2500 nm for group B does not make sense. The lower particle size fraction of this size range consists of accumulation mode particles which have completely different generation processes and consequently chemical composition than the larger particle size fraction. Consequently, there must be a (smooth) transition in this size range from an accumulation mode density to a coarse mode density.*

**Response:** Thank you for your comments. Due to the lack of size-resolved effective density measurements, assuming a constant effective density in this size range does induce uncertainties. However, there is at least a direct measurement made at a urban background site, which may result in less uncertainty than constructing a size-resolved effective density function solely based on the assumptions of the chemical composition.

The assumed effective density of the unmeasured size range does exhibit significant uncertainty. A short paragraph about this has been added:

The constructed size-resolved urban aerosol $\rho_{eff}$ functions represent the first attempt to compile previous measured data and extend those to unmeasured size ranges to obtain a continuous function that can be applied in PSD conversions when direct measurements of the size-resolved $\rho_{eff}$ are unavailable. With the method and measured data detailed in this paper, one can adjust or re-construct the $\rho_{eff}$ functions according to their own use. It is acknowledged that the number of compiled data is limited, and uncertainties could be significant for size ranges where direct measurements are unavailable. To reduce uncertainties, more direct measurements are needed in the future. The effective density can be affected by different atmospheric processes and is often site-specific and temporally variant. Caution should be taken when applying the $\rho_{eff}$ functions. In addition, we compiled the data in a step-wise manner, which resulted in sudden changes at the border of certain size ranges. Other interpolation methods can be used to smooth the $\rho_{eff}$ functions.

*L216-248: Similar comment as the last one. Also, the estimate for the density of coarse mode particles without direct measurements is very uncertain and contains several inconsistencies. The discussion, which includes several densities (SOA, ammonium sulfate, ammonium nitrate), which are rather associated with the accumulation mode, shows that there are again large differences in material densities and consequently large uncertainties in the resulting effective density. Furthermore, it is not clear why this estimate of coarse mode effective density starts at different cut-off diameters for all three groups.*

**Response:** A paragraph about the uncertainties and the limitation of the effective density functions has been added. Indeed the secondary organic and inorganic materials are typically associated with the accumulation mode. But they are also found in some coarse particles (e.g. Hu et al., 2012).

The estimate of coarse mode effective density does start at different cut-off diameters. In general, we would like to start the estimate from 1 μm as the coarse mode is defined from 1 μm in this paper. However, we utilized the direct measured data from China up to the size of 3.2 μm (higher size bins were disregarded as the authors claimed unusually high fraction of mineral dust at the study site (Hu et al., 2012)).

Therefore, the estimate in Group A starts at 3.2 µm.  Group B adopted the apparent density of $PM_{2.5}$ measured at a German urban site.   Thus, the estimate for the unmeasured coarse mode starts at 2.5 µm.

*L271-277 and L285-292: Large parts of this text is a repetition of the text in the previous sections.*

    **Response:**  Most sentences in these two paragraphs have been deleted.

*Section 4: Methodology for analyzing urban aerosol PSD observations*

*In this section the authors describe how they treat the data of particle size distributions from the literature and how they extract or calculate number, surface, volume and mass size distributions from the various types of available data. Typically, number size distribution data were taken from figures in the respective manuscripts. These were fitted using 3-4 modal log-normal fits and then converted into surface, volume and mass size distributions. Mass distributions were typically calculated using the effective density data from section 3.*

*L343-345: If parameters of log-normal fits to the data were already available in a few publications, these were used for further analysis and not the general approach was performed. This makes sense, however, not applying the typical approach of extracting the size distribution data from the figures and fitting them with log-normal distributions means that this chance to validate the own approach and algorithm is not made use of. This would have provided some idea on how well this works. Especially for the very low particle number concentrations for larger particle sizes (which were typically very close to zero in linear plots), I would expect large relative uncertainties in the derived values. This could have easily been assessed, using those data where both, figures and fitted log-normal distributions were available.*

    **Response:** Thank you to the reviewer for bringing this up.  The lognormal fitting and the extraction process do induce uncertainties in the re-produced urban PSDs.  The uncertainties may not be important for the dominant mode, where the concentrations are high. But they might be non-negligible for the size range where the concentrations are close to zero.  We added a section discussing the uncertainties in the re-produced PSDs.  Due to the relatively small uncertainties in the dominant mode of the PSDs, we mainly compared the dominant mode in the discussion of geographical variations.

**7 Uncertainties in the extraction and lognormal fitting of urban aerosol PSDs**

The urban aerosol PSDs analyzed in this study are the extracted and fitted PSDs from previously reported measurements.  The data extraction and lognormal fitting process introduced some uncertainties compared with the original data.  Typically, the extraction process can obtain accurate data for the dominant peak of the PSD, where the concentrations are high.  However, as the PSDs were primarily reported in the form of figures with a linear y-axis scale, relatively large uncertainties exist for the size ranges with low concentrations due to the limited resolution of the extraction process by using pixel picking in the figures.  Similarly, the lognormal fitting process can accurately capture the dominant mode of the PSD, as the high concentrations are weighted greater in the nonlinear least-squares curve fitting (the fitting quality of individual PSDs can be visually checked in the Supplement).  However, relatively large uncertainties exist between the originally reported data and the fitted PSDs for the size ranges where the concentrations are relatively low compared with the dominant peak.

These uncertainties, to some extent, affect the comparison of the PSDs in regard to their magnitudes.  For example, most of the number PSDs exhibit a dominant mode in the UFP regime, while the accumulation mode often appears as a tail of the dominant mode.  Although the number PSDs are fitted well in the UFP regime, the fitted number PSDs in the accumulation mode exhibit relatively large uncertainties.  Therefore, we mainly compared the dominant mode in the discussion of geographical variations in urban aerosol PSDs.  The uncertainties are expected to affect the comparison of the normalized PSDs to a lesser extent.  Since the magnitude of individual PSDs are normalized, only the shape of the main mode is emphasized and compared, while the size ranges with relatively large uncertainties become less important.  Therefore,

the uncertainties from the data extraction and lognormal fitting would not significantly influence the findings on the geographical variations of the urban aerosol PSDs. The uncertainties potentially affect the absolute concentrations estimated in Figure 8. Since the fitting quality of the main mode is typically good and the size range with relatively larger uncertainties do not contribute substantially to the total number concentrations, this influence may not be significant. The mass PSDs include measurements with gravimetric methods employing inertial impactors. These PSDs exhibit low size resolution due to the limited number of size bins of the impactors. The fitted mass PSDs have much greater uncertainties in their shape compared to the number PSDs since the fitted curves were interpolated among the limited data points.

This study constructed size-resolved urban aerosol $\rho_{eff}$ functions by using measured and assumed $\rho_{eff}$ in a step-wise manner. This resulted in some sharp changes at the borders of the assumed size ranges. Using the $\rho_{eff}$ functions in the PSD conversions caused some step-like artifacts. In reality, the $\rho_{eff}$ and converted PSDs should transition smoothly with particle size. With the construction method and the previously measured data detailed in the paper, readers can re-construct the $\rho_{eff}$ functions in an interpolated manner to obtain smooth functions.

Both the extraction and fitting processes induce uncertainties. To quantify the uncertainties in each re-produced PSD requires us to obtain the originally measured data. However, it is impractical to contact the authors of each paper to obtain their measured data. In addition, quantifying the uncertainties of the lognormal fitting is uncommon in the literature. Highly cited papers (e.g. Hussein et al., 2004) published in *Atmospheric Chemistry & Physics* reported the lognormal fitting parameters without indicating the uncertainties of their fitting. We present each extracted and fitted PSD in the Supplement and the fitting quality can be easily checked visually by the readers.

*L359-361: The fitted number PSDs were converted to surface area and volume PSDs assuming spherical particles and converted to mass PSDs using the size-resolved effective density functions from section 3. In section 3 large effort was made to estimate effective densities and particle shape for three different geographical regions. Here, the simple assumption of spherical particles is made. In addition, the effective density functions from section 3 have large uncertainties und partially huge steps at the borders of the individual size ranges (see comment above). These introduce steps in calculated mass distributions (see e.g. Fig. 3 lower right panel). What is the uncertainty introduced into the resulting size distributions by these effects? What is the uncertainty due to the extraction of the size distribution data from the printed figures – especially for the larger particles (with very low number concentrations) and converted surface, volume, and mass distributions? What is the resulting uncertainty in the size-integrated concentrations? How well does the integration of the fitted log-normal distributions agree with the integration of the raw data?*

**Response:** The morphological features are already considered in the effective density. $\rho_{eff}$ is defined as the ratio of the measured particle mass ($m_p$) to the volume calculated from $D_{em}$ assuming spheres (DeCarlo et al., 2004; Hu et al., 2012; McMurry et al., 2002; Qiao et al., 2018).

A section discussing the uncertainties has been added. The variability and limitations on using the constructed effective density function have been discussed.

Due to the lack of raw data, we cannot quantify how well the integration of the fitted log-normal distributions agree with the integration of the raw data. It is impractical to contact the authors of each paper to obtain the raw data.

[revised manuscript text omitted]

*L366-381: Here, for the first time also optical size distribution measurements are mentioned as source for coarse mode size distribution data. However, unlike for the aerodynamic measurements, where several pages of methodology development were presented to convert d_aero into d_mob, here, nothing is stated about how*

*optical particle diameters were converted into electrical mobility diameters. Even more, in L430-432 it is stated that optical particle diameters were assumed to be equivalent to electrical mobility diameters – due to the lack of information needed to convert one to the other. This has to result in wrong electrical mobility diameters. Depending on particle size – and thus particle composition – the differences between optical and electrical mobility diameters are very different. It is not clear to me, why for the development of effective densities and conversion of aerodynamic diameters into electrical mobility diameters a lot of effort was made and many uncertainties were accepted (including uncertainties due to generalized assumptions for multiple measurements), while for conversion of optical diameters no such efforts (e.g. assumption of chemical composition and thus refractive indices; assumption of measurement geometry and laser wavelength) were made. Also it is not clear why here a uniform particle density (for all particle sizes) was applied while for the aerodynamic diameter conversion and calculation of mass distributions a size-dependent density was used. This very different treatment of similar features of the size distributions does not make sense to me.*

> **Response:** Thanks to the reviewer for this comment. During the early stages of this study, we did consider correlating the assumed chemical composition and particle shape with the refractive indices. However, as the settings of the OPS were not stated in most of the collected studies, we cannot confidently come up with a method to convert the optical diameter to electrical mobility diameter.

*L403: The dependency of D_a/D_em on shape and particle density is interesting for the details of the analysis, but not really a result that needs to be presented in the main paper. Move this Figure to the Supplement.*

> **Response:** This figure has been moved to the Supplement.

*L446-454: What is the overall uncertainty of this approach to derive the log-normal size distributions from the gravimetric measurements? E.g. what kind of uncertainty does the limited size resolution of the measurement data introduce?*

> **Response:** The limited number of size bins in the gravimetric measurements induce greater uncertainties in the shape of the re-produced PSDs compared to the number PSDs (e.g. measured with SMPS or APS). A discussion has been added to the section on PSD uncertainties.
>
> The mass PSDs include measurements with gravimetric methods employing inertial impactors. These PSDs exhibit low size resolution due to the limited number of size bins of the impactors. The fitted mass PSDs have much greater uncertainties in their shape compared to the number PSDs since the fitted curves were interpolated among the limited data points.

*L455-462: How is particle diameter defined in the columnar volume PSD measurements? Also, for this type of data the conversion is very simple (e.g. using a single particle density for all particle sizes). How are these PSDs comparable to those determined with the other methods?*

> **Response:** We feel that the columnar volume PSDs are not so relevant to the analysis. The columnar volume PSDs have been removed from this paper.

*L480-488: The comparison of PM2.5 calculated from several size distributions with locally measured PM2.5 concentrations is used to validate the presented methods. Also, here, no uncertainty is given for the calculated PM2.5 concentrations from the size distribution data. The error bars, provided for the local measurements of PM2.5, were calculated as the standard deviation of the measured concentrations over the same time intervals as for the size distribution measurements. If just the averages are compared, the standard deviation of the average should be used as uncertainty, not the standard deviation of the individual data points. Furthermore, as long as no information is provided on the comparability of the different measurement sites (e.g. are both next to each other or apart from each other, do they represent traffic or other local sources in a similar way, etc.) large differences between measured concentrations are expected and the comparison is not very valuable*

*to validate the methodology. As a result, PM2.5 concentrations differ in Fig. 7 by approximately 10 – 80%. Is this the uncertainty of the method?*

> **Response:** The uncertainty of the local measurement of $PM_{2.5}$ was calculated as the standard deviation of the average. A clarification has been made.

> The error bars represent the standard deviations of the mean $PM_{2.5}$.

> Although we chose the closest $PM_{2.5}$ monitoring stations we could find to the PSD sampling sites, they are not next to each other, but apart some distance. We do realize that this creates a difference in the aerosol population being measured. However, currently, this is the best method we can use to validate the methodology. Due to the lack of the originally reported data, we cannot quantify the uncertainties of the integration of the fitted lognormal distributions ($PM_{2.5}$).

*L511-545: Large parts of this text (on Figures 3, 5, 6) is a repetition of what was presented earlier in Section 4 and therefore could be omitted. Generally, it is not very clear to me what kind of new information is presented in Section 4.2.*

> **Response:** Section 4.2 has been moved to the Supplement.

*Section 5: Determining size-resolved urban aerosol respiratory tract deposited dose rates*

*Also, here, a large number of assumptions is made (e.g. tidal volume, breathing frequency, validity of deposition function, which depends on breathing routes and other factors) for calculation of the respiratory tract deposited dose rates. In addition, hygroscopic growth of particles, which depends on particle size and chemical composition is not considered. All this introduces an unknown overall uncertainty to the resulting RTDDRs, which is not even discussed. With this uncertainty together with the uncertainty of the size distributions, it is questionable, whether significant differences for different regions can be determined. In addition, adding this type of analysis to the manuscript, broadens the focus of the manuscript without adding significant results. I suggest removing the whole discussion of respiratory tract deposition from the paper and to sharpen the focus of the manuscript or at least to massively shorten this discussion and to move large parts of the methodology description into the supplement. If differences in RTDDRs between different regions are not significant, this "application" of PSDs could be shortened to a single short paragraph that presents the general effect of PSDs on respiratory tract deposition.*

> **Response:** The entire section on the inhaled deposited dose rate has been removed to maintain focus in the paper.

*Section 6: Evaluation of urban aerosol PSD that penetrates through a building ventilation system filter*

*This approach is another attempt to apply the worldwide urban PSDs to calculate secondary outcomes, in this case the penetration through filters of ventilation systems. Again, massive assumptions are made like similar ventilation and filtration systems in buildings worldwide with similar types of filters. Different fractions of time spent indoors worldwide, different tightness of buildings worldwide, different fractions of buildings equipped with ventilation systems worldwide, the impact of aerosol hygroscopicity, charge, filter age, ventilation system geometry, and ventilation rate were also not considered, nor the resulting uncertainties discussed or estimated. With all these assumptions and uncertainties, I doubt that significant differences in global penetration into buildings can be determined from the data. Therefore, I do not think it makes sense to broaden the focus of the manuscript by including this topic. At least, also this application of particle size distributions could be massively shortened to a short paragraph, which discussed the general influence of particle size on filtration and transport into buildings. In any case, Figures that just support the analysis like Fig. 9, can easily be moved into the supplement.*

**Response:** The entire section of urban aerosol filtration by building ventilation systems has been removed to maintain focus in the paper.

*Section 7: Overview of existing data on urban aerosol PSDs around the globe*

*This section repeats a lot of information which was already presented earlier. New information is given in the form of long lists of facts (number of publications per year, number of measurements per city or country) which also could be summarized in tables in the Supplement. Little general conclusions about temporal evolution and spatial distribution of measurements is provided beyond the listing of the individual data. It would safe a lot of space of the manuscript if all these lists were presented in the Supplement and if some general observations about temporal and spatial distribution of the measurements would be provided in a single paragraph or very short section. This could be complemented with a single Figure, in which the main content of Fig. 10 and Fig. 11 are summarized.*

**Response:** The content in this section has been trimmed by 50%.

*Section 8.1: Geographical variations in magnitude and shape of sub-micron aerosol number PSDs*

*Here, absolute and normalized (to the maximum) sub-micron number PSDs are shown for all measurements and summarized for individual regions. Very large variability is found between the individual PSDs both, for the global summary as well as for the individual regions. With an approximate variability of a factor 5-10 between the various PSDs of individual regions, the uncertainty of the typical PSD for each region is larger than the differences between those of the various regions (20-30%). The large variability within the individual regions is not surprising due to temporal variations at each individual measurement site (e.g. diurnal pattern of emissions, differences due to meteorology and air mass transport – dust pollution events, etc.) and variations between individual measurement locations (e.g. city size, environment around the city – ocean, desert, ..., land use differences – traffic, industry, residential). Thus, while there are tendencies (I would not call this "trends" as done in L692), the differences are not really significant. Therefore, it would be good to indicate the uncertainty range of the individual PSDs in Figure 13.*

**Response:** The discussion on the uncertainties of the re-produced PSDs have been added to Section 7. The term "trends" has been changed to "tendencies".

*Figure 14 and Figure 15 contain more or less the same information (i.e. the location of the mode diameter for each measurement). Since Figure 15 also includes information on absolute concentrations, I suggest to move Fig. 14 into the supplement and to present Fig. 15 only.*

**Response:** Figure 14 has been moved to the Supplement.

*I think it is not necessary to state that there is no clear inverse relationship visible between total number concentration and mode diameter in Fig. 15. I think the plot shows that with increasing mode diameter, number concentrations tend to decreases. This could be supported and quantified with a correlation analysis.*

**Response:** The sentence has been removed. The correlation analysis indicates a very weak negative correlation between the two (Spearman's Rho = -0.11).

*It would be good if it would be attempted to explain the results found in this analysis and not just reporting them. Are there any explanations for the observed behavior of the size distribution differences across the globe?*

**Response:** We tried to explain the geographical variations in the PSDs in Section 6.4.

*Section 8.2: Geographical variations of mass PSDs*

*Here, a similar analysis as in section 8.1, but with mass distributions instead of number distributions is performed. The smaller number of available measurements and the even larger variability (possibly due to the large uncertainties introduced in the conversion) does not allow separation of typical features of different locations.*

*Here, the authors describe the observations and mainly present information about mass size distributions, which is common textbook knowledge. Unfortunately, no real conclusion about e.g. observed differences at different locations or under different conditions (season, meteorology, measurement environment, etc.) is drawn, nor are differences which were observed tried to be explained (with the exception of dust contributions in a few areas).*

> **Response:** Indeed, due to the less number of mass PSDs that cover both the coarse and submicron regimes, we did not find obvious geographical tendencies in the mass PSDs, except in West Asia. Therefore, the length of the discussions on mass PSDs is shorter.

*Section 8.3: Intra-city variations of number PSDs between urban background and traffic sites*

*Here, general differences in number PSDs were discussed, observed between traffic sites and urban background sites. Often, smaller mode diameters were found for the traffic sites, compared to the urban background sites. This can be explained with the small diameter of traffic exhaust particles and their growth during aging. While this section provides interesting general information, it is partially unnecessarily lengthy. The first paragraph is partially repeating itself. The last two paragraphs provide a rather lengthy discussion to present evidence that traffic is a source for nucleation mode particles, which is mostly common knowledge. By streamlining this section, it could easily be shortened by 40-50%. I also do not think it is necessary to present Fig. 19 in the main text. This Figure only presents a summary of other measurements which is used to support the statement that traffic emits nucleation mode particles. If at all, this Figure could be shown in the supplement.*

> **Response:** We have shortened this section by ~50%. Figure 19 has been moved to the Supplement.

*L829, 830, 831: I would not call the maxima of the size distributions "peaks" as done here. Either name them "maxima" or "mode diameters".*

> **Response:** The term "peak" has been changed to "maximum" or "mode diameter".

*L837: Long-range transport does not occur within cities. This is transport over much larger spatial scales.*

> **Response:** The word "long-range" has been removed.

*Section 8.5: Temporal variations of urban PSDs*

*This section presents a large number of references which have discussed temporal trends of number size distributions and particle as well as gas phase concentrations on different time scales from hours to years. Apparently, this section was generated upon a request of previous reviewers who asked for an analysis of temporal trends in the size distribution data from all over the globe.*

*Unfortunately, no information about changes in particle sizes or changes in concentrations of particles of certain sizes, taken from the extensive data set of the authors, is provided in this section. A long list of references is provided which dealt with size distribution changes, however no information about their results or general features is given. In the second paragraph, information on temporal changes in concentrations of particles or trace gases is provided, however, again with no relationship towards size distributions. The PSD data which were investigated by the authors for geographical differences were not analyzed at all for potential temporal variations. This is a pity and was certainly not the idea of the reviewers who asked for information on temporal trends.*

*I also think it would be interesting to analyze the vast amount of size distribution data from all over the globe and from 20 years of measurements for potential temporal trends. The way temporal trends are presented in the current version of the manuscript is not very helpful in this respect and therefore could also be removed from a more focused manuscript, which, however then would not present any of the temporal information.*

> **Response:** Thank you for the comment regarding the temporal variations. It is an interesting topic, however, it is not the focus of this paper. There are difficulties in investigating temporal variations in PSDs. Long-term data at the same sampling sites are generally not available for most of the cities included in this study.
>
> Since the paper is primarily focused on spatial variations, this section has been removed.

*Section 9.1 and 9.2: Geographical variations in size-resolved urban aerosol number and mass respiratory tract deposited dose rates*

*The analysis from above, where geographical variations of number and mass PSDs are investigated, is extended here, by multiplying these number PSDs with size-resolved respiratory tract deposition functions. Similar graphs as in the analysis above are presented and some of the differences are discussed.*

*As the authors state, there are huge variabilities of particle number and mass concentrations at individual particle sizes (typically ~2 orders of magnitude). These uncertainties are further increased by multiplication with an uncertain respiratory tract deposition function, where many variables are either assumed or physical effects were neglected. Due to these very large uncertainties, which are not presented in the mean dose rate functions in Figures 21 and 23, I have strong doubts that the differences between different regions, which were discussed in the text, are significant. On the other hand, most of the remaining fractions of the text either repeat previous description of the deposition functions or present rather common knowledge.*

*Overall, I do not see which robust new information is presented in this section and I think it could be reduced to a short paragraph which discusses the general influence of particle size distribution on respiratory tract deposition.*

> **Response:** The entire section has been removed.

*Section 10: Implications for indoor air quality and aerosol filtration in building ventilation systems*

*Here, generally the same comments as for section 9.1 and 9.2 apply. Also, in this section, the highly uncertain median particle size distributions from previous analysis are multiplied with a particle size-dependent function (here, filtration efficiency for two types of filters) to obtain results (here, number size distributions and number concentrations that penetrate into the buildings). While the filtration efficiencies of the two types of filters are probably well known, the actual penetration of ambient particles into buildings depends on many factors. Particles penetrate into buildings through cracks and openings in the buildings, through air exchange systems, through open windows and doors and through multiple different types of ventilation systems. Globally, large differences in the fraction of buildings that is equipped with different kinds of ventilation can be expected. Therefore, multiplication of the particles size distributions with filtration efficiency curves of a certain filter type introduces large additional uncertainty to the indoor particle number distribution calculation, in addition to the already huge uncertainty in median particle size distributions.*

*Also, here, I doubt that any of the observed geographical differences in calculated penetrated particles into the buildings are significant (see, e.g. Figs. 27 and 28: the error bars only show the standard deviation of the underlying size distributions, not those uncertainties due to differences in air exchange technology and still the differences are not significant) or that the resulting indoor concentrations are on a solid basis. Potential geographical differences in the type of air exchange (filtration systems, open windows, etc.) typically applied*

*in different regions of the world are not even considered. With all this taken into account, I wonder what the robust information is, that is provided in Section 10. Also, the long discussion on potential changes in filtration efficiencies in filtration systems does not add substantially in this regard.*

> **Response:** The original purpose of the discussion of this section is to demonstrate how differences in outdoor PSDs may affect the filtration process, since the interactions between filter fibers and particles are size-dependent. As the reviewer suggested to focus on the main findings of this work, instead of attempting to provide a broad coverage of various applications, this section has been removed.

*Section 11: Framing future research directions for urban aerosol PSDs*

*I disagree with the authors that this manuscript is a critical review (L1137) that provides a comprehensive overview of urban aerosol number and mass PSD observations. Types of PSD measurements like, e.g. using aerosol mass spectrometric techniques, have not been covered or even mentioned at all. While a review provides an overview over the state of the art in a certain field, here only a large number of available PSD measurements was summarized in order to obtain average or median size distributions for different areas across the globe.*

> **Response:** The sentence has been removed.

*I also disagree with the statement (L1145) that this approach shows significant geographical variations in PSDs. As mentioned above, the variability within each group of PSDs and the uncertainty due to the applied method to extract the PSDs and to convert between number and mass distributions are at least an order of magnitude larger than the differences between the geographical locations. Unfortunately, no real treatment of uncertainty and variability was done in this manuscript.*

*The statement (L1170) that this compilation of urban aerosol PSDs demonstrates the need for a transition from size-integrated mass concentration measurements to PSD measurements is not supported by the findings of the authors. Of course, PSD measurements provide more information (but potentially also more uncertainty) than PMx measurements, however I do not see how the results presented by the authors demonstrate the "need" for PSD measurements.*

> **Response:** The sentence has been modified:
>
> The compilation of urban aerosol PSD observations in this review demonstrates the benefit of routinely measuring urban PSDs that include the nucleation, Aitken, accumulation, and coarse modes.

*I also do not understand how a transition from PM2.5 measurement to urban aerosol PSD monitoring can help support new legislation for diesel engines (L1186). There is no direct connection between emission of ultrafine particles at the tailpipe and measured ambient concentrations; furthermore, there are many other sources for UFP in urban environments.*

> **Response:** The sentences have been removed.

*The additional suggestion that such PSD measurements should span the entire UFP regime, including sub-3m nanocluster aerosol (L1190) seems very unrealistic. Measurement of such small particles or clusters is extremely hard to perform and cost intensive and will introduce large additional uncertainty. This is a field, which should be covered by research efforts and not by routine monitoring.*

> **Response:** The sentences have been removed.

*The last paragraph of this section (L1201-1215) seems to fit better into an introduction or a motivation than into this final section.*

**Response:** This paragraph has been moved to the introduction.

*Tables:*

*I think Tables 1 and 2 could easily be moved into a supplement. They do not contain important information that is needed to understand the paper, but rather additional details*

**Response:** Tables 1 and 2 have been moved to the Supplement.

**Anonymous Referee #5**

*I thank the authors for incorporating reviewer comments and providing a comprehensive "response to reviewer comments" document. Some minor comments below:*

*1. When introducing nucleation, Aitken, and accumulation modes, it might be helpful to specify the diameter ranges of each. While these distinctions are based on the formation process, e.g. the explanation for high accumulation mode contributions in EA and CSSA as direct primary emissions suggests a diameter-based cutoff in practice. (e.g. Yue et al. 2009 doi:10.1029/2008JD010894)*

>    **Response:** Thank you for this comment. We specified the size ranges of different modes. We also added the recommended reference.

>    Of particular importance are measurements of urban aerosol particle size distributions (PSDs). The ambient aerosol PSD is the result of direct particle emissions, in-situ formation processes, atmospheric interactions between particles or between particles and gaseous compounds, and deposition processes. Typically, nucleation mode particles (3 to ~20 nm) are freshly formed via the nucleation of gaseous molecules and ions (Brines et al., 2015; Charron and Harrison, 2003; Zhu et al., 2002a). Aitken (~20 to 100 nm) and accumulation (100 to 1000 nm) mode particles are often associated with primary emissions from combustion sources and condensation of secondary materials (Yue et al., 2009). Coarse mode particles (>1000 nm) generally result from mechanical processes, such as aerodynamic resuspension and abrasion. Nucleation mode particles can be removed relatively quickly via coagulation due to their high diffusivity (Hinds, 2012). They can also grow into the Aitken mode during new particle formation events (Cai et al., 2017; Xiao et al., 2015). Aitken mode particles may further form accumulation mode particles via coagulation and condensation. Accumulation mode particles can have a long lifetime due to their low gravitational settling velocity and slow coagulation rate among themselves. In the urban environment, nucleation and Aitken mode particles generally dominate number PSDs, due to the abundance of primary emission sources, such as power generation, traffic, and industrial activities. Their concentrations are high close to emission sources, while decreasing rapidly with distance from the source (Zhu et al., 2002a). Particle size can grow during transport by condensation of secondary materials. Coarse mode particles in the urban environment often contain road dust (Almeida et al., 2006), tire debris (Adachi and Tainosho, 2004; Rogge et al., 1993) and biological particles (e.g. pollen) (Saari et al., 2015). Due to their high gravitational settling velocities, their number concentrations can be 2-4 orders of magnitudes lower than other modes.

*2. Sec 9.2 "This is attributable to the abundant accumulation mode particles in CSSA due to strong air pollution" - suggest replacing "strong" with "high".*

>    **Response:** Upon the request from Anonymous Referee #6, this section has been removed.

*3. Sec. 11: "This is especially important given that UFP number concentrations and PM2.5 mass concentrations are not representative of each other, as particles that contribute to the two size-integrated metrics often originate from different sources" - suggest replacing "contribute to" with "dominate". (All sources necessarily \*contribute\* to both metrics, but their relative contributions can be quite different.)*

>    **Response:** The word "contribute" has been changed to "dominate".

*4. Sec 11: "The compilation of urban aerosol PSD observations in this review demonstrates the need for a transition from size-integrated PM2.5 mass concentration measurement to broader size range PSD measurements" - I think this goes too far. Suggest replacing "transition from" with "need to complement" PM2.5 mass concentration measurements with broader PSD measurements, because regulations are based on PM2.5 mass for which we have lots of epi data - which the authors show is lacking for PSDs because of the lack of long-term PSD measurements. Same comment for the next paragraph.*

**Response:** This sentence has been rephrased:

The compilation of urban aerosol PSD observations in this study demonstrates the benefit of routinely measuring urban PSDs that include the nucleation, Aitken, accumulation, and coarse modes.

Future urban aerosol PSD measurements should aim to span the entirety of the UFP regime. Achieving continuous urban aerosol number PSD observations from the nucleation to coarse modes at the global-scale remains a challenge given the cost of sensitive aerosol instrumentation required for the detection of UFPs and the collection of different measurement techniques needed to detect particles across such a wide size range. While advancements in low-cost optical particle sensing for detection of aerosols down to approximately $D_{em}$ = 300 to 500 nm have been made in recent years, efforts are still needed to develop low-cost condensation particle counters, differential mobility analyzers, and diffusion chargers for measurement of PSDs down to the UFP regime. The combination of routine PM$_{2.5}$ measurements with condensation particle counters that measure most of the UFP regime could potentially be a cost-effective approach to routinely monitoring both fine particle mass concentrations and UFP number concentrations in the near future.

*Note that many countries especially in the Global South do not even have PM2.5 mass monitoring; saying that is futile and pushing for more expensive PSD measurements seems counter-productive. (A BAM costs $20k; SMPS is ~$80k!)*

**Response:** We agree with the reviewer. This paragraph has been rephrased.

Future urban aerosol PSD measurements should aim to span the entirety of the UFP regime. Achieving continuous urban aerosol number PSD observations from the nucleation to coarse modes at the global-scale remains a challenge given the cost of sensitive aerosol instrumentation required for the detection of UFPs and the collection of different measurement techniques needed to detect particles across such a wide size range. While advancements in low-cost optical particle sensing for detection of aerosols down to approximately $D_{em}$ = 300 to 500 nm have been made in recent years, efforts are still needed to develop low-cost condensation particle counters, differential mobility analyzers, and diffusion chargers for measurement of PSDs down to the UFP regime.

*Finally, given the emphasis on UFP as distinct from PM2.5 mass, perhaps an alternative can be complementing PM2.5 mass monitoring networks with CPCs or other UFP measuring devices. An SMPS/OPS combination runs close to $100k, while a CPC can be below $20k. A BAM and a CPC are also easier to operate than an SMPS, which is an important consideration for routine monitoring worldwide.*

**Response:** The following sentence has been added.

[revised manuscript text omitted]